# Somatic mutations of *CADM1* in aldosterone-producing adenomas and gap junction-dependent regulation of aldosterone production

Aldosterone-producing adenomas (APAs) are the commonest curable cause of hypertension. Most have gain-of-function somatic mutations of ion channels or transporters. Herein we report the discovery, replication and phenotype of mutations in the neuronal cell adhesion gene *CADM1*. Independent whole exome sequencing of 40 and 81 APAs found intramembranous p.Val380Asp or p.Gly379Asp variants in two patients whose hypertension and periodic primary aldosteronism were cured by adrenalectomy. Replication identified two more APAs with each variant (total, *n* = 6). The most upregulated gene (10- to 25-fold) in human adrenocortical H295R cells transduced with the mutations (compared to wildtype) was *CYP11B2* (aldosterone synthase), and biological rhythms were the most differentially expressed process. *CADM1* knockdown or mutation inhibited gap junction (GJ)-permeable dye transfer. GJ blockade by Gap27 increased *CYP11B2* similarly to *CADM1* mutation. Human adrenal zona glomerulosa (ZG) expression of GJA1 (the main GJ protein) was patchy, and annular GJs (sequelae of GJ communication) were less prominent in CYP11B2-positive micronodules than adjacent ZG. Somatic mutations of *CADM1* cause reversible hypertension and reveal a role for GJ communication in suppressing physiological aldosterone production.

Primary aldosteronism (PA) is a common cause of hypertension and is surgically curable when it is a consequence of an aldosterone-producing adenoma (APA) in one of the adrenal glands[1]. Whereas physiological adrenal secretion of aldosterone is regulated by (and inversely related to) salt intake, the autonomous aldosterone production by APAs is due to hallmark somatic mutations[2,3]. Pathogenic *KCNJ5* variants have the highest apparent prevalence in most cohorts of PA patients except African Americans[4,5]. However, these *KCNJ5* variants are usually found in relatively large APAs[6], which are the easiest to recognize radiologically, but whose cells paradoxically resemble the cortisol- rather than aldosterone-secreting zone of the normal adrenal gland[7]. APAs with pathogenic *CACNA1D*, *ATP1A1* and *ATP2B3* variants are typically smaller than *KCNJ5*-mutant APAs and

resemble the physiological aldosterone-secreting cells of the adrenal zona glomerulosa (ZG)[8–10]. These smaller, ZG-like APAs are often overlooked on a CT scan, or considered equivocal on routine adrenal pathology. However, once a specific antibody for CYP11B2 (aldosterone synthase) became available, as ex vivo ligand, their identity and frequency were readily confirmed, and an inverse correlation between signal and size became apparent[8,10–13]. Their radiological recognition was facilitated by the development of dexamethasone-([11]C)-metomidate and ([18]F)-chloro-2-fluoroethyletomidate PET-CT as in vivo ligands for CYP11B2 (refs. [14–16]). We, therefore, wished to determine whether CYP11B2-dense small APAs have new mutations and whether (depending on patient demography and ascertainment) *KCNJ5*-mutant APAs may now be outnumbered by other genotypes. Whole-exome sequencing (WES) of

✉e-mail: elena.azizan@ukm.edu.my; morris.brown@qmul.ac.uk

sequential APAs indeed found only nine with pathogenic *KCNJ5* variants versus 22 with other genotypes. These included a new combination of *GNA11* and *CTNNB1* variants, whose discovery, replication and phenotype have been separately reported[17]. A further APA had a somatic mutation of the cell adhesion molecule 1 (*CADM1*) gene. Herein we report the phenotype and replication of this mutation in other cohorts. Studies of CADM1 function in other tissues[18–21], and the discovery of deleterious mutation in APAs, led us to several unsuspected pathways of aldosterone regulation. Of these, we concentrated on gap junction (GJ)-dependent regulation of aldosterone production. As cell–cell communication in adrenal ZG is considered critical for regulating aldosterone production[22–26], we hypothesized that the link between cell–cell interaction and aldosterone production may be through GJs, which form only when the two hemichannels of GJ (called connexons) on apposed cells come together[27,28]. Our hypothesis was prompted by the suppression of GJ intracellular communication (GJIC) and increased production of glucagon that follow the silencing of *CADM1* in pancreatic islet cells[18].

## Results

### WES of 40 APAs identifies a new *CADM1* somatic mutation

WES was performed in 40 APAs from predominantly European ancestry patients with PA (Supplementary Table 1). The APAs were consecutive, except for two that antedated our previous WES (*n* = 13)[8] and two that were excluded by prior targeted sequencing; 31 had a known aldosterone-driver somatic mutation, including 11 *CACNA1D* and nine *KCNJ5* variants (Supplementary Tables 1 and 2). A small (13 × 7 mm) APA, with dense CYP11B2 expression, had a new mutation in the intramembranous portion of *CADM1* (P1 of Fig. 1a–c, Extended Data Fig. 1 and Supplementary Table 3). The only prior association of *CADM1* with aldosterone synthesis was as one of the transcripts upregulated in APAs[29]. All known somatic mutations and the new mutation in *CADM1* were confirmed by Sanger sequencing. The likely relevance of the *CADM1* variant was apparent from its intramembranous position and SIFT score of 0, suggesting a deleterious substitution. The resected adrenal showed the typical features of a ZG-like APA on histopathology (Extended Data Fig. 1a). Immunohistochemistry (IHC) showed dense membranous staining of CADM1 both in areas staining strongly for CYP11B2 and in the adjacent ZG and adrenal medulla (Extended Data Fig. 1b,c), supporting previous reports of twofold to fourfold upregulation of *CADM1* RNA expression in ZG and APAs compared to zona fasciculata (ZF)[29–31].

### Discovery of recurrent *CADM1* somatic mutations in APAs

Initial search for further examples of *CADM1* variants was undertaken by Sanger sequencing. Fifty-three APAs, negative for known variants in previous targeted sequencing studies, were sourced from four centers in the UK and Japan. No pathogenic *CADM1* variants were found in these. We then re-interrogated a previous WES of 81 APAs from a German cohort, in which no gene with recurrent somatic mutations had been found, other than those previously reported (Supplementary Table 4). One of the 81 APAs had a *CADM1* somatic mutation, p.Gly379Asp, altering the adjacent residue to the one altered in the index case (P2 of Fig. 1b,c). Patients P1 and P2 were both males in their forties whose resistant hypertension was cured by adrenalectomy (Table 1). Both appeared to have episodic hyperaldosteronism, detected by serial aldosterone (Aldo) and renin measurements that identified variable diagnostic aldosterone–renin ratios (ARR).

Further investigation by targeted sequencing of 43 APAs without known aldosterone-driver mutations from a French cohort identified three further cases, one with p.Val380Asp and two with p.Gly379Asp variants in *CADM1* (P3–P5 of Fig. 1c and Table 1). Finally, a sixth case was discovered by targeted sequencing of 200 APAs in a US cohort, harboring the p.Val380Asp variant (P6 of Table 1). The range of prevalence of the *CADM1* variants is between 0.5% and 1.0% (Supplementary Table 4), with 0.93% in the largest cohort harboring three variants.

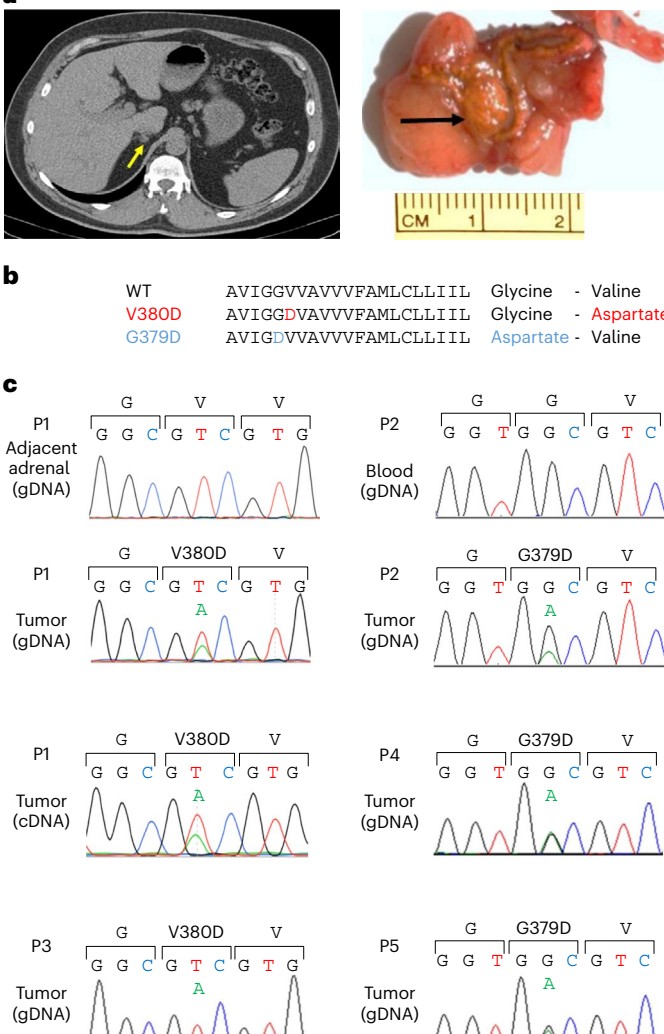

**Fig. 1 | Discovery of *CADM1* somatic mutations in APAs. a**, APA of patient P1 as seen on CT scan (yellow arrowhead) and in adrenal tissue (black arrowhead). The axial CT image of patient P1's adrenal identified a 13 × 7 mm right adrenal nodule. Macroscopic view of 5-mm adrenal slices reveals the solitary adenoma. IHC imaging of *CADM1*-mutant APAs from patients P1–P5 are shown in Extended Data Figs. 1 and 2. **b**, Affected protein residues in the TM domain of CADM1. Protein sequence showing the mutations in adjacent amino acids. **c**, Sanger sequencing chromatograms of the two *CADM1* somatic mutations found in APAs (P1–P5). The two somatic mutations in *CADM1* translate to a p.Val380Asp (V380D) and p.Gly379Asp (G379D) mutant CADM1. The somatic mutations were found in neither the blood (of P2) nor the adjacent adrenal gland gDNA (of P1). cDNA sequence of APA from patient P1 suggests expression of both WT and mutant CADM1 protein in the adenoma.

IHC of the available APAs showed strong staining for CYP11B2 and weak staining for CYP11B1, supporting a ZG-like APA histopathology (Extended Data Fig. 2a,b). IHC of these APAs also found dense membranous staining of CADM1 similar to the index case (Extended Data Fig. 2c). Taken together with the finding of the mutation in the cDNA of the index APA (P1 of Fig. 1c), the consequences of these *CADM1* variants appear due to expression of an abnormal protein rather than an absence of expression, as occurs in some malignant tumors[32].

### Functional analyses of *CADM1* variant in human adrenal cells

*CADM1* has been known by various names, each reflecting an aspect of its function, such as synaptic cell adhesion molecule (SynCAM1) and immunoglobulin superfamily member 4 (IGSF4)[32–36].

**Table 1 | Blood sample measurements of PA patients with *CADM1*-mutant APAs**

| ID | Sex | Age at adrenalectomy | *CADM1* variant | Pre-adrenalectomy | | | Postadrenalectomy | | |
|---|---|---|---|---|---|---|---|---|---|
| | | | | Aldo (pmol l⁻¹) | Renin (mU l⁻¹) | ARR (pmol mU⁻¹) | Aldo (pmol l⁻¹) | Renin (mU l⁻¹) | ARR (pmol mU⁻¹) |
| P1 | M | 48 | V380D | 573 | 10.0 | 57.3 | 143 | 27.0 | 5.3 |
| | | | | 147 | 11.0 | 13.4 | | | |
| | | | | 537 | 10.0 | 53.7 | | | |
| | | | | 647 | 5.0 | 129.4 | | | |
| P2 | M | 45 | G379D | 363 | 16.6 | 21.9 | 92 | 12.5 | 7.3 |
| | | | | 791 | 3.4 | 232.5 | | | |
| | | | | 563 | 8.7 | 64.7 | | | |
| P3 | F | 37 | V380D | 570 | <5 | >114.0 | N/A | N/A | N/A |
| P4 | M | 57 | G379D | 548 | <5 | >109.6 | 114 | 5.0 | 22.8 |
| P5 | M | 47 | G379D | 524 | <5 | >104.8 | 337 | 40.7 | 8.2 |
| P6 | F | 52 | V380D | 860 | 3.28ᵃ | 262.2 | 241 | 13.2ᵃ | 18.3 |

Serial Aldo, renin and ARR measurements of PA patients with *CADM1*-mutant APAs (patient's ID: P1–P6), pre-adrenalectomy and postadrenalectomy. Note: fluctuating ARR for patient P1 and for patient P2, with only one positive ARR diagnostic for PA. ᵃValues shown are on the basis of a conversion factor of PRA (ng ml⁻¹ h⁻¹) to DRC (mU l⁻¹) of 8.2. ARR cut-off value used: 91 pmol mU⁻¹.
F, female; M, male; N/A, not available.

The immunoglobulin ectodomains pair with those of adjacent cells, but can also—like those of many single transmembrane (TM) domain proteins—be shed, by ADAM10, and maybe ADAM17 (ref. [19]). Isoforms of CADM1 vary in susceptibility to shedding[20], with the 11 residues of exon 9 providing a nonglycosylated 'stalk' facilitating access to the sheddase. The two isoforms established by reverse transcription PCR (RT–PCR) and sequencing to be most abundant in human adrenal, both in APA and adjacent cortex, are encoded by 10 or 11 of the full-length 12 exons (442 and 453 amino acids; Supplementary Fig. 1). In most of the analyses performed, both isoforms of both variants were studied.

To determine whether the mutations in *CADM1* influence aldosterone production, human adrenocortical H295R cells were transduced with wild-type (WT), mutant or sh-*CADM1* (Fig. 2a–c). It was interesting to note that human embryonic kidney HEK293T cells (used for lentivirus production) transfected with WT *CADM1* appeared in clusters, while a more uniform monolayer of cells was observed in cells transfected with empty vector (EV) and mutant *CADM1* (Fig. 2a). Cells transduced with mutant *CADM1* increased expression of *CYP11B2*, assessed by quantitative PCR (qPCR), by 10- to 24-fold (Fig. 2d), which is many times larger than the typical effect of aldosterone-driver mutations in ion channels or transporters[37–40]. The enhanced expression of *CYP11B2* was paired with an increase in aldosterone production, although of smaller magnitude (2- to 4-fold) than that of the enzyme (Fig. 2e). Conversely, silencing of *CADM1* reduced both aldosterone secretion and expression of *CYP11B2* (Fig. 2d,e).

**Protein modeling predicts effects on the tertiary structure**

The shed ectodomains (or a secreted, truncated isoform) can compete with the intact ectodomains and regulate the role of CADM1 in adhesion[41]. Intracellular actions are also important. Either the remnant C-terminal fragment (CTF), after shedding and further cleavage by γ-secretase, can traffic to and activate pathways within cell organelles (for example nucleus, mitochondria)[19,42,43], or CADM1 may be activated by dimerization at the TM domain[44,45]. Indeed, the mutations we found lie within the key *AviGGvia* motif predicted to be the dimerization site in a cluster of 11 single TM domain proteins[44]. Western blots of various cell types transfected with WT or mutant *CADM1* were undertaken to investigate the effects of the mutations on ectodomain shedding. Mouse 3T3 cells were studied initially as they lack native CADM1. The variants of both isoforms caused substantial loss of full-length CADM1, attributable in part to increased shedding, as evidenced by increased N-terminal fragments (NTF) in the cell medium (Supplementary Fig. 2a).

There was no definite increase in CTFs, and the clearest difference was a decrease in their size. No γ-secretase products could be detected.

Transfected human adrenocortical (H295R) cells showed less difference than 3T3 cells between WT and mutant *CADM1*, except for the shorter CTF (Fig. 3a and Supplementary Fig. 2b,c). As their abundance was low compared to full-length CADM1, the interest in their smaller size arises from the following putative explanation. ADAM10 (and ADAM17) are intramembranous enzymes (sheddases) with an ectodomain that includes the active site. No consensus cleavage sequence is established for their substrates, which are cleaved at a defined distance outside the plasma membrane[46,47]. Because both enzyme and substrate are semifixed in position, the cleavage point may be determined by where the substrate meets the enzyme active site[19]. This point is predicted to be in close proximity to the membrane surface, ~10 to 20 residues beyond the TM domain (Fig. 3b), with the exact site depending on the angle of exit. A perpendicular ectodomain will be cut within fewer residues from the membrane than an angulated ectodomain, yielding a shorter CTF (Fig. 3c). Introduction of a charged, unpaired aspartic acid residue (G379D and V380D) in the hydrophobic lipid bilayer is likely to affect the tilt angle of the TM domain. Indeed, two independent molecular modeling exercises, based on structures of other single TM domain proteins, predicted a doubling of the angle, from 49° for WT to 90° for V380D (Fig. 3d,e and Supplementary Fig. 2d,e). In the first model, straightening of the TM domain was associated with the shortening of the mutant TM helix in dimers with WT CADM1 (Supplementary Fig. 2d). The second model found mutant homodimers to be straighter (more perpendicular to the membrane) than WT (Supplementary Fig. 2e).

**CADM1 variants inhibit gap junction intercellular communications (GJICs)**

Transcriptome analysis of laser-dissected cortical zones showed that *GJA1* (encoding GJ alpha-1, formerly connexin 43) is the predominant GJ mRNA expressed in the human adrenal cortex, only threefold lower, on average, in ZG and APAs than in ZF (Supplementary Table 5a)[31,48]. The array data from a similar study showed the highest expression in ZF and zona reticularis, and twofold to threefold lower levels in ZG and aldosterone-producing micronodules (APMs, previously known as aldosterone-producing cell clusters or APCCs)[49].

To explore whether *CADM1* regulates GJ formation and communication, we studied the transfer between cells of a GJ-permeable dye calcein red, as a measure of GJIC. Transfer of calcein red was inhibited

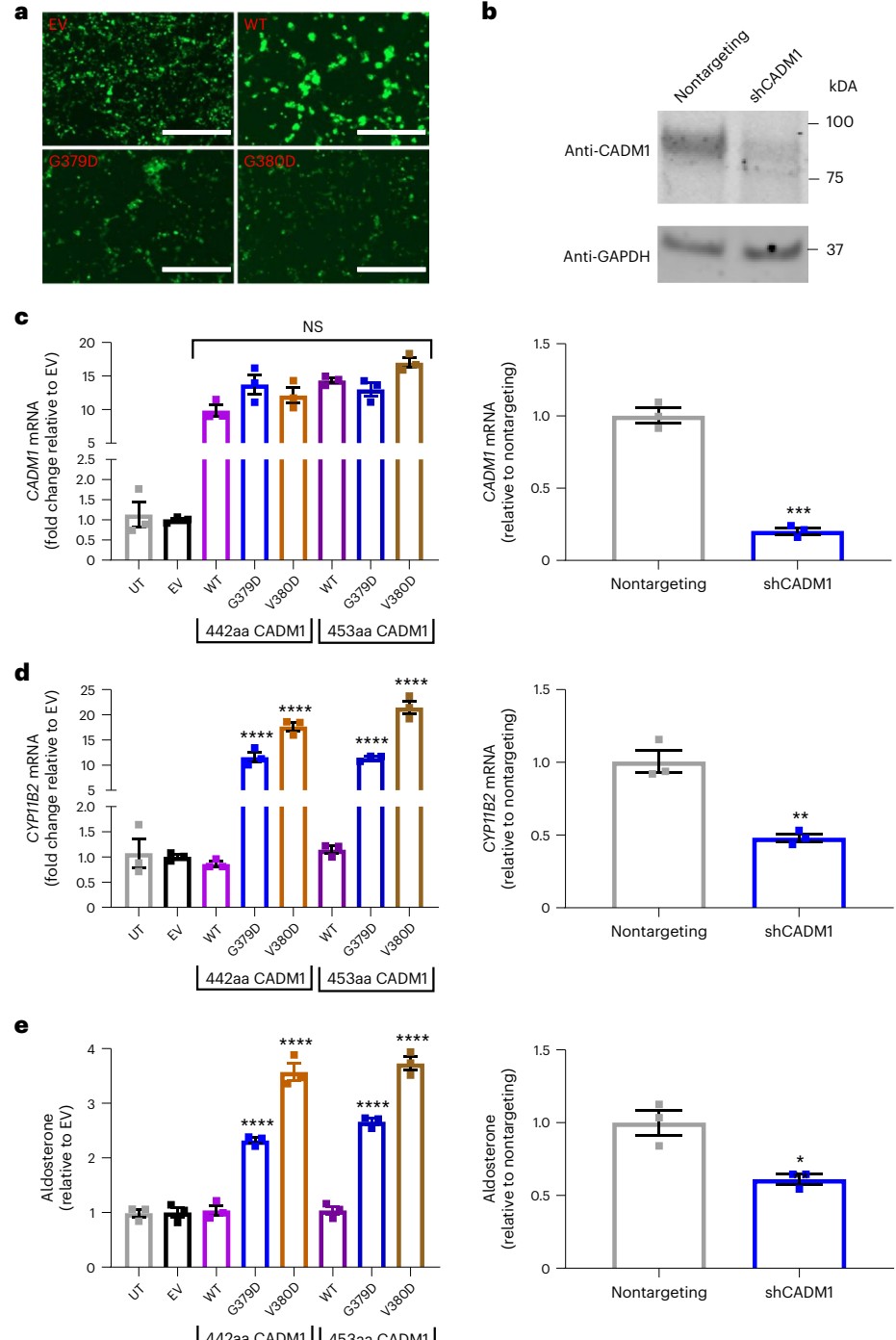

**Fig. 2 | *CADM1* variants increase *CYP11B2* expression and aldosterone production. a**, Transfection of EV, WT, G379D and V380D *CADM1*. Fluorescent images of HEK293 cells transfected with pLOC EV, WT or mutant *CADM1* during the production of lentiviruses for transduction of *CADM1* into H295R cells. Transfected cells expressed tGFP (green) present in the pLOC vector. Cells overexpressing WT *CADM1* appeared in clusters, whereas a more uniform, monolayer distribution of cells was observed in cells overexpressing mutant *CADM1* or EV. Scale bar, 400 μm. **b**, Silencing of *CADM1* by shRNA. Western blot of H295R cells transduced with EGFP-tagged shRNA either nontargeting or targeting *CADM1* (sh*CADM1*). Immunoblotting was performed with anti-CADM1 (Sigma-Aldrich, S4945) and anti-*GAPDH* antibodies on the same blot. Immunoblots showed a reduction in CADM1 protein in sh*CADM1* transduced cells compared to nontargeting transduced cells with similar total protein (estimated by GAPDH expression). The experiment was repeated once independently with similar results. Full-length blots are provided as Source Data Fig. 2. **c**–**e**, *CADM1* mRNA expression (**c**), *CYP11B2* mRNA expression (**d**) and

aldosterone production (**e**), each in human adrenal cells overexpressing WT or mutant *CADM1* or silenced for *CADM1*. Transduction of *CADM1* (G379D or V380D) as its short (442 amino acids) or long (453 amino acids) isoform increased *CYP11B2* (encoding aldosterone synthase) and aldosterone production. No effect was seen with overexpression of WT *CADM1*, whereas a decrease in *CYP11B2* expression and aldosterone production was seen with silencing *CADM1* (*shCADM1*). All values are expressed as fold change relative to EV or nontargeting control cells. Data are presented as mean values ±s.e.m. $n = 3$ independent wells. Statistical analysis was performed using one-way ANOVA followed by Dunnett's multiple comparisons test for overexpression experiments. For *CADM1* mRNA expression, $F = 49.00, P < 0.0001$; for *CYP11B2*, $F = 158.4, P < 0.0001$; for aldosterone, $F = 149.0, P < 0.0001$. Two-sided Student's *t*-test was performed on silencing experiments. When compared to WT or sh*CADM1*, *$P = 0.0130$, **$P = 0.0029$, ***$P = 0.0003$, ****$P < 0.0001$. NS, not significantly different between WT and mutant *CADM1*. The statistics used to produce these plots are provided as Source Data Fig. 2. UT, untransduced cells.

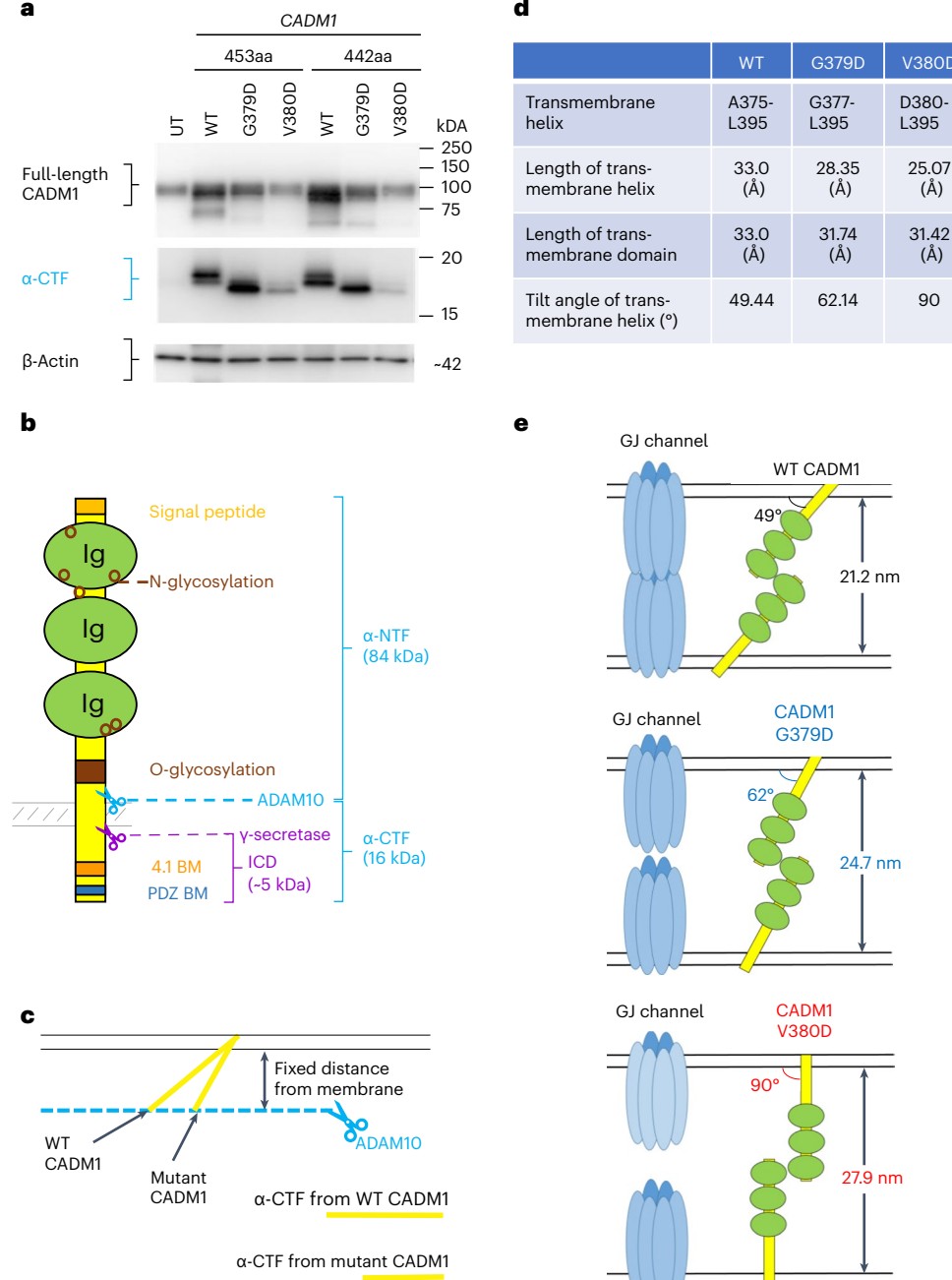

| | WT | G379D | V380D |
|---|---|---|---|
| Transmembrane helix | A375-L395 | G377-L395 | D380-L395 |
| Length of trans-membrane helix | 33.0 (Å) | 28.35 (Å) | 25.07 (Å) |
| Length of trans-membrane domain | 33.0 (Å) | 31.74 (Å) | 31.42 (Å) |
| Tilt angle of trans-membrane helix (°) | 49.44 | 62.14 | 90 |

**Fig. 3 | Mutant *CADM1* affects protein structure leading to changes in intercellular distance. a**, CADM1 variants have shorter α-CTF. Western blot of H295R cell lysates transfected with WT or mutant *CADM1* (G379D or V380D) in a pCX4bsr vector using a custom-made anti-CADM1 C-terminal antibody. Shown are the protein bands for glycosylated full-length CADM1 (~100 kDa) and α-CTF CADM1 (15–20 kDa). Complete immunoblot for CADM1 is shown in Supplementary Fig. 2b. Total protein was estimated by immunoblotting β-actin. The experiment was repeated twice independently with similar results. Full blots are provided as Source Data Fig. 3. **b**, ADAM10/γ-secretase-mediated cleavage of CADM1. Schematic representation of the CADM1 protein. Ectodomain shedding due to cleavage by proteases can result in the formation of intracellular CTF and secreted NTF. Cleavage by the protease ADAM10 after the O-glycosylation site leads to the formation of α-CTF (blue scissors), whereas cleavage before the O-glycosylation site leads to β-CTF. The CTF can undergo further cleavage by γ-secretase (purple scissors) to release an intracellular domain. Glycosylation

sites (N- and O-) are shown in brown. The 4.1- and PDZ-binding motifs are shown in orange and blue, respectively. Ig, Immunoglobulin domain. Schematic adapted from refs. 19,35. **c**, Change in angle of the TM helix in mutant CADM1 can result in shorter α-CTF. Schematic diagram showing that the fixed distance that ADAM10 cleaves CADM1 from the cell membrane (dashed blue line with scissors) and an increase in the angle of the TM helix in mutant CADM1 could result in a shorter length of cleaved α-CTF. **d**, Predicted changes in TM helix in mutant CADM1 compared to WT. Effect of the CADM1 variants on angle and length of the TM helix in the cell membrane lipid bilayer as predicted by protein modeling data. The 3D structures of the TM domain (residues, A375–L395) of WT, G379D mutant and V380D mutant CADM1 were analyzed by using QUARK program (https://zhanggroup.org/QUARK/). **e**, Predicted structural consequences of mutant CADM1 on intercellular distance. Schematic representation of change in angle resulting in an increase in intercellular distance in mutant cells.

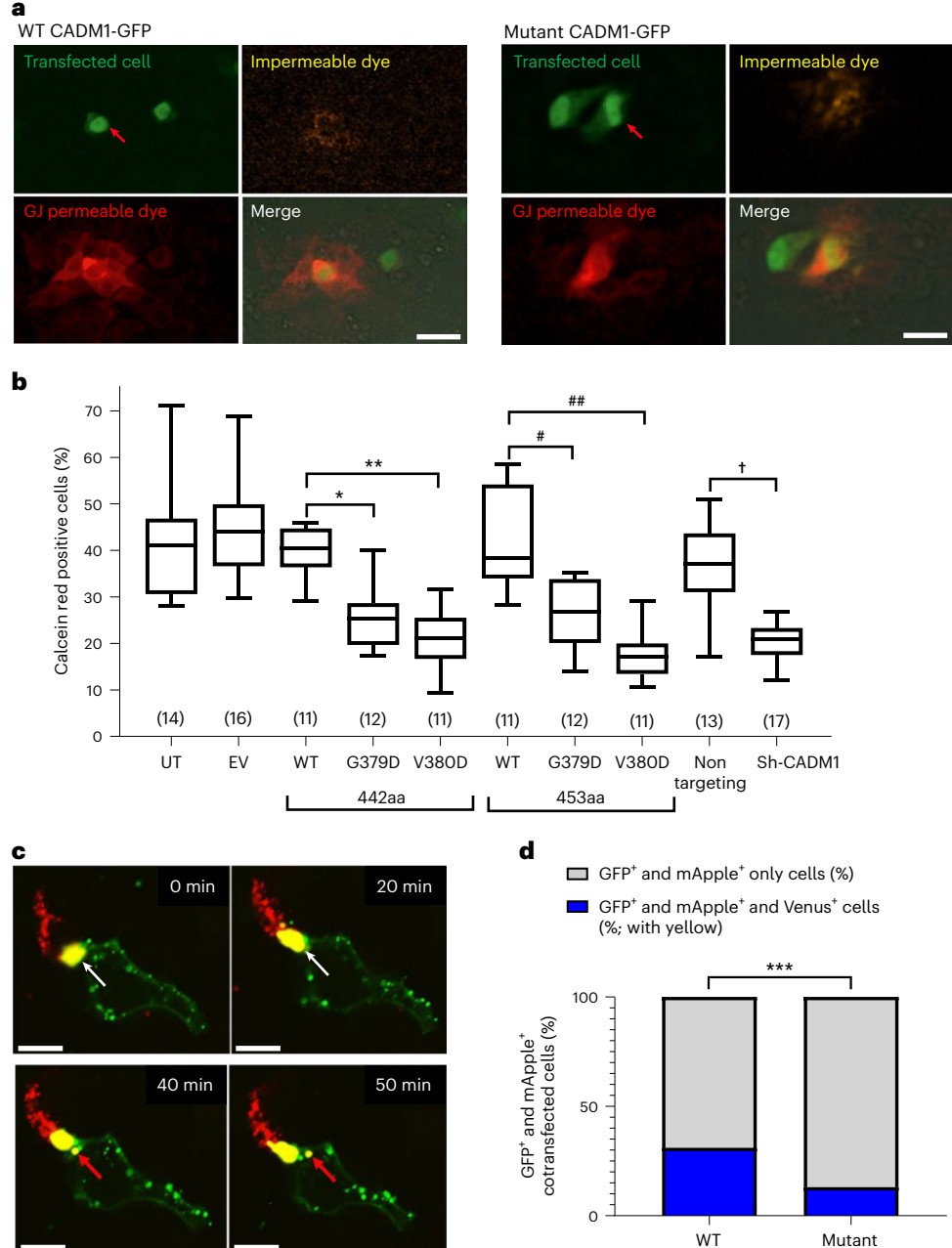

**Fig. 4 | *CADM1* variants inhibit GJ communication. a**, GJ-mediated communication as detected by a GJ-permeable dye. H295R cells were transfected with either WT or mutant CADM1-GFP vectors. A single transfected cell (red arrowhead) was then injected with the GJ-impermeable dye (WGA) and the GJ-permeable dye (calcein red). Representative images of cells with the 442-amino acid isoform of CADM1 1-h post-dye injection are shown. Image is representative of 11 independent experiments. Representative images at 0 h are shown in Supplementary Fig. 3a. Scale bar, 20 µm. **b**, GJ-mediated communication reduced in cells expressing mutant CADM1. Quantification of the experiment performed in **a**. The percentage of calcein red-containing cells within a 50-µm radius from an injected cell (green nucleus, marked by orange WGA dye in the cytoplasm) was calculated per total number of cells within the radius. There were up to twofold fewer red cells around a CADM1 variant transfected cell compared to WT transfected cells. Center line represents the median. Upper and lower bounds of box represent interquartile range. Upper and lower whiskers represent maximum and minimum values in the range, respectively. Statistical analysis was performed using one-way ANOVA ($F = 20.68$, $P < 0.0001$) and Sidak's multiple comparison test. *$P = 0.028$, **$P < 0.0001$, #$P = 0.0120$, ##$P < 0.0001$, †$P < 0.0001$. The number

($n$) of dye-injected cells per experimental group is shown in parentheses. **c**, GJ plaque formation (yellow) detected using GJA1-mApple (red) and GJA1-Venus (green). Time-lapse imaging of cocultured H295R cells transfected with either GJA1 tagged with mApple (red) or GJA1 tagged with Venus (green) was performed to study GJ plaque formation. The four serial frames illustrate GJA1-mApple and GJA1-Venus colocalizing (yellow), indicating GJ plaque formation. Internalization of plaque (formation of an annular GJ) is highlighted by white arrowheads (pre-internalization) and red arrowheads (postinternalization). Scale bar, 9 µm. **d**, GJ plaque formation reduced in cells expressing mutant CADM1. Quantification of GJ formation in H295R cells cotransfected with GJA1-mApple and either WT ($n = 212$) or mutant *CADM1* (mutant, $n = 291$) vectors tagged with GFP. GJ communication was detected by internalization of GJA1-Venus in cells coexpressing GJA1-mApple and GFP, shown as a percentage of mApple and GFP expressing cells. Statistical analysis was performed using two-sided Fisher's exact test. ***$P = 0.000205$; 95% CI, 1.423–3.147. The statistics used to produce these plots are provided as Source Data Fig. 4. CI, confidence interval. WGA, wheat germ agglutinin.

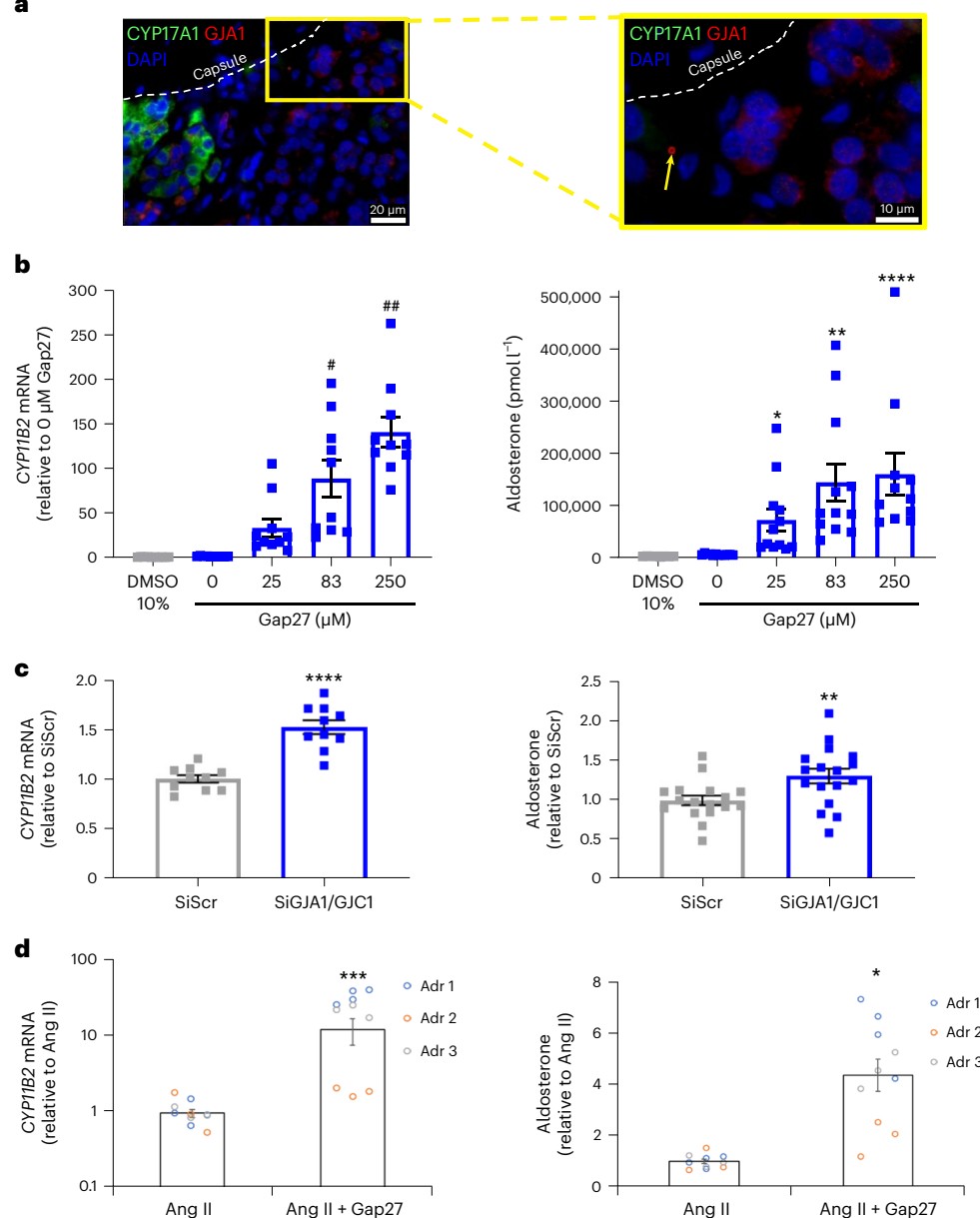

**Fig. 5 | Inhibition of GJ communication increases aldosterone production.**
**a**, GJA1 expressed in subcapsular non-ZF cells. Human adrenal section stained with mouse anti-CYP17A1 (green) and rabbit anti-GJA1 (red) antibodies. AGJ (yellow arrowhead) is present in subcapsular cells not expressing the ZF marker CYP17A1. This region immunostain for DAB2 but not for CYP11B2 (white box in Supplementary Fig. 6a). Dashed line demarcates border with capsule. Left image, ×63 magnification; right image, ×100 magnification. **b**, Connexin mimetic peptide Gap27 increases *CYP11B2* expression and aldosterone production in Ang II-stimulated H295R cells. *CYP11B2* (*n* = 10) and aldosterone (*n* = 12 except for 250 μM Gap27, *n* = 11) are increased in stimulated H295R cells treated with Gap27, which selectively blocks GJ communications. Results expressed as fold change relative to cells treated with 0 μM Gap27. The effect of Gap27 on unstimulated cells is shown in Supplementary Fig. 8a. Ten percent dimethyl sulfoxide (DMSO 10%) treatment on unstimulated cells (*n* = 8) was used as control for enhanced cell membrane permeability. Statistical significance measured using the Kruskal–Wallis *H* test; *CYP11B2*, $\chi^2(4) = 43.03$, $P = 1.02 \times 10^{-8}$ and aldosterone, $\chi^2(4) = 42.25$, $P = 1.48 \times 10^{-8}$, respectively. Post hoc testing was performed using Dunn's multiple comparison test (compared to 0 μM Gap27).

For *CYP11B2*, #*P* = 0.0039, ##*P* = 0.002. For aldosterone, *P = 0.0310, ***P = 0.0005, ****P < 0.0001. **c**, Silencing of GJ increases *CYP11B2* expression and aldosterone production. *CYP11B2* (*n* = 10) and aldosterone (*n* = 17) is increased in H295R cells with decreased GJ communications due to cosilencing of the genes *GJA1* and *GJC1* (*SiGJA1/GJC1*) compared to the silenced scramble RNA control (*SiScr*). *GJA1* and *GJC1* mRNA and protein expression is shown in Supplementary Fig. 8e,f. Results expressed as fold change relative to SiScr cells. Statistical analysis performed using two-sided Student's *t*-test. **P = 0.0097, ****P < 0.0001. **d**, Gap27 increases *CYP11B2* expression and aldosterone production in Ang II-stimulated primary adrenal cells. *CYP11B2* (*n* = 10) and aldosterone (*n* = 10) are increased in stimulated primary adrenal cells treated with Gap27. Effect of Gap27 in unstimulated primary adrenal cells is shown in Supplementary Fig. 8g. mRNA expression was normalized by β-actin. Results expressed as fold change relative to Ang II-stimulated cells. Statistical analysis was performed using two-sided Student's *t*-test. ****P = 0.0005, *P = 0.0438. Data are presented as mean ± s.e.m. *n* = biological independent replicates from three independent experiments. The statistics used to produce these plots are provided as Source Data Fig. 5.

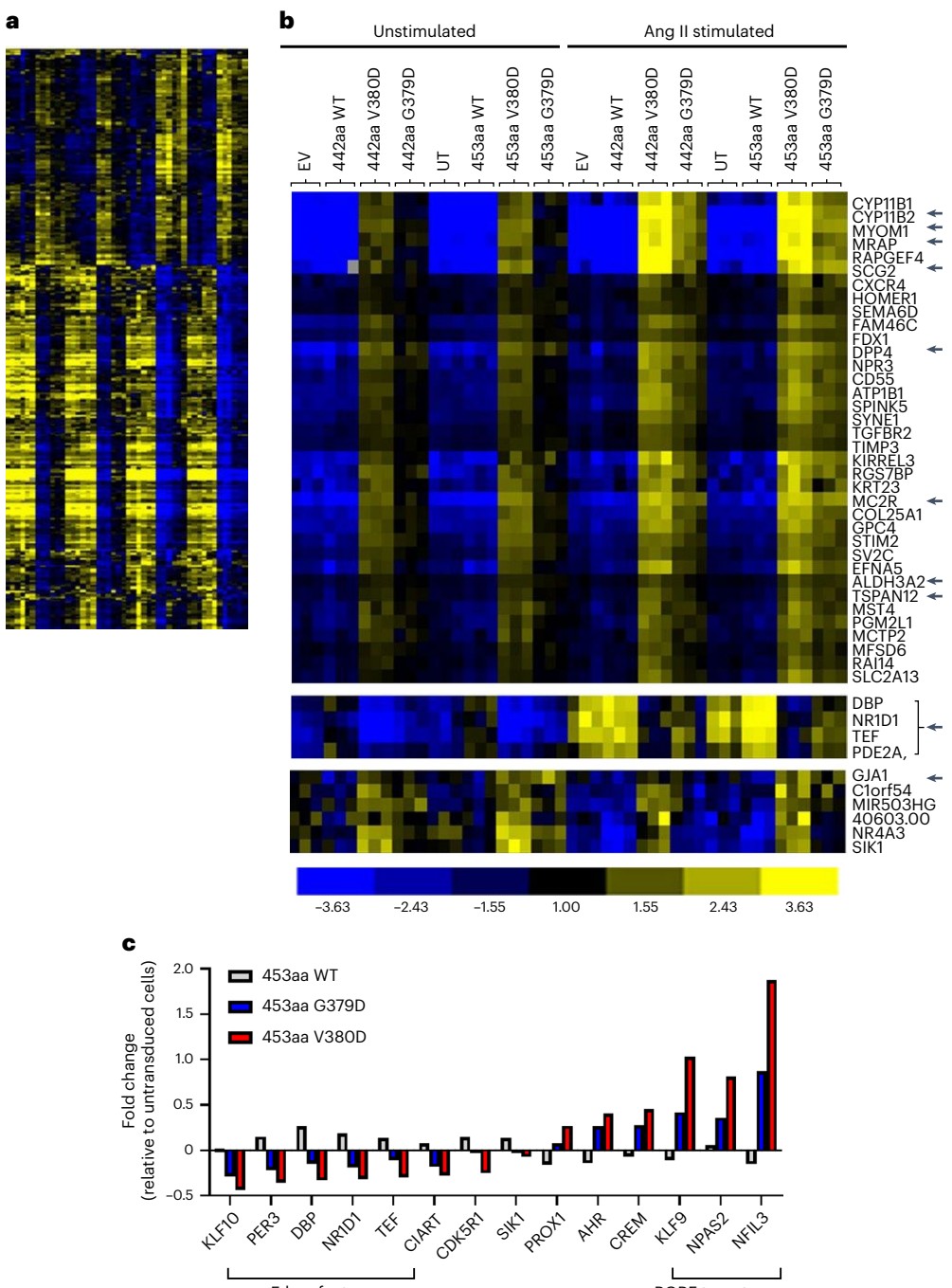

**Fig. 6 | *CADM1* variants affect genes associated with the 'biological rhythms' process. a**, Heatmap of differentially expressed genes in *CADM1*-mutant cells. Heatmap of differentially expressed genes in H295R cells transduced with EV, WT (442- and 453-amino acid isoforms), mutant *CADM1* (V380D and G379D in both isoforms) or untransduced (UT). The columns represent different conditions (in triplicates) in the transduction experiments as labeled in **b**. Each row represents a gene whose mean expression was either upregulated >1.5-fold or downregulated <0.7-fold in variant transduced cells, compared to WT (two-sided one-way ANOVA *P* < 0.001). Centroid clustering was performed using cluster 3. Yellow represents upregulation and blue represents downregulation of genes as specified by color bar in **b**. The full heatmap is provided as Source Data Fig. 6. **b**, Zoomed image of heatmap in **a** displaying top 36 genes and clusters including genes associated with 'biological rhythms' process and GJA1. The top 36 genes were differentially regulated whether unstimulated (left half of heatmap) or

stimulated by Ang II (right half, 10 nM for 24 h). However, for genes associated with 'biological rhythms' (highlight by bracket with arrow), the differential regulation was more apparent when the transduced cells were stimulated by Ang II. To note, transduction of H295R cells with both G379D and V380D mutant *CADM1* substantially increased *GJA1* mRNA expression in both isoforms. Genes of interest that are mentioned in the main text are arrowed. **c**, Genes associated with 'biological rhythms' process were differentially regulated in *CADM1*-mutant cells. RNA-seq of mutant *CADM1*-transduced cells reveals the most upregulated process associated with the 425 differentially expressed genes was 'biological rhythms'. Clock genes associated with E-box factors and RORE target genes are highlighted by brackets. Graph shows the fold change of mRNA expression relative to UT cells for genes associated with this process in WT and mutant *CADM1* (453-amino acid isoforms).

in cells transfected with mutant *CADM1* (G379D or V380D; Fig. 4a,b and Supplementary Fig. 3a). Silencing of *CADM1* also inhibited transfer (Fig. 4b and Supplementary Fig. 3b,c). Similar inhibition of dye transfer was obtained by soluble recombinant CADM1 ectodomains (Supplementary Fig. 3d,e; $16.7 \pm 4.2\%$ versus $38.9 \pm 9.6\%$ for Fc control, $n = 5$, $P = 0.008$ by Mann−Whitney $U$ test)[41]. These three cases are consistent with the homophilic adhesion of CADM1 ectodomains enabling connexons of opposing cells to make contact. When homophilic adhesion is inhibited, whether by silencing, competition from recombinant ectodomains, or shedding of NTF (effectively equals ectodomains), GJIC is suppressed. However, the increased shedding of NTF from mutant CADM1 transfected into mouse cells (Supplementary Fig. 2a) was not seen in medium from variant-transduced H295R cells (Supplementary Fig. 2c). An alternative explanation for the inhibition of GJIC by *CADM1* variants is that the predicted straightening of their ectodomains would push apposed cells beyond the reach of their connexons, which is an estimated maximum of 20–30 nm (Fig. 3d,e)[50–52].

To show directly whether the inhibition of GJIC is due to failure of GJ channel formation, we made use of the dynamic GJIC process, which includes the internalization of a GJ plaque from two adjacent cells, followed by the formation of an annular GJ (AGJ) in one of the two cells comprising of GJ protein from both. This was done by labeling GJA1 with either the mApple or Venus fluorophores, and separate transfections of H295R cells with each of these constructs. Time-lapse of these cells cocultured together (Supplementary Video) found that GJ formation between adjacent cells (identified by co-expression of mApple and Venus fluorophores) was transient, and the formation of an AGJ was evidence of the brief encounter (Fig. 4c). To explore the effect of the *CADM1* variants on GJ formation, GJA1-mApple cells were also cotransfected with GFP-labeled WT or mutant *CADM1*. In this experiment, only a minority of cells would express the three fluorophores (GFP, Apple and Venus)−those that (1) were double-transfected with *CADM1*-GFP and GJA1-mApple, (2) had come into contact with a GJA1-Venus transfected cell and (3) had the AGJ internalized in the double-transfected cell (Supplementary Fig. 3f). In the 24-h post-transfection images used for quantification, cell-contact had usually passed, leaving the triple-colored, AGJ-containing cells to be counted, as a proportion of all double-transfected cells. Such cells were more numerous with WT compared to mutant *CADM1* transfection (Fig. 4d).

## GJIC in physiological and pathological human adrenals

To quantify GJA1 expression in human adrenal ZG, and explore its anatomical relation to aldosterone-producing cells, we undertook laser capture microdissection for qPCR of *GJA1* and zonal marker genes, and IHC of 15 adrenals, including our index case. Data from a previous microarray of 21 adrenals showed *LGR5* and *GSTA3* to be the most selective markers for ZG and ZF, respectively (Supplementary Fig. 4a,b)[31,48]. There was high expression of *GJA1* in ZG and ZF relative to the expression of *LGR5* and *GSTA3* in each zone (Supplementary Fig. 4c). qPCR of RNA from a further three adrenals showed ZG expression of *GJA1* to be, on average, a quarter of that in ZF (Extended Data Fig. 3a and Supplementary Fig. 4c).

On IHC with all relevant controls (Extended Data Fig. 3b and Supplementary Fig. 5a,b), GJA1 protein expression was variable in quantity and distribution. At most ZG regions, staining was punctate on or inside the plasma membrane, differing from the diffuse linear appearance in ZF (Extended Data Fig. 4 and Supplementary Fig. 5c). Staining with two different antisera was comparable, and absent when the primary antibody was omitted or competed by a specific peptide (Extended Data Fig. 3b and Supplementary Fig. 5b,c). A similar pattern of staining for GJA1 was seen on immunofluorescence. Because CYP11B2 itself is rarely expressed in adult human adrenal outside APMs, we used positive staining for VSNL1 and DAB2, and negative staining for CYP17A1, as markers of ZG (Supplementary Fig. 5d and Extended Data Fig. 5)[7,13]. On high-resolution microscopy, AGJs were visualized,

and GJA1 punctate staining in ZG was seen on cell membranes (Fig. 5a, Extended Data Fig. 5 and Supplementary Fig. 5e). The presence of AGJ confirms the dynamic formation and internalization of GJs[50,53]. GJA1 intensity was also lower in APMs than in adjacent ZG (Extended Data Figs. 6 and 7 and Supplementary Fig. 6a–c). On semiquantification of GJA1 IHC, scored 0–3 by two independent pathologists in 14 adrenals from two cohorts, there was a rank order of GJA1 intensity as follows: ZF > ZI (zona intermedia) > ZG > APM (Supplementary Fig. 6d,e). GJA1 expression in APAs varied between tumors and between cell types, being sparse and punctate in CYP11B2-dense cells, and linear in foamy ZF-like cells (Supplementary Fig. 7a). A stabilizing partner of GJA1 is the tight-junction protein, TJP1, through its PDZ (postsynaptic density protein, Drosophila disc large tumor suppressor and zonula occludens-1 protein) domain[54]. We observed highly selective TJP1 expression in ZG (Supplementary Fig. 7b). By contrast, its expression was substantially reduced in APMs (Extended Data Fig. 7b and Supplementary Fig. 7c).

## Modulation of GJIC regulates aldosterone production

Having demonstrated the presence of GJs in the ZG, we investigated the role of GJIC in aldosterone production. The connexin mimic peptide Gap27 targets the SRPTEKTIFFI sequence in GJA1 (ref. 55), and probably GJC1, which differs in protein sequence by just one amino acid residue. Treatment of angiotensin II (Ang II)-stimulated H295R cells with Gap27 increased *CYP11B2* mRNA expression and aldosterone secretion by many fold, in a dose-dependent manner, the maximum response exceeding the 15- to 30-fold effect of transducing mutant *CADM1* (Fig. 5b and Supplementary Fig. 8a). Such increases in *CYP11B2* expression are rarely seen with pharmacological interventions but are comparable to differences between APAs and adjacent adrenal, for example, 78-fold for our index case. Silencing of *GJA1* was less effective (Supplementary Fig. 8b–d). However, RNA sequencing (RNA-seq) data showed that, unlike in primary human adrenal tissues, H295R cells express twofold to threefold more *GJC1* than *GJA1* (Supplementary Table 5b). Indeed, silencing of both GJs increased *CYP11B2* expression and aldosterone secretion (Fig. 5c and Supplementary Fig. 8e,f), mirroring the effects of the *CADM1* variants.

To discover whether GJs would also regulate aldosterone production in ZG cells from primary human adrenal cultures, we repeated the Gap27 experiments in cells collected from the adrenal cortex adjacent to four APAs (Adr 1–4) and one cortisol-producing adenoma (Adr 5). In Ang II-stimulated primary cells adjacent to APAs, Gap27 increased *CYP11B2* expression by ~12-fold and aldosterone secretion by ~4-fold, albeit with large variability between different adrenals (Fig. 5d). Less of an effect was seen on unstimulated cells (Supplementary Fig. 8g). No increase in Ang II-stimulated *CYP11B2* or aldosterone production was seen in cells adjacent to the cortisol-producing adenoma (Adr 5; Supplementary Fig. 8h).

The effect of Gap27 on spontaneous and Ang II-triggered calcium oscillations was interrogated in H295R cells, using Fluo-4AM (Supplementary Note and Supplementary Fig. 9).

## RNA-seq of mutant *CADM1* cells reveals clock genes enrichment

Some of the common somatic mutations, for example, *KCNJ5* and *ATP1A1*, combine loss of their physiological activity with a gain of function[5,8,9,56]. To explore further pathways by which CADM1 might influence aldosterone production, and whether these are physiological actions of native CADM1 or purely pathological, we performed RNA-seq of H295R cells transduced with vector alone, WT *CADM1* and mutant *CADM1* (both isoforms and both variants), and H295R cells transduced with a *CADM1*-specific or nontargeting shRNA. Additionally, RNA-seq was also performed on the index APA with p.Val380Asp variant and two other ZG-like APAs with somatic mutations of *ATP2B3* or *CACNA1D*.

In the main experiment, *CYP11B2* was the most upregulated gene (Fig. 6a,b and Supplementary Table 6a), averaging 14-fold across

isoforms and variants, followed by a neuroendocrine gene not usually associated with adrenal cortex, secretogranin-2 (*SCG2*). Several other genes, upregulated twofold to tenfold, have strong adrenal associations: *DPP4*, *MC2R* and *TSPAN12* are selectively expressed in APMs compared to ZG[49]; *MC2R* and *MRAP* are required for ACTH stimulation of cortisol; the rap guanine nucleotide exchange factor *RAPGEF4* is exclusive to adrenal cortex and brain; *SCG2* is unique to adrenal medulla, brain/pituitary; and the nonmuscle myosin *MYOM1* distinguishes ZG- from ZF-like APAs[8]. In every case, as for *CYP11B2* itself, the p.Val380Asp variant and longer (453 amino acids) isoform were more effective than p.Gly379Asp and 442-amino acid isoform.

The shRNA experiment confirmed the reduction in *CYP11B2* seen previously on qPCR (Supplementary Fig. 10, Supplementary Table 6b and Supplementary Note).

Gene-annotation enrichment analysis and functional annotation clustering, using the Database for Annotation, Visualization and Integrated Discovery (DAVID), revealed biological rhythms as the most significant process, together with cell junction and synapse (Supplementary Table 7). A plot of expression change for each of the 14 biological rhythms genes showed that RORE and E-box genes formed two clusters with, respectively, up- and downregulated transcription (Fig. 6c). In most instances, small changes in cells transduced by WT *CADM1* were in the opposite direction to mutant. Six of the 14 genes have significant diurnal patterns in mouse adrenal, with the peak for RORE genes out of phase with E-box genes (Supplementary Fig. 11). The steroidogenic gene *STAR* was increased by 25% ($P = 1.0 \times 10^{-8}$)[57]. One of the RORE-target genes, *NPAS2*, can replace *CLOCK* itself in dimers with *BMAL1* (ref. [58]), and is also upregulated in the index APA compared to other APAs with different genotypes (Supplementary Table 8). DAVID identified two significant Kyoto Encyclopaedia of Genes and Genomes (KEGG) pathways, 'aldosterone synthesis and secretion' and 'axon guidance', with 5.8-fold and 4.5-fold enrichment, respectively (Supplementary Table 9).

The most striking finding among genes upregulated in both sets of RNA-seq (variant versus WT transduced H295R cells, and the comparison of APAs) is the 19-fold upregulation of *AQP2* (Supplementary Table 8). On IHC of the index APA, AQP2 appears in dense inclusion bodies (Extended Data Fig. 8). These are thought to be spironolactone bodies, assumed to be a form of aggresome originating in endoplasm reticulum (ER). Similar bodies are seen when AQP2 accumulates in ER as a consequence of the mutations that cause diabetes insipidus by preventing glycosylation and transport to the plasma membrane[59]. In two APAs from the replication cohort, there was patchy cytoplasmic staining for AQP2 (Extended Data Fig. 8). Supporting evidence that the rise in AQP2 may be relevant to aldosterone regulation came from its measurement in the RNA samples from Gap27-treated human adrenocortical cells and H295R cells transduced with *CADM1* variants. We found that, in Ang II-stimulated primary adrenal cells treated with Gap27 and H295R cells transduced with mutant *CADM1* (which increased *CYP11B2* expression the most), *AQP2* expression was also substantially increased (Extended Data Fig. 9).

## Discussion

The discovery of pathogenic *KCNJ5* variants in 30–40% of APAs, followed by several others, explained the onset of autonomous aldosterone production in most of these tumors, and highlighted smaller APAs as a distinct, easily overlooked, sub-tier in which size is inversely proportional to the density of aldosterone synthase[8,11,12]. These discoveries confirmed that aldosterone secretion is exquisitely sensitive to changes in membrane potential, but may have illuminated pathology more than physiology. Conversely, the six patients with p.Gly379Arg or p.Val380Arg *CADM1* variants present an exceptional cause of PA. Following the discovery that primate and rodent adrenals have different origins, even rare human mutations are clues to important adrenal biology that could not be anticipated from rodent

experiments—in this case, that GJs suppress aldosterone production in most of human ZG[25,60].

CADM1, a member of the immunoglobulin superfamily, brings cells of the same or different type into contact and is best known as a CNS synaptogenic protein. Its importance was previously established in another endocrine tissue, pancreatic islets, where CADM1 enables β-cells to synapse with each other and with endothelial cells[21]. The neuronal features of ZG have been less remarked, but many of the molecules upregulated in APMs and ZG-like APAs are enriched in CNS, particularly cerebellum[14,49,61]. Herein, on transduction of *CADM1* variants into adrenocortical cells, the gene with tenfold increase in expression, second only to *CYP11B2* itself, was secretogranin-2 (*SCG2*, associated with neuroendocrine granules). In these experiments, the only KEGG pathways with significant enrichment were aldosterone synthesis and axon guidance, and the most differentially expressed biological processes were synapses, cell junctions and biological rhythms.

Of these three processes, our functional experiments concentrated on the specific triangle of *CADM1* variant, inhibition of GJIC and *CYP11B2* stimulation. A seemingly unique feature of human ZG is its sharp demarcation between foci of dense CYP11B2 expression, so-called APMs and the majority of cells where CYP11B2 is paradoxically switched off. Could GJIC be the cause? The role of GJIC in mutant CADM1 effects on aldosterone production was suggested by two observations. One was our previously reported inhibition by GJIC of glucagon secretion from islet α cells, and release of this inhibition by *CADM1* knockdown[18]. The other observation was the change in protein orientation in another instance of spontaneous mutations creating a charged residue within the membrane domain of a single-pass membrane protein[62]. On the basis of fluorescence and Fourier transform infrared spectroscopy, G380R and A391E variants of FGFR3 straightened the TM helix tilt angle relative to the membrane. Structural modeling indicated a similar change, from acute to perpendicular orientation, for each of the *CADM1* variants. We inferred that adjacent cells might be pushed beyond the reach of proteins that can only connect over a short distance. Like the grappling irons of historical naval warfare, GJs between two cells require each cell's connexons to be in close proximity, estimated at <30 nm[50,63].

GJs in ZF facilitate steroid hormone secretion, opposite to our observations in ZG. This again resembles pancreatic islet cells, in which slightly inconsistent literature suggests an opposite role for GJIC between α and β cells[18,64,65]. In our experiments on isolated cells, competitive blockade of the most abundant adrenocortical GJ protein increased *CYP11B2* expression and aldosterone secretion by at least as much as *CADM1* mutation. This is consistent with a tonic role for GJIC in the suppression of *CYP11B2* expression in human ZG, outside the APMs, but further experiments will be required on fresh slices, retaining the physiological architecture (Supplementary Note).

Of the processes modified by these *CADM1* variants, biological rhythms were not only the most significant but showed reciprocal change in clock genes whose endogenous changes in expression peak, respectively, at day or night. Several somatic mutations are associated with specific clinical phenotypes[14,17]. Our two initial patients had striking variability of plasma aldosterone at presentation. They are clearly too few to establish an association, but the interesting question is whether the phenotype explains the apparent rarity. A recent report that 24-h urine measurements find a much higher prevalence of PA than single-time measurements in blood suggested that patients missed by the latter have exaggerated or reversed diurnal rhythms[66].

The individual cell clocks in tissue are coordinated by both extrinsic signals and cell–cell communication, including GJA1 (refs. [67,68]). Whether CADM1's regulation of clock genes, GJIC and synapses points to a role in coordination needs further investigation, together with the contribution of one further molecule, AQP2. This was the most upregulated gene in the RNA-seq of the index APA, and almost the only gene upregulated both in this APA and the variant-transduced H295R

cells. AQP2 was densely present in the spironolactone-like inclusion bodies of the index APA (Supplementary Note).

In summary, we report a somatic mutation hotspot within the membrane dimerization domain of CADM1. The mutations, three each of p.Gly379Arg and p.Val380Arg, caused a many-fold increase in aldosterone production, and a form of hypertension that was cured by unilateral adrenalectomy. The mutations inhibited GJ communication between aldosterone-producing cells, and pharmacological inhibition of GJs replicated the effect of *CADM1* variant on aldosterone production. GJs may underlie the suppression of aldosterone production in most human adrenal ZG.

## Online content

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

# Article

Xilin Wu[1,2,3], Elena A. B. Azizan [1,4,33] ✉, Emily Goodchild[1,2,3,33], Sumedha Garg [1,5,33], Man Hagiyama [6,33], Claudia P. Cabrera [2,7,33], Fabio L. Fernandes-Rosa [8,33], Sheerazed Boulkroun [8,33], Jyn Ling Kuan[9], Zenia Tiang[9], Alessia David [10], Masanori Murakami [11], Charles A. Mein[12], Eva Wozniak[12], Wanfeng Zhao[13], Alison Marker[13], Folma Buss [14], Rebecca S. Saleeb[15], Jackie Salsbury[1,2,3], Yuta Tezuka [16], Fumitoshi Satoh[16,17], Kenji Oki [18], Aaron M. Udager[19], Debbie L. Cohen[20], Heather Wachtel[21], Peter J. King[22], William M. Drake[2,3], Mark Gurnell [23], Jiri Ceral [24], Ales Ryska[25], Muaatamarulain Mustangin[26], Yin Ping Wong[26], Geok Chin Tan[26], Miroslav Solar[24], Martin Reincke [11], William E. Rainey[27], Roger S. Foo [9], Yutaka Takaoka [28], Sandra A. Murray[29,34], Maria-Christina Zennaro [8,30,34], Felix Beuschlein [31,32,34], Akihiko Ito [6,34] & Morris J. Brown [1,2,34] ✉

[1]Endocrine Hypertension, Department of Clinical Pharmacology and Precision Medicine, William Harvey Research Institute, Queen Mary University of London, London, UK. [2]NIHR Barts Biomedical Research Centre, Barts and The London School of Medicine and Dentistry, Queen Mary University of London, London, UK. [3]St Bartholomew's Hospital, Barts Health NHS Trust, London, UK. [4]Department of Medicine, Faculty of Medicine, Universiti Kebangsaan Malaysia, Kuala Lumpur, Malaysia. [5]Clinical Pharmacology Unit, University of Cambridge, Cambridge, UK. [6]Department of Pathology, Faculty of Medicine, Kindai University, Osakasayama, Japan. [7]Centre for Translational Bioinformatics, William Harvey Research Institute, Queen Mary University of London, London, UK. [8]Université Paris Cité, PARCC, Inserm, Paris, France. [9]Cardiovascular Disease Translational Research Programme, Department of Medicine, National University of Singapore, Singapore, Singapore. [10]Centre for Bioinformatics, Department of Life Sciences, Imperial College London, London, UK. [11]Medizinische Klinik und Poliklinik IV, Klinikum der Universität, Ludwig-Maximilians-Universität München, Munich, Germany. [12]Barts and London Genome Centre, School of Medicine and Dentistry, Blizard Institute, London, UK. [13]Department of Histopathology, Addenbrooke's Hospital, Cambridge, UK. [14]Cambridge Institute for Medical Research, The Keith Peters Building, University of Cambridge, Cambridge, UK. [15]Centre for Microvascular Research, William Harvey Research Institute, Queen Mary University of London, London, UK. [16]Division of Nephrology, Endocrinology, and Vascular Medicine, Tohoku University Hospital, Sendai, Japan. [17]Division of Clinical Hypertension, Endocrinology and Metabolism, Tohoku University Graduate School of Medicine, Sendai, Japan. [18]Department of Molecular and Internal Medicine, Graduate School of Biomedical and Health Sciences, Hiroshima University, Hiroshima, Japan. [19]Department of Pathology, University of Michigan Medical School, Ann Arbor, MI, USA. [20]Renal Division, Department of Medicine, Perelman School of Medicine at the University of Pennsylvania, Philadelphia, PA, USA. [21]Department of Surgery, Hospital of the University of Pennsylvania, Philadelphia, PA, USA. [22]Department of Endocrinology, William Harvey Research Institute, Queen Mary University of London, London, UK. [23]Metabolic Research Laboratories, Welcome Trust-MRC Institute of Metabolic Science, and NIHR Cambridge Biomedical Research Centre, Cambridge Biomedical Campus, Cambridge, UK. [24]1st Department of Internal Medicine–Cardioangiology, Charles University Faculty of Medicine in Hradec Kralove and University Hospital Hradec Kralove, Hradec Kralove, Czech Republic. [25]Department of Pathology, Charles University Faculty of Medicine in Hradec Kralove and University Hospital Hradec Kralove, Hradec Kralove, Czech Republic. [26]Department of Pathology, Faculty of Medicine, Universiti Kebangsaan Malaysia, Kuala Lumpur, Malaysia. [27]Division of Metabolism, Endocrinology, and Diabetes, University of Michigan, Ann Arbor, MI, USA. [28]Department of Computational Drug Design and Mathematical Medicine, Graduate School of Medicine and Pharmaceutical Sciences, University of Toyama, Toyoma, Japan. [29]Department of Cell Biology, University of Pittsburgh School of Medicine, Pittsburgh, PA, USA. [30]Assistance Publique-Hôpitaux de Paris, Hôpital Européen Georges Pompidou, Service de Génétique, Paris, France. [31]Klinik für Endokrinologie, Diabetologie und Klinische Ernährung, UniversitätsSpital Zürich (USZ) und Universität Zürich (UZH), Zurich, Switzerland. [32]Present address: Medizinische Klinik und Poliklinik IV, Klinikum der Universität, Ludwig-Maximilians-Universität München, Munich, Germany. [33]These authors contributed equally: Elena A. B. Azizan, Emily Goodchild, Sumedha Garg, Man Hagiyama, Claudia P. Cabrera, Fabio L. Fernandes-Rosa, Sheerazed Boulkroun. [34]These authors jointly supervised this work: Sandra A. Murray, Maria-Christina Zennaro, Felix Beuschlein, Akihiko Ito, Morris J. Brown. ✉e-mail: elena.azizan@ukm.edu.my; morris.brown@qmul.ac.uk

## Methods

### Genetic analysis of patient cohort

All patients were confirmed to have a diagnosis of PA on the basis of a high aldosterone to renin ratio, ± hypokalemia, followed by confirmatory testing, when appropriate. Subtyping was performed by cross-sectional imaging (CT/MRI) and either adrenal vein sampling or [11]C-metomidate PET-CT, as per local institutional and Endocrine Society guidelines[1,69]. All patients gave written informed consent for the study according to local ethics committee guidelines.

The index case was recruited from Addenbrooke's Hospital, University of Cambridge (Cambridgeshire Research Ethics Committee), as part of 40 APAs to be sequentially whole exome sequenced. The second case was recruited from the University Hospital Munich—one of 81 APAs whole exome sequenced in the APA working group for the European Network for the Study of Adrenal Tumors, ENS@T (local Ethics Review Board, University Hospital Munich). Cases 3–5 were from Paris, recruited between 1999 and 2016 within the COMETE (COrtico- et Medullo-surrénale, les Tumeurs Endocrines) network (CPP Ile de France II Ethics Committee) identified by targeted sequencing. The final case was recruited from the Perelman School of Medicine at the University of Pennsylvania (with ethical approval from the Institutional review board, University of Pennsylvania), identified by next-generation sequencing.

In addition to the above, targeted Sanger sequencing was performed on a further 53 APAs from four centers as follows: Addenbrooke's Hospital, University of Cambridge; St Bartholomew's Hospital, Queen Mary University of London; Tohoku University Hospital, Sendai; and Graduate School of Biomedical and Health Sciences, Hiroshima University, Hiroshima.

### Nucleic acid extraction

Genomic DNA (gDNA) was extracted from APA and adjacent adrenal tissue using QIAamp DNA Mini kit (Qiagen; UK cohort) or QIAamp DNA midi kit (Qiagen; French cohort). If the adjacent adrenal tissue next to an APA was not available for use as control, gDNA was extracted from blood using the salt extraction method. Total RNA was extracted using the Trizol method (Life Technologies) and either Invitrogen PureLink RNA mini kit or RNeasy mini kit (Qiagen). DNase I (Invitrogen) treatment was performed for all samples. If RNA later-preserved tissue was not available, total RNA was extracted from FFPE tissue blocks, cut from ten 4-µm FFPE sections using the RNeasy FFPE kit (Qiagen). RNA from H295R or primary adrenal cells was extracted using the PureLink RNA mini kit (without Trizol).

### WES of Cambridge and German cohorts

WES of 40 pairs of APAs and adjacent adrenal from the Cambridge cohort and the 81 APAs from the German cohort are as previously published[9,17].

### Sanger sequencing of CADM1

Sanger sequencing was performed to confirm the somatic mutations of CADM1 identified by WES and to seek out further CADM1 variants in a replication cohort that was separate from those whole exome sequenced. PCR was performed with the primers listed in Supplementary Table 10, using AmpliTaq Gold Fast PCR Master Mix (Thermo Fisher Scientific), as per the manufacturer's instructions; 100 ng of DNA in a final volume of 25 µl containing 400 nM of each primer, 200 µM deoxynucleotide triphosphate and 1.25 U Taq DNA Polymerase (Sigma-Aldrich). Sequencing was performed using BigDye Terminator v3.1 Cycle Sequencing Kit (Applied Biosystems; for French and German cohort); or commercially using Eurofin Sanger sequencing services (for UK cohort). GATC Viewer v.1.00 or UniPro UGENE v.1.28.1 were used for sequencing alignment.

### Targeted next-generation sequencing (US cohort)

DNA was extracted from CYP11B2-positive tumor and adjacent normal adrenal cortex from formalin-fixed paraffin-embedded (FFPE) tissue

sections using the AllPrep DNA/RNA FFPE Kit (Qiagen). Next-generation sequencing (NGS) libraries were constructed from FFPE-extracted DNA using a custom Ion AmpliSeq panel (Thermo Fisher Scientific) that targets the full-coding region of CADM1—as well as other known aldosterone-driver genes (ATP1A1, ATP2B3, CACNA1D, CACNA1H, CLCN2, CTNNB1, KCNJ5 and GNAS)—and sequenced using the Ion Torrent NGS System (Thermo Fisher Scientific)[70,71].

### Immunohistochemistry

IHC of control adrenals was performed using the EnVision FLEX+, Mouse, High pH kit (Dako, K8012). Fourteen adrenals with a unilateral APA undergoing follow-up treatment for PA at the University Hospital Hradec Kralove (n = 11) and the National University of Malaysia (UKM) Medical Center (n = 3) were used. As a non-APA adrenal control, one adrenal (control adrenal 1) from a 48-year-old Malay male with a large epigastric mass undergoing follow-up treatment for a pancreatic neuroendocrine tumor was also used. Usage of adrenals for investigations was approved by the local research ethics committee (the University Hospital Hradec Kralove Ethics Committee and UKM research ethics committee). Case detection and PA subtype identification were in accordance with local guidelines[72–74]. Incubation with primary antibodies (listed in Supplementary Table 11) was performed at room temperature for 30 min. Slides were also counterstained with Hematoxylin 2 (Thermo Fisher Scientific). The slides were mounted using DPX mounting medium (Merck Milipore) and images were captured using 3DHISTECH Pannaromic MIDI scanner (Software version 1.18).

For IHC scoring, a minimum of three ×20 images in selected areas of the scans that showed recognizable zonation of the adrenal cortex were captured using CaseViewer (Software version 2.4). All scoring of IHC scans were performed by two histopathologists independently after standardization using images captured from eight adrenals with 0 representing no staining, +1 representing weak staining, +2 representing intermediate staining and +3 representing intense staining (representative images of scores are shown in Supplementary Fig. 6e). An adrenal cortex region is considered APM, ZG, Z, or ZF based on IHC staining scores of CYP11B2, CYP17A1 and KCNJ5 as shown in Supplementary Table 12.

IHC for mutant CADM1 adrenals was performed as control adrenals (for 184T) or as per described below (for French cohort). In brief, IHC against CADM1 and AQP2 used citrate solution (0.935%, Vector Laboratories; 30 min at 98 °C) for antigen unmasking while IHC against CYP11B2 and CYP11B1 used Trilogy solution (5%, Sigma-Aldrich; 30 min at 98 °C). Endogenous peroxidases were inhibited by incubation in 3% hydrogen peroxide (Sigma-Aldrich) in water for 10 min. Nonspecific staining was blocked with 10% normal goat serum and 1× PBS (CADM1) or 1× TBS (AQP2) for 30 min; or Tris 0.1 M pH 7.4, 10% normal goat serum, 10% BSA and 0.1% SDS for 90 min (CYP11B2); or Tris 0.1 M pH 7.4, 10% horse serum and 0.5% SDS for 60 min (CYP11B1). The slides were incubated with primary antibody (listed in Supplementary Table 11) for 1 h at room temperature (CADM1 and AQP2) or overnight at 4 °C (CYP11B2 and CYP11B1). Sections were washed, incubated 30 min with affinity-purified goat anti-rabbit antibody (1/400, Vector Laboratories), and then washed and incubated with an avidin-biotin-peroxidase complex (Vectastain ABC Elite; Vector Laboratories) for 30 min. Slides were developed using diaminobenzidine (Vector Laboratories) and counterstained with hematoxylin (Sigma-Aldrich). Images were acquired using a Vectra automated imaging system software v.3.0.5 (PerkinElmer).

### Immunofluorescence staining

For immunofluorescence staining, antigen retrieval was done by water bath incubation with pH 6.0 citrate buffer (Sigma-Aldrich) at 95 ºC for 35 min. Samples were quenched by incubation with 0.1% Sudan black B, diluted in 70% ethanol for 25 min. Primary antibodies were incubated at 4 ºC overnight whereas Alexa Fluor (AF) conjugated secondary antibodies diluted in blocking serum were incubated at

room temperature for 1 h (antibodies listed in Supplementary Table 13). Wheat germ agglutinin (WGA) staining was performed with 5 μl ml$^{-1}$ AF 647 conjugate (Invitrogen, W32466) before incubation with protein block serum-free (Dako). All sections were incubated with 300 nM DAPI stain, mounted with Vectashield Antifade mounting medium and cured for 24 h. Visualization of fluorophores was by TissueFAXS SL Q + upright epifluorescence microscope and viewed with TissueFAXS slide viewer 7.0 software. FIJI ImageJ software v1.8.0_72 was used to prepare the images for presentation.

## Cell culture and drug treatments

Functional studies were performed using NCI-H295R cells (human adrenocortical cell line) or primary human adrenal cells. Both H295R and primary adrenal cells were grown in culture medium consisting of DMEM/Nutrient F12-Ham supplemented with 10% FBS, 100 U penicillin, 0.1 mg ml$^{-1}$ of streptomycin, 0.4 mM L-glutamine and 1% insulin-transferrin-sodium selenite. Primary human adrenal cells were collected from adrenalectomy in culture medium and dispersed within 1–2 h of retrieval. 'Normal' adrenal tissue next to an APA was digested in 3.33 mg ml$^{-1}$ collagenase type XI-S from histolyticum (Sigma-Aldrich, C9697) for 2 h, washed twice with PBS, and resuspended in culture medium. After 5–7 d, the primary adrenal cells were used for drug treatments, seeded between 1.5 and 1.75 × 10$^5$ cells per well and serum-starved in unsupplemented medium for 6 h before treatment. Cells were treated with Ang II at a concentration of 10 nM and Peptide Gap27 at 10–250 μM, for 24 h.

NIH-3T3 fibroblast cells used in Western blot experiments (due to their negligible endogenous expression of CADM1) were grown in DMEM culture medium supplemented with 10% FBS. HEK293T cells used to produce lentiviruses for transduction of CADM1 were cultured in high glucose DMEM supplemented with 10% FBS, 100 U penicillin, 0.1 mg ml$^{-1}$ of streptomycin and 0.4 mM L-glutamine. All cells were maintained at 37 °C in 5% CO$_2$.

## Generation of pLOC plasmids

cDNA from H295R cells was used as a template to amplify the *CADM1* gene, using Q5 DNA polymerase. Topo-FLAG- and MYC-tag sequences were incorporated up- and downstream of *CADM1*, respectively, using a TF-*CADM1*-F 5′ primer and M-*CADM1*-R 3′ primers (listed in Supplementary Table 14). The amplified PCR product was cloned into a pENTR vector using directional TOPO cloning technology (pENTR/D-TOPO Cloning Kit, Invitrogen). The Q5 site-directed mutagenesis kit (New England Biolabs) was used to insert the extra 33-bp sequence of the SP1 (453-amino acid isoforms), using Ins-Ex9A-F and Ins-Ex9A-R primers. The G379D and V380D variants were introduced via site-directed mutagenesis, using the primers G379D-F, G379D-R and V380D-F, V380D-R, respectively. The pENTR constructs were used for transfer of cloned cDNAs into pLOC vectors (kindly gifted by Celso Gomez-Sanchez, University of Mississippi) in an *att*L/*att*R recombination reaction using LR Clonase II enzyme mix (Invitrogen).

## Generation of shRNA

The primers 5′-tccaattgtagaggataagtcatctgtTTTTTGGAAAAGCTTATCGATAC-3′ and 3′-tccaattgtagaggataagtcatctgccGGGGATCTGTGGTCTCAT-5′ were used to clone shRNA targeting *CADM1* into pLVTH vectors using Q5 site-directed mutagenesis kit (NEB). The pLVTH was kindly gifted from Didier Trono (Addgene plasmid, 12262). The stem-loop sense (5′-GTCTACTGAATAGGAGATGTT-3′) and anti-sense (5′-AACATCTCCTATTCAGTAGAC-3′) sequences consisted of 21 bp each. For scrambled (nontargeting control) shRNA, the primers 5′-tccaacgaggttattacgtaaggtattTTTTTGGAAAAGCTTATCGATAC-3′ and 3′-tccaacgaggttattacgtaaggtatccGGGGATCTGTGGTCTCAT-5′ were used for cloning. The sense and anti-sense shRNA sequences used were 5′-ATACCTTACGTAATAACCTCG-3′ and 5′-CGAGGTTATTACGTAAGGTAT-3′, respectively.

## Generation of pCX4bsr plasmids

The full-length cDNA for human *CADM1* was obtained from human lung mRNA (BioChain) by reverse transcription and polymerase chain reaction with a primer set of 5′-agtctgaggcaggtgcccgacat-3′ and 5′-cagttggacacctcattggaac-3′. The double-stranded cDNA was subcloned into the Bluescript vector via the EcoRV site by TA cloning, and the formed vector (pBs-453-*CADM1*) was determined by sequencing to have no mutations. The EcoRI-XhoI fragment containing the cDNA insert was subsequently subcloned into the pCX4bsr vector via the EcoRI and NotI site after the XhoI and NotI sites were blunt-ended with the Klenow large fragment. An XhoI site was generated in the cDNA insert of pBs-453-*CADM1* via site-directed mutagenesis using the oligonucleotides 5′-tctcgagca-3′. This allowed the generation of the 442-amino acid isoform of CADM1 using XhoI and AccI. PCR was performed using these two pBluescript vectors as DNA templates, with the primers shown in Supplementary Table 15, which also allowed the introduction of the two mutations, as appropriate, via site-directed mutagenesis. The PCR products were subcloned into pTA2 vectors (Toyobo) by TA cloning, and sequenced to confirm the successful generation of WT, G379D or V380D mutant CADM1 in both 442- and 453-amino acid isoforms. Finally, the DNA inserts were cut out from the vectors by EcoRI digestion, and subcloned into pCX4bsr vector via the EcoRI site.

## Gene delivery using transfection

Overexpression of pCX4bsr and pLOC plasmids was achieved using Lipofectamine 3000 (Invitrogen). Cells were plated at 70% confluency and transfected after 24 h, using 1 μg of plasmid DNA and Opti-MEM (Gibco, Thermo Fisher Scientific). Cotransfection of H295R cells with *pLOC* and *GJA1* vectors, or *GJA1* and *GJC1* targeting siRNA, was carried out by electroporation using Invitrogen's Neon Transfection System. One microgram of DNA or 20 nM of siRNA was used for the 10 μl system and electroporated at the following conditions: 1,100 V, 40 ms, one single pulse.

## Gene delivery using lentivirus

Lentivirus particles were generated using HEK293T cells transfected with pMD2, psPAX (both kindly gifted by Didier Trono; Addgene plasmid, 12259 and 12260), and pLOC vectors using polyethylenimine. Virus particles were concentrated using Amicon-15 filter units (Merck). Titration of the lentivirus vectors was achieved via flow cytometric (FACS) method (BD FACSDiva Software v6.1.3, BD Biosciences, and Flowing Software v.2.5.1) utilizing the GFP on the pLOC vectors[75]. Transduction was performed with 8 μg ml$^{-1}$ of Polybrene using a multiplicity of infection (MOI) of 3–5 TU per cell.

## Aldosterone concentration measurements

Aldosterone concentrations were measured using a Homogeneous Time Resolved Fluorescence assay by Cisbio. The fluorescent signals were read at 665 nm and 625 nm using a FLUOstar Omega plate reader software v5.7 (BMG Labtech). Due to the potential interaction of FBS in the medium with the aldosterone assay, all treatments were performed using serum-free medium.

## Quantification of protein expression using western blotting

Cells transfected with pCX4bsr plasmids were lysed and immuno-blotted using a custom-made rabbit polyclonal anti-CADM1 antibody[76] (1:2,000), generated against the C-terminal peptide sequence EGGQNNSEEKKEYFI[77] or an anti-β-actin antibody (1:2,000; Medical and Biological Laboratories).

For experiments involving silencing of *CADM1* using shRNA and silencing of *GJA1* and *GJC1* using siRNA, cells were lysed and immune-blotted using the following primary antibodies; anti-CADM1 C-terminal (Sigma-Aldrich, S4945; 1:5,000), anti-CADM1 N-terminal 3E1 (Medical & Biological Laboratories, CM004-3), anti-β-actin (Medical &

Biological Laboratories, PM053; 1:2,000), anti-GAPDH (Sigma-Aldrich, G8795; 1:40,000), anti-GJA1 (Sigma-Aldrich, C6219; 1:8,000) and anti-GJC1 (Invitrogen, PA5-79311; 1:2,500). Secondary antibodies used include anti-mouse (Sigma-Aldrich, A3682; 1:40,000), anti-chicken (Medical & Biological Laboratories, PM010-7; 1:5,000) and anti-rabbit (Sigma-Aldrich, A0545; 1:40,000) antibodies.

## Quantification of mRNA expression using RT−PCR

One microgram of RNA was reverse transcribed to cDNA using either Applied Biosystem's High Capacity RNA-to-cDNA kit or AMV Reverse Transcriptase System (Promega). qPCR was used to quantify the level of mRNA expression of the genes of interest. This was performed using a C1000 Touch Thermal Cycler machine, CFX Manage TM Software v3.1 (Bio-Rad) or ABI 7900 Real-time PCR system, 7900 SDS v2.4.1 (Thermo Fisher Scientific) and the TaqMan Fast Advanced Gene Expression Master Mix (Applied Biosystems, 4369514). The genes of interest were identified using commercially available probes from Thermo Fisher Scientific, with Assay ID listed in Supplementary Table 16. Results were analyzed using the $2^{-\Delta\Delta CT}$ method, using the housekeeping gene 18S rRNA or β-actin for normalization, unless otherwise specified.

## Dye transfer assays

H295R cells of $2.5 \times 10^5$ were plated and transfected with pLOC vectors described above. Forty-eight hours after transfection, the cells were transferred into the on-stage $CO_2$ incubator of a confocal laser microscope (C2, Nikon), which was also equipped with a micromanipulation system. A single, GFP+ cell was injected with 200 μg ml$^{-1}$ of AF 647-conjugated WGA (Invitrogen) and 20 μg ml$^{-1}$ of calcein red AM (CellTrace, Invitrogen). The micromanipulation system was set as follows: microcapillary angle: +60°, FemtoJet 5247 injection pressure: 200 hpa, injection time 2 s, postinjection pressure: 100 hpa. Fluorescent and differential interference contrast images were captured immediately after injection to confirm dual-color labeling of the GFP+ cells, followed by repeat images taken 1 h after injection. WGA, which is impermeable through GJ, was used to identify the original cell injected, while the GJ permeable calcein red was used to visualize intercellular GJ communication. The number of calcein red positive cells within a 50-μm radius of the WGA labeled, original cell injected was counted and expressed as a percentage of the total number of cells within the 50-μm radius. The same technique was used to measure dye transfer in H295R cells treated with soluble CADM1 immunoglobulin-ectodomain, using DiI (Invitrogen) as the GJ impermeable dye and BCECF (Dojindo) as the GJ permeable dye. In these experiments, H295R cells were treated with either 10 μg ml$^{-1}$ of soluble CADM1 (secreted form of CADM1 fused to the Fc portion of immunoglobulin G (IgG)[41] or 10 μg ml$^{-1}$ of Fc control (secreted form of CADM1 that lacks 3 immunoglobulin-like loops, fused to the Fc portion of IgG) for 24 h before injection of dyes.

## Protein modeling

Three-dimensional structures of the TM domain (residues, A375−L395) of WT, G379D and V380D mutant CADM1 was modeled using QUARK (March 2018)[78]. These structures were embedded in the lipid bilayer, which was composed of 128 dipalmitoylphosphatidylcholine molecules and then soaked in explicit water molecules[79,80]. The SPC water model was used to solvate the protein and counter ions. These molecules were subjected to structural optimization by using GROMACS (2019 version)[81] with the GROMOS 53a6 force field. Energy minimization and molecular dynamics analysis were carried out in accordance with the established methods[82]. The 3D structures of the stalk region of CADM1 for both isoforms (residues, Y329−V376 in SP4 and residues, Y329−V385 in SP1) were also analyzed by using QUARK. The most stable structure of each isoform was joined to the 3D structure of Ig-like domains (residues, K50−T345), which was obtained from ModBase[83] (model ID, 7f7df202e8d6aa361f9ca3847afe4608) by using the PyMOL Moledular Graphics System software (version 2.1, Schrödinger). The secondary structure of each isoform was analyzed by DSSP[84]. The distances between the most separated atoms in the stalk regions were measured by PyMOL. The tilt angle of the TM helix of WT, G379D and V380D with respect to the bilayer was analyzed by hydrophobic mismatch[85]. The length of the shortest TM helix (V380D) was used as thickness of the lipid bilayer. The tilt angle of the TM helix, $\theta$, was given by the following formula:

$$\theta = \sin^{-1}\left(\frac{h_{V380D}}{h}\right)$$

where $h$ is the length of the TM helix of CADM1.

After the confirmation of validity in these CADM1 structures based on the trajectories and 3D quality under the Ramachandran plot, to determine the model structure of *cis*-homodimer for the 442- and 453-amino acid isoforms of CADM1, docking simulation was performed by using ZDOCK v.3.0.2 software[86]. Tetramer models were simulated and analyzed using the same procedure as for dimer modeling. Two thousand docking runs were performed under the definition of Ig-like V-type domain (residues Q45−T139) as a docking site. The docking poses of *trans*-tetramer for isoform 1 and *trans*-tetramer for isoform 4 were classified into six groups, respectively. The complex with the highest ZDOCK score was selected for each binding model. Intercell membrane distance deduced by *trans*-homophilic binding of WT, G379D and V380D were calculated by $d = l \times \sin\theta$, where $l$ is the length of extracellular domain of CADM1 *trans*-tetramer and θ is the angle of the ectodomain to the cell membrane defined by equation 1. Statistical analysis was performed using $R$ software. Differences in intercell membrane distances were evaluated using one-way ANOVA with Tukey's post hoc test. Data represent the mean ± s.d., with $P < 0.05$ regarded as statistically significant.

The protein structure of the TM domain of CADM1 was also separately modeled using Phyre2 v.2 (ref. [87]). TMDOCK[88] was used to predict the insertion and homodimerization of the TM region of the WT and Phyre predicted variant (G379D and V380D) structures in the cell membrane bilayer.

## Visualizing GJ formation in H295R cells

Electroporation was used to cotransfect one pool of H295R cells with the aforementioned pLOC vectors containing WT or mutant *CADM1* and the mApple-tagged *GJA1* vectors. A second pool of H295R cells was transfected with Venus-tagged *GJA1* vectors. After 48 h, the cells were trypsinized, mixed and reseeded into eight-well glass-bottom chamber slides coated with poly-L-lysine (Ibidi). Forty-eight hours post-trypsinization, GJ formation in the cells was visualized using a Spinning Disk Confocal microscope and the NIS-Elements v4.5 software (Nikon).

## Laser capture microscopy (LCM)

Ten micrometer thick slices of fresh frozen 'normal' adrenal adjacent to an APA were mounted onto Zeiss-membrane slides under RNAse-free conditions for LCM. Serial sections were fixed and stained with 1% Cresyl violet (Sigma-Aldrich), allowing visualization of the different zones of the adrenal. ZG and ZF cells were collected by LCM technique using a Zeiss PALM microbeam laser dissection system (Carl Zeiss). Samples were frozen on dry ice immediately post-collection and stored at −80 °C. ZG and ZF samples dissected from eight to ten slices from the same adrenal were pooled for each RNA extraction, using Applied Biosystems' PicoPure RNA Isolation Kit.

## Calcium oscillation experiments

H295R cells were plated in 24-well glass bottom plates (Ibidi) at $1.5 \times 10^5$ cells per well. Cells were treated with 10 nM of Ang II or 250 μM of peptide Gap27 for 24 h, before incubation with 2.5 μM of Fluo-4AM for 45 min. Subsequently, cells were washed with PBS and re-instilled

with the relevant drugs diluted in serum-free, phenol-red-free medium. Imaging was performed using LSM800 microscope (Zeiss) and analyzed using Fiji ImageJ v.1.52p Java 1.8.0_172. Fluo-4AM calcium measurements for individual cells were recorded and expressed as 'mean cell fluorescence' and graphed over time to visualize calcium flux/oscillations within each cell. To allow comparison between drug treatments, all microscope parameters (for example, laser power, collection bands, pixel size, detector gain and offset) were kept constant throughout the experiment. The baseline fluorescence (F0) for each cell was calculated by averaging the lowest fluorescence intensity of ten frames of a 50-frame 'window', repeated at two further distinct time points where the calcium flux was identified to be at baseline. In traces where no oscillations/'events' were identified, the 50-frame windows were set to start at three equally spaced intervals[89–91].

### RNA-seq of H295R cells expressing *CADM1* variants

Extracted RNA from H295R cells transduced with empty pLOC vector (EV), WT *CADM1* and mutant *CADM1* (both isoforms and both variants), treated with or without Ang II, was sequenced by the Barts and the London Genome Center, Blizard Institute in London. Quality control was performed using Agilent RNA 6000 nano reagent kit and run on an Agilent 2100 bioanalyzer (Agilent). Only samples with a RIN number >0.8 were sequenced. Library preparation using the mRNA library prep method was performed using NEBNext Ultra II RNA Library Prep Kit from Illumina. Sequencing was performed on Illumina's NextSeq 500 system, using the high-output 150-cycle kit v2.5 and bcl2fastq Conversion Software. Partek Flow (Partek) was used for RNAseq analysis. Sequences were aligned hg38 with STAR −2.6.1d and annotated genes to Ensembl Transcripts release 93 with Partek's own annotation tool. Partek's GSA tool was used to generate lists of differentially expressed genes. Genes with a fold change >1.5 or <0.7 and ANOVA $P < 0.001$ between WT and variant transduced cells were selected for analysis using DAVID Bioinformatics Resources v.6.8 (refs. [92],[93]).

### Statistics and reproducibility

All parametric data are presented as mean ± s.e.m., or median ± interquartile range, for nonparametric data. Statistical analysis was performed using GraphPad Prism v8.4.3 or R software v.3.5.0. All statistical tests were two-sided where applicable. For parametric data, comparisons between two groups were made using Student's *t*-test, or by one-way analysis of variance (ANOVA) for groups of three or more. For nonparametric data, the Mann–Whitney test was performed when comparing two groups, or the Kruskal–Wallis test for groups of three or more. Post hoc analysis between three or more groups was performed using Dunn's or Sidak's multiple comparisons test. Normality testing was performed using Shapiro–Wilk Test. $P < 0.05$ were considered statistically significant.

Immunostaining in Extended Data Fig. 1–8 and Supplementary Figs. 5a–e, 6a–c, e and 7a–c were only performed once with no primary antibody controls and only after positive tissue controls showed the expected staining. Time-lapse imaging of GJ formation in cocultured H295R cells shown in Fig. 4c was performed in two independent experiments.

### Reporting summary

Further information on research design is available in the Nature Portfolio Reporting Summary linked to this article.

## Data availability

The RNA-seq data used to generate Fig. 6, Supplementary Fig. 10, and Supplementary Tables 5b and 6–9 are provided in Supplementary Data 1–3. The WES raw data are publicly available from the Sequence Read Archive (https://www.ncbi.nlm.nih.gov/sra/docs/) under accession numbers PRJNA732946 and PRJNA729738. Source data are provided with this paper.

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

## Acknowledgements

We thank C. E. Gomez-Sanchez (University of Mississippi Medical Center, Jackson, MS, USA) and D. Trono (Queen Mary University of London, London, UK) for their kind gifts of custom-made CYP11B2 antibody, and pLVTH, pMD2 and psPAX vectors, respectively. We are grateful to J. Hogenesch, L. A. Luong, J. Zhou, G. Argentesi and S. M. O'Toole for their thoughtful discussions. This work was supported by the following UK funding bodies: National Institute for Health Research (NIHR, Efficacy and Mechanisms Evaluation project 14/145/09 to M.J.B., W.M.D., M.G.), the British Heart Foundation (PhD studentship FS/14/75/31134 to M.J.B. supporting S.G.), Barts Charity (Project MGU0360 to M.J.B. and W.M.D.), NIHR Barts Biomedical Research Center (BRC-1215-20022 supporting M.J.B.), NIHR Integrated Academic Training Clinical Lectureship (to W.M.D. and M.J.B., supporting X.W.), NIHR Cambridge Biomedical Research Center (BRC-1215-20014 supporting M.G.), UKRI (MR/S007776/1 to F. Buss) and Wellcome Trust (104955/Z/14/Z to A.D.). E.A.B.A. was funded by the UK-MY Joint Partnership on Noncommunicable Diseases 2019 program (NEWTON-MRC/2020/002) and the Royal Society-Newton Advanced Research Fellowship (NA170257/FF-2018-033). C.P.C. was funded by the NIHR as part of the portfolio of translational research of the NIHR Biomedical Research Center at Barts and The London School of Medicine and Dentistry. E.G. was supported by a Medical Research Council (UK) Fellowship, MR/S006869/1. This research used Queen Mary's Apocrita HPC facility (https://doi.org/10.5281/zenodo.438045), supported by QMUL Research-IT. Immunofluorescence images were acquired in the Blizard Advanced Light Microscopy Suite. The work in Japan was supported by the Japan Society for the Promotion of Science KAKENHI grants (18K07414 to Y. Takaoka, 17K08680 to M.H., 21K08557 to K.O., and 15K15113, 18K07049 to A.I.); the Ministry of Education, Culture, Sports, Science and Technology-Supported Program for the Strategic Research Foundation at Private Universities 2015-19 (to A.I.); grant from Takeda Science Foundation (to M.H.); and grants from the Ministry of Health, Labour, and Welfare, Japan (20FC1020 to F.S.). M. Murakami was supported by the Japan Heart Foundation/Bayer Yakuhin Research Grant Abroad and a postdoctoral fellowship of the Uehara Memorial Foundation. The work in Paris was supported through institutional funding from Inserm, by the Agence Nationale de la Recherche (ANR-18-CE93-0003-01) and the Fondation pour la Recherche Médicale (EQU201903007864 to M.-C.Z.). Other funding bodies include the National Medical Research Council and Biomedical Research Council, Singapore (to R.S.F.); the National Science Foundation grant MC-2011577 (to S.A.M.); the large research infrastructure project BBMRI.cz LM2023033 (to A.R.); Else Kröner-Fresenius-Stiftung in support of the German Conn's Registry–Else Kröner Hyperaldosteronism Registry (2013_A182, 2015_A171 and 2019_A104 to M.R.), the European Research Council under the European Union's Horizon 2020 research and innovation program (grant agreement 694913; to M.R. and grant agreement 633983 to F. Beuschlein), the Deutsche Forschungsgemeinschaft (DFG) within the CRC/Transregio 205/1 'The Adrenal: Central Relay in Health and Disease' to M.R. and F. Beuschlein, and by the Clinical Research Priority Program (CRPP HYRENE) and the University Research Priority Program (URPP ITINERARE) of the University of Zurich (to F. Beuschlein).

## Author contributions

X.W., E.A.B.A., E.G., S.G., M.H., F.L.F.-R., S.B., M.-C.Z., F. Beuschlein, A.I. and M.J.B. conceived and designed the experiments/analyses. S.A.M. provided critical interpretation. C.P.C., J.L.K., Z.T., C.A.M., E.W. and R.S.F. contributed to WES or RNA-seq production, validation and analysis. A.D. and Y. Takaoka generated the protein modeling data. R.S.S helped with calcium oscillation experiments. F. Buss provided the *GJA1* constructs. J.S. undertook clinical assessments of the index case. Y. Tezuka, F.S., K.O., P.J.K., A.M.U. and W.E.R. undertook searches for additional *CADM1* variants. W.M.D., M.G., M.R., D.L.C., H.W., M.-C.Z. and M.J.B. recruited patients in whom *CADM1* variants were found. J.C., A.R. and M.S. collected and contributed adrenal samples for genotyping and/or IHC, which was performed by W.Z., A.M. and M. Murakami. The IHC scoring was performed by Y.P.W. and G.C.T. with imaging by E.A.B.A. M. Mustangin undertook quantification of GJA1 in adrenal IHC. Immunofluorescence staining was performed by E.G. The manuscript was prepared by X.W., E.A.B.A. and M.J.B. All authors read and approved the manuscript.

## Competing interests

The authors declare no competing interests.

## Additional information

**Extended data** is available for this paper at https://doi.org/10.1038/s41588-023-01403-0.

**Correspondence and requests for materials** should be addressed to Elena A. B. Azizan or Morris J. Brown.

**a.**

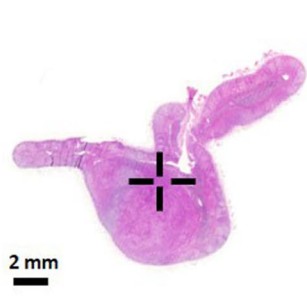
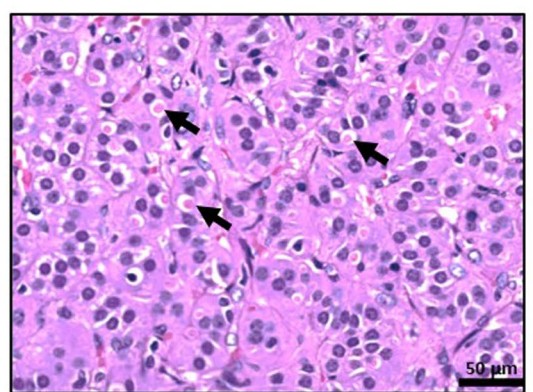

**b.**

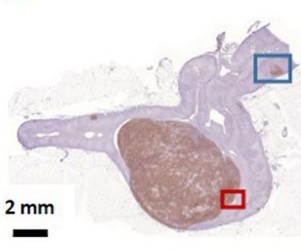
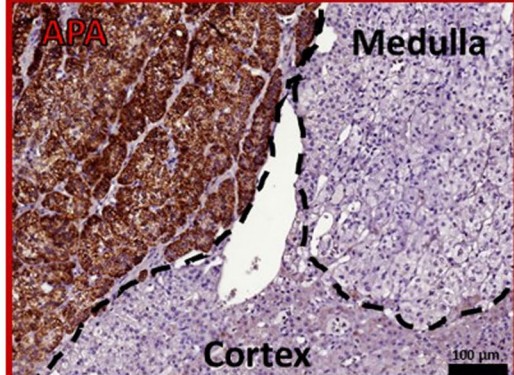
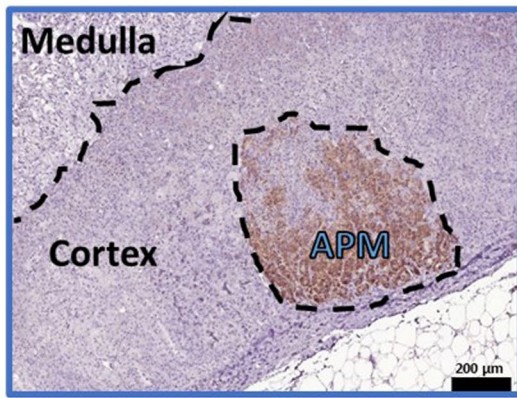

**c.**

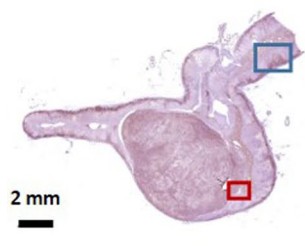
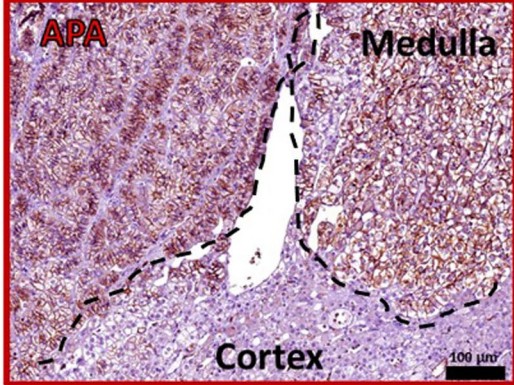
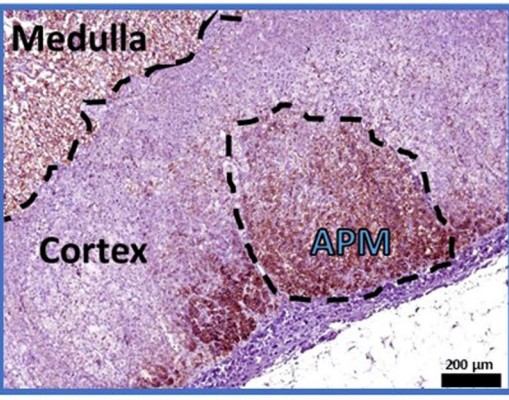

**Extended Data Fig. 1 | Characteristics of *CADM1* V380D mutant APA.**
**a**, Hematoxylin and eosin staining of *CADM1* V380D mutant APA (P1). Low and high power views of hematoxylin and eosin staining in APA of P1 displaying compact cells and numerous spironolactone bodies (black arrows). **b,c**, Immunohistochemistry for CYP11B2 (**b**) and CADM1 (**c**) in *CADM1* V380D mutant APA (P1). CADM1 is highly expressed in the APA (red box) and adjacent aldosterone-producing micronodules (APM, blue box), with membranous staining also seen in the medulla. The magnified area of the red box shows the border between adrenal medulla and APA, whereas the blue box shows CADM1 peri-capsular staining of the outer cortex. Scale bar as indicated on image.

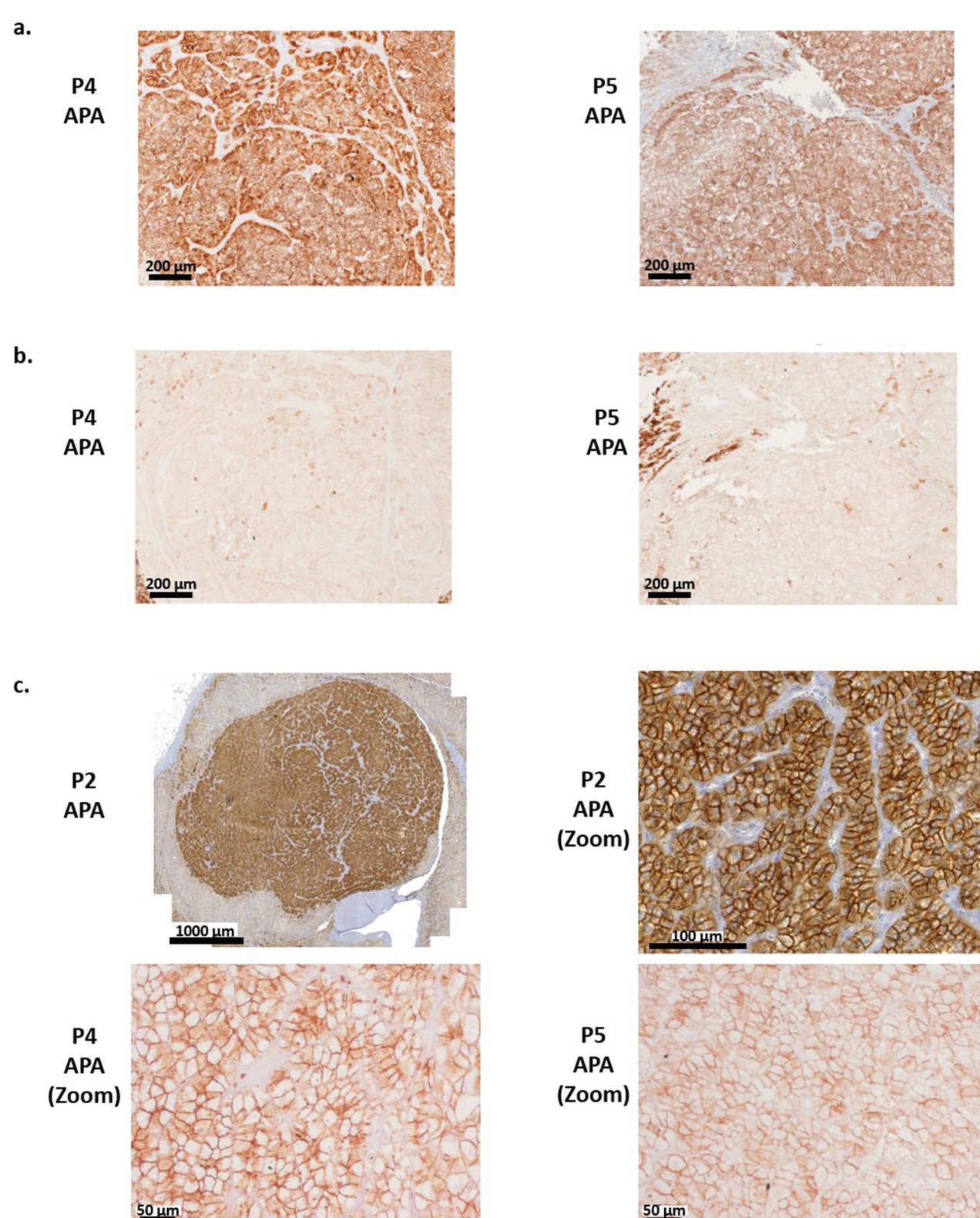

**Extended Data Fig. 2 | Characteristics of *CADM1* G379D mutant APAs.**
**a,b**, Immunohistochemistry for CYP11B2 and CYP11B1 in *CADM1* G379D mutant APAs. Staining of CYP11B2 (**a**) and CYP11B1 (**b**) was inversed (strong CYP11B2 staining in APA with faint CYP11B1). A similar pattern was seen with *CADM1* V380D mutant APAs. **c**, Immunohistochemistry for CADM1 in *CADM1* G379D mutant APAs (P2, P4, and P5). Cross-sectional scans and zoomed images of CADM1 staining of APA from patients P2, P4, and P5. Again as seen with *CADM1* V380D mutant APAs (Extended Data Fig. 1c), the CADM1 protein appears membranous and is most highly expressed in the outer cortex and adenoma. Scale bar as indicated on image.

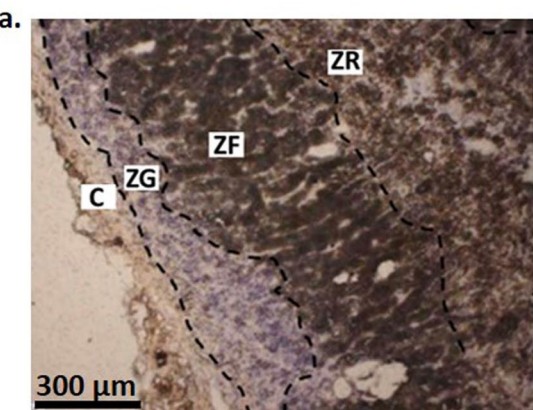

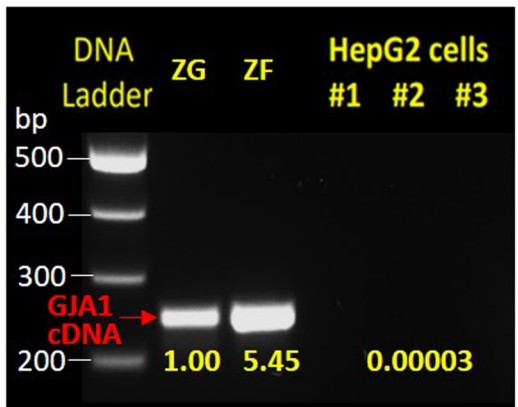

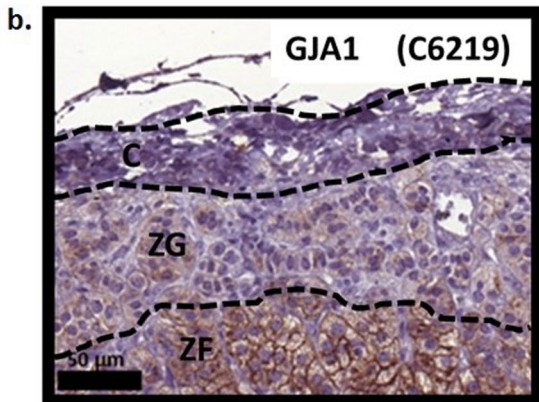

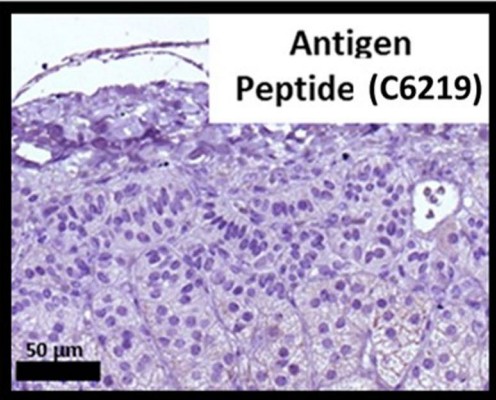

**Extended Data Fig. 3 | GJA1 expression in human adrenals. a,** *GJA1* cDNA expression in laser capture microdissected (LCM) zona glomerulosa (ZG) samples. *Left, s*elective cresyl violet staining of ZG for LCM RNA sample extraction. The capsule (C), zona fasciculata (ZF), and zona reticularis (ZR) do not retain the cresyl violet dye. *Right,* PCR of cDNA from LCM ZG and ZF RNA samples. HepG2 cells were used as a negative control as they do not express GJA1. ZF LCM samples had 5.45-fold higher expression of *GJA1* cDNA than ZG LCM samples. Full-length gels are provided as Source Data Extended Data

Fig. 3. **b,** Immunohistochemical GJA1 staining in the absence and presence of the antigen peptide. Adrenal sections from patient P1 were immunostained with GJA1 antibody (C6219, Sigma-Aldrich) in the absence or presence of the competing antigen peptide. Comparison of the serial sections show true staining of GJA1 protein in ZG/pericapsular adrenal cells, although less than ZF cells. Identification of adrenal zones were based on IHC for ZG and ZF markers shown in Supplementary Fig. 4c.

a.

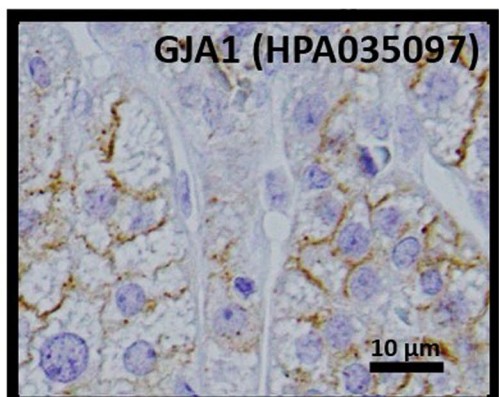
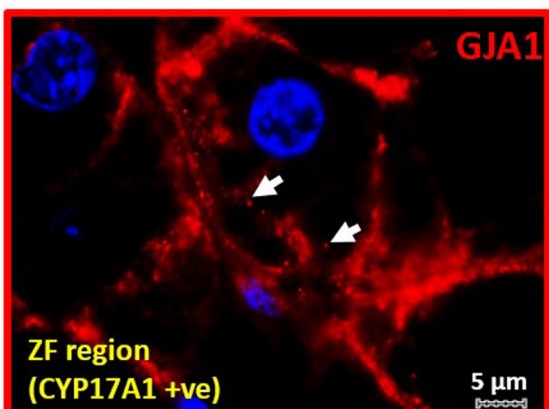

b.

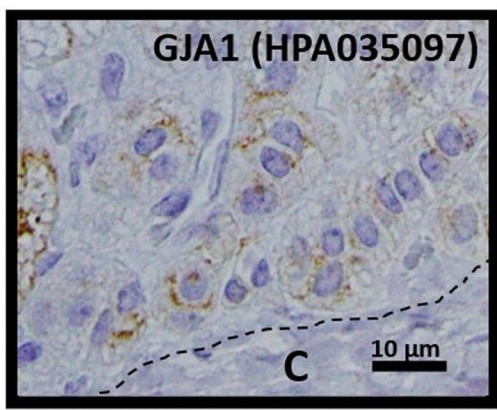
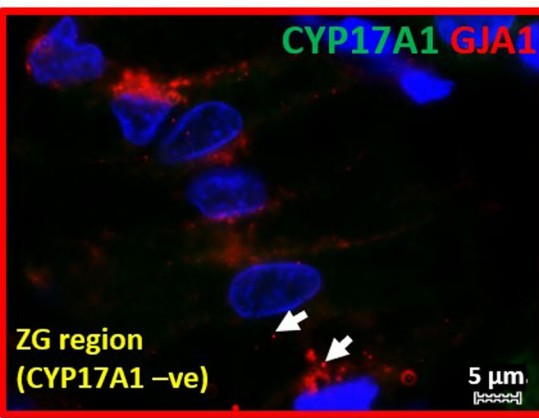

**Extended Data Fig. 4 | Different cellular expression of GJA1. a**, Zona fasciculata (ZF) staining of GJA1 in regions of CYP17A1 positive (+ve) cells. Membranous GJA1 staining (anti-GJA1, HPA035097, Sigma-Aldrich) in ZF, on IHC (*left*) and immunofluorescence (IFC, *right*). These cells far from the capsule of the adrenal cortex were positive for CYP17A1 (image of CYP17A1 shown in Extended Data Fig. 2e). Nuclei of cells were stained with either hematoxylin (IHC) or DAPI (IFC, blue). Puncta staining detected by IFC similar to that seen with IHC is highlighted by white arrows. **b**, Zona glomerulosa (ZG) staining of GJA1 in regions of CYP17A1 negative (-ve) cells. Punctate staining of GJA1 (anti-GJA1, HPA035097, Sigma-Aldrich) at sites of ZG cell contacts, on IHC (*left*) and IFC (*right*). These cells adjacent to the capsule (C) were negative for CYP17A1. Nuclei of cells were stained with either hematoxylin (IHC) or DAPI (IFC, blue). Puncta staining detected by IFC similar to that seen with IHC is highlighted by white arrows.

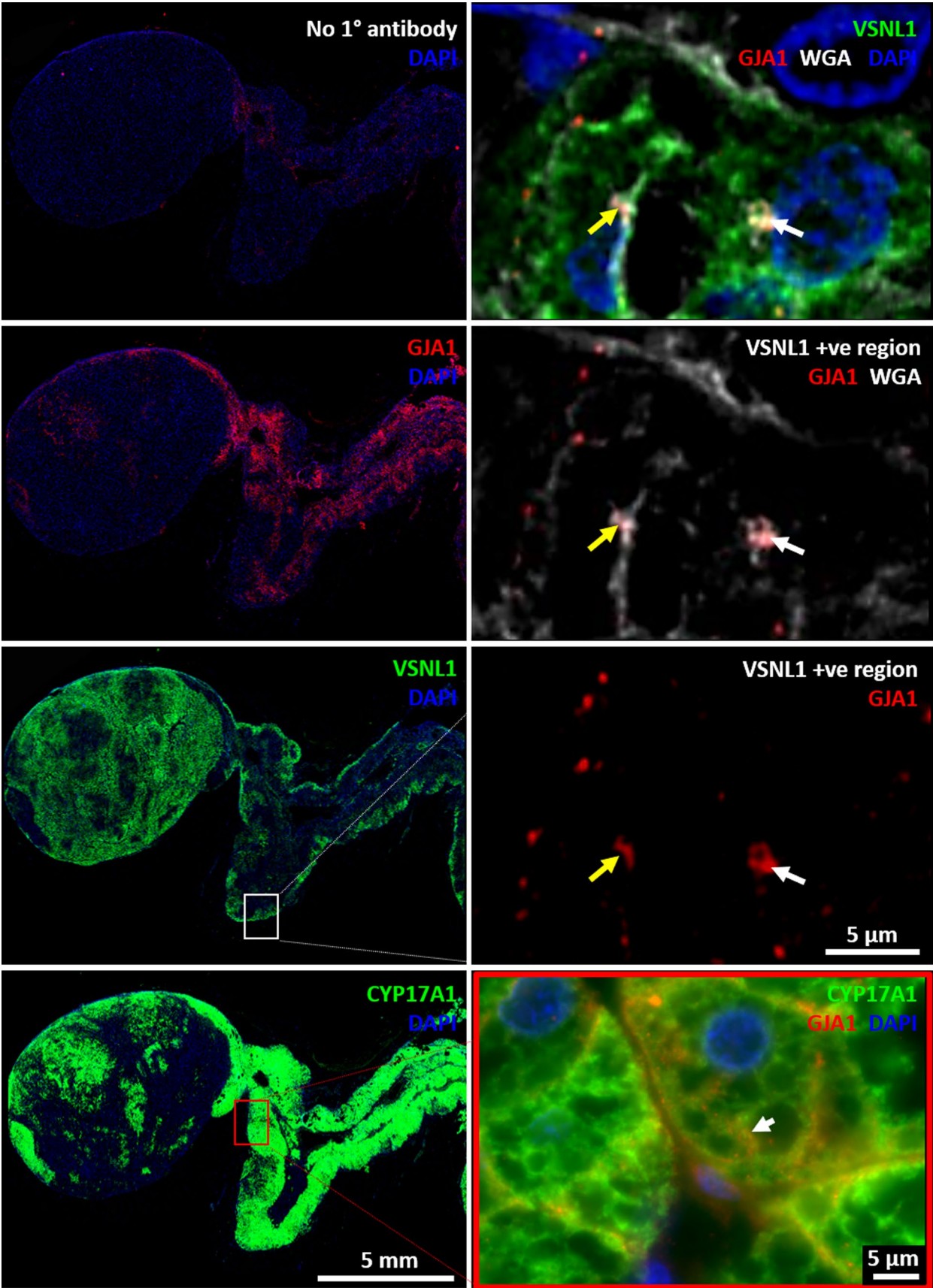

**Extended Data Fig. 5 | GJA1 expression in peri-capsular cells expressing VSNL1 (ZG marker) co-localizes with wheat germ agglutinin (WGA; cell membrane marker).** IFC staining of CYP17A1, VSNL1 (a ZG marker), and GJA1 was performed on serial sections from an adrenal harboring an APA. IFC staining color-coded as indicated in image. A no primary (1°) antibody control was performed to take into account autofluorescence that occurs in the region. GJA1 punctate expression, annular GJ (white arrow), and GJ plaque (yellow arrow) co-localizes with WGA in VSNL1 positive (+ve) cells. *Left*, scans of the serial sections. *Right*, zoomed image within boxed regions in scans. *Bottom*, image of the CYP17A1 positive cells shown in Extended Data Fig. 4.

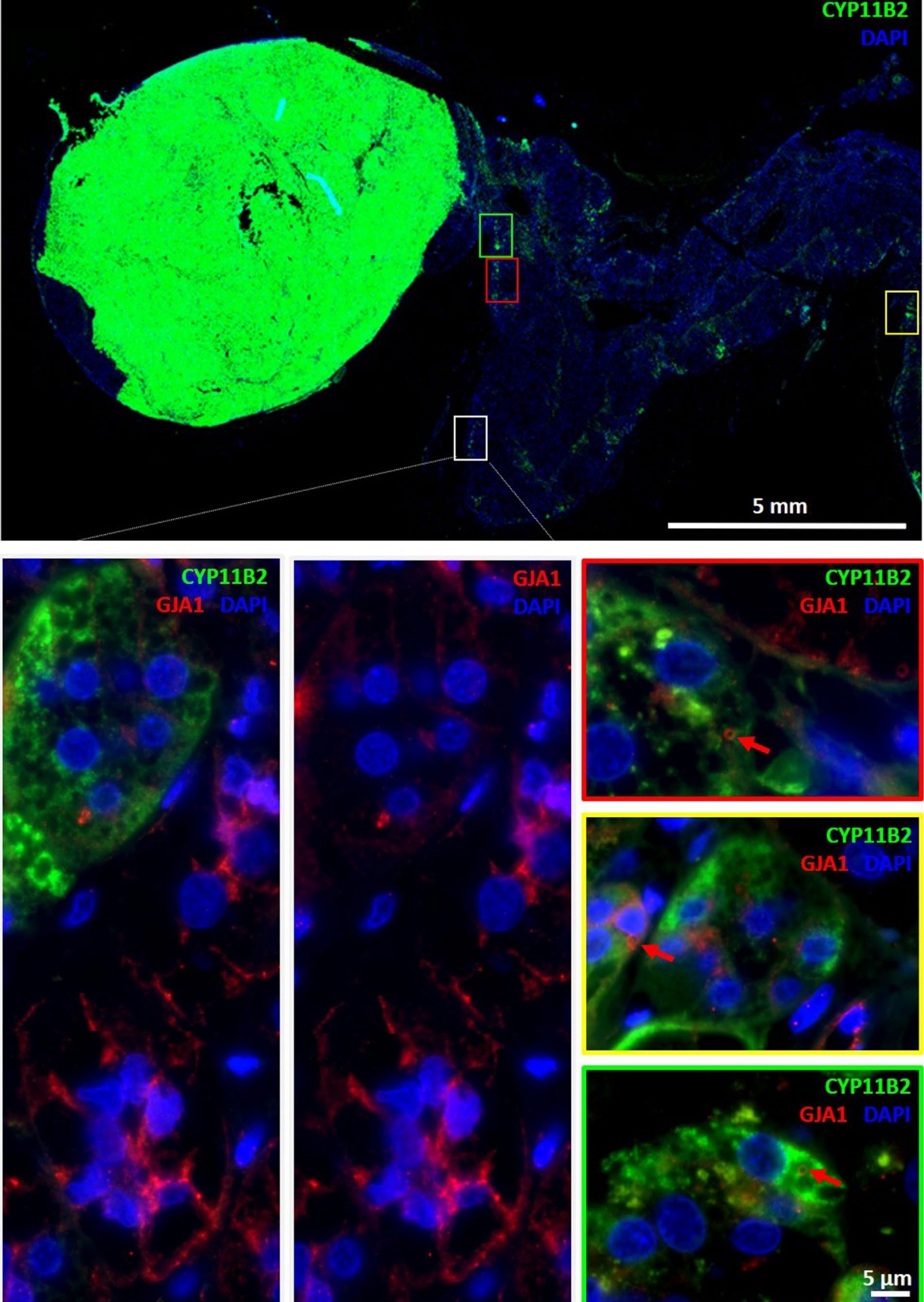

**Extended Data Fig. 6 | Presence of annular GJA1 in APM regions (CYP11B2-expressing cells).** IFC staining of CYP11B2 and GJA1 was performed on serial sections of the same adrenal shown in Extended Data Fig. 5. IFC staining is color-coded as indicated in image. GJA1 expression in APM regions was lower compared to adjacent adrenal cortex. Annular GJs (red arrows) were present in APM. *Top*, scan of the serial adrenal section; *Bottom*, zoomed images of ZF region and APM regions highlighted in scan.

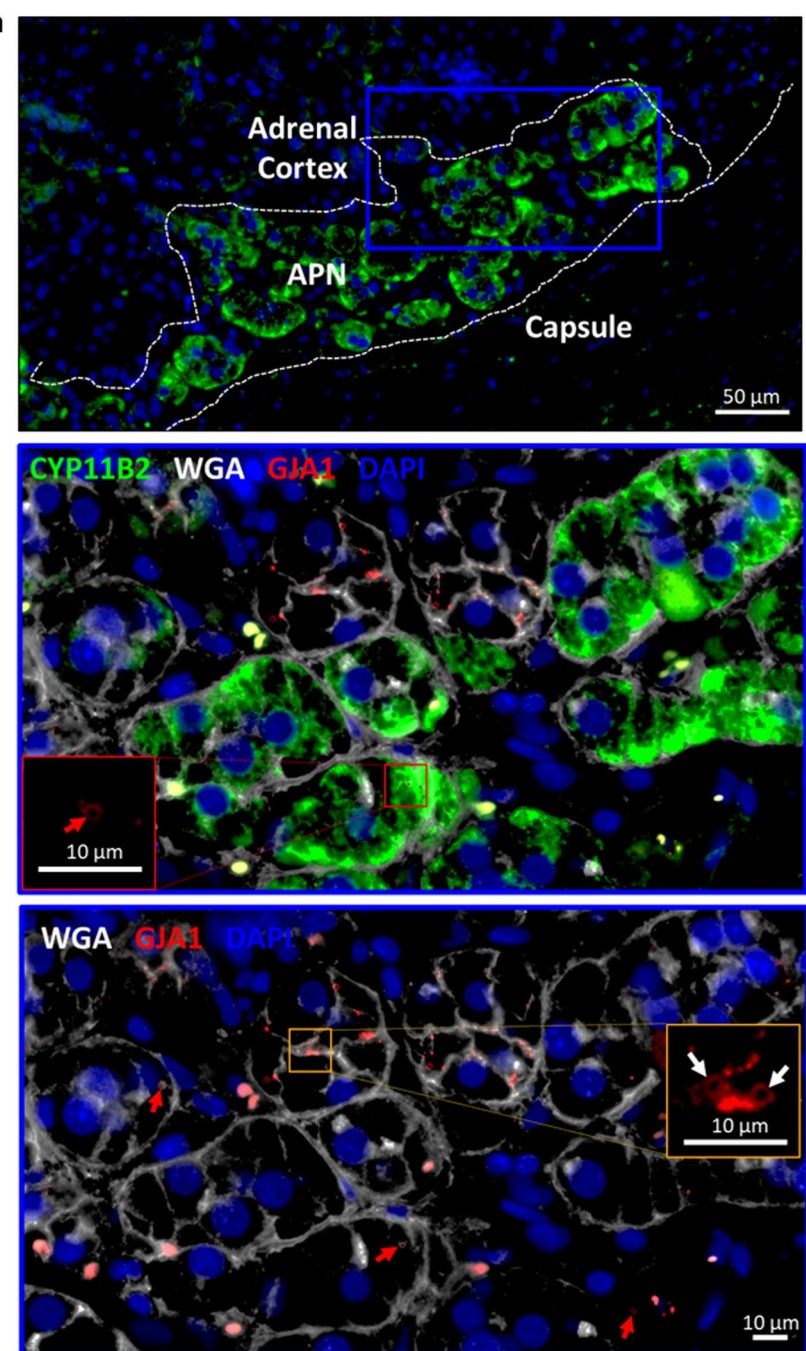

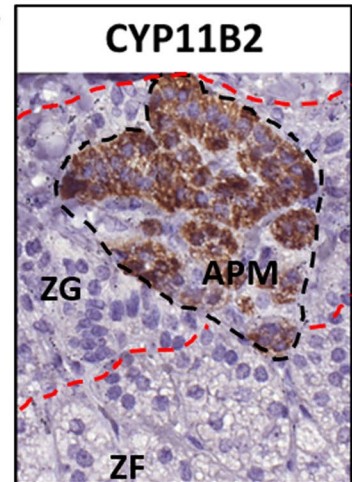

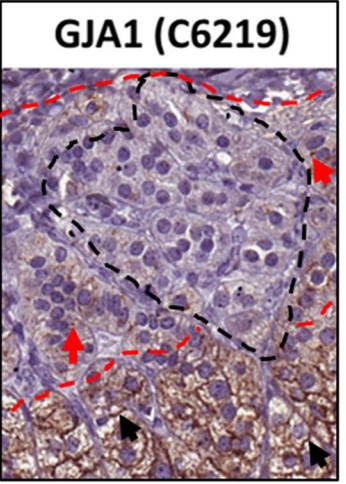

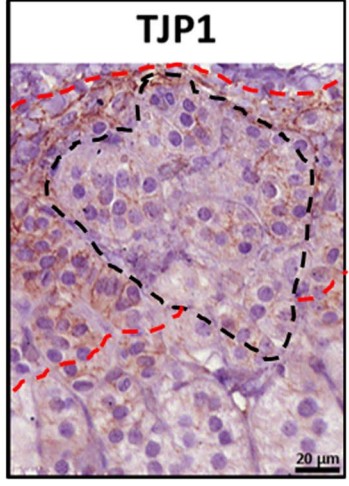

**Extended Data Fig. 7 | GJA1 and TJP1 expression is decreased in CYP11B2-expressing ZG cells. a**, APM regions (CYP11B2-expressing cells) have less GJA1 expression on cell membrane. IFC staining of WGA (white), CYP11B2 (green), and GJA1 (red) in an adrenal harboring an APA. Reduced membranous expression of GJA1 and a few annular GJs (red arrows) are seen in CYP11B2-positive cells (red box insert). Linear gap junction plaques (orange box insert) with budding annular GJs (white arrows) can be seen where WGA and GJA1 co-localize (orange staining).

**b**, TJP1 expression is also decreased in APM compared to adjacent ZG. IHC for CYP11B2, GJA1, and TJP1 in adrenal sections from patient P1. IHC for GJA1 was performed using the same antibody as in Extended Data Fig. 2b. GJA1 and TJP1, another cell junction protein, had decreased expression in APM (demarcated by black dashed lines) compared to adjacent ZG (demarcated by the red dashed lines). Membranous GJA1 staining in ZF and punctate GJA1 staining in ZG are highlighted by black and red arrows, respectively.

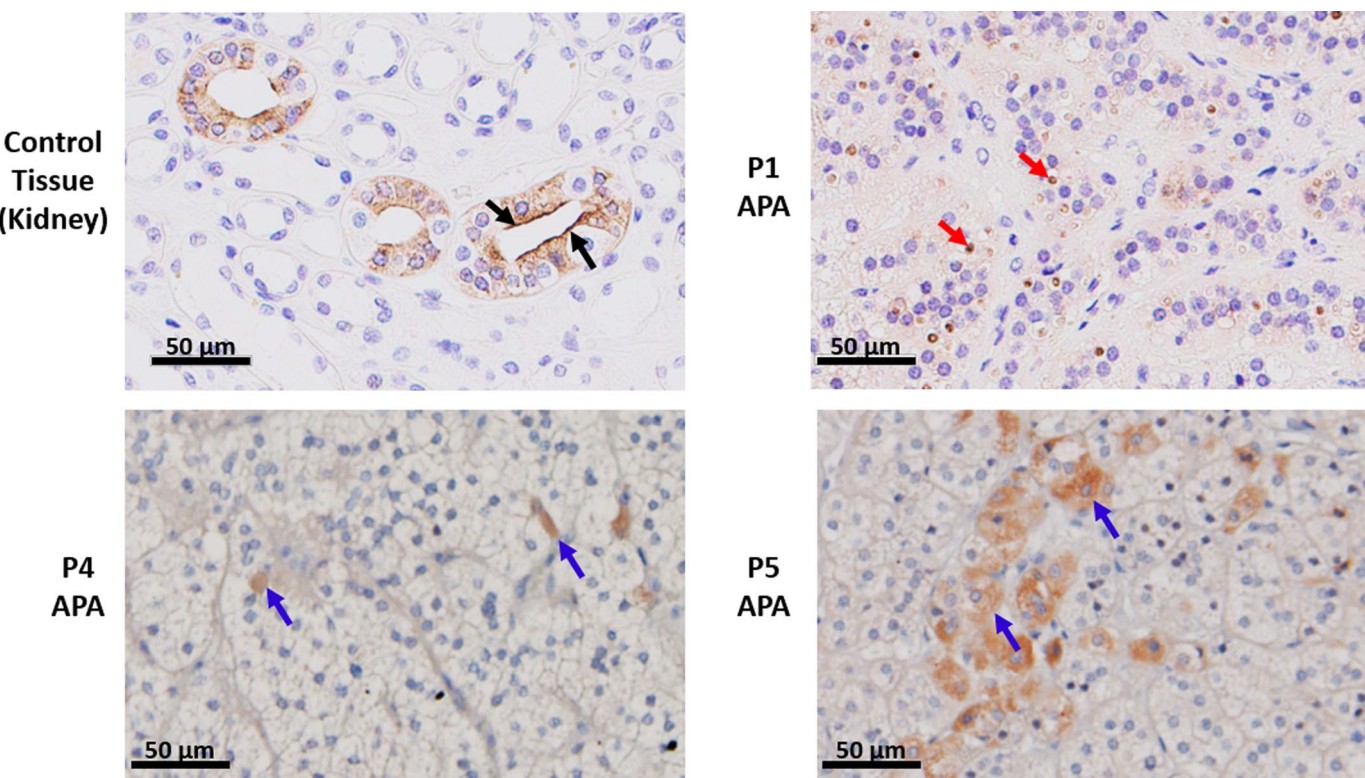

**Extended Data Fig. 8 | Heterogeneous subcellular expression of AQP2 in CADM1-mutant APAs.** IHC for APQ2 was performed on a positive control tissue section (from a human kidney) and on adrenal tissue sections containing APA of P1 (top right), APA of P4 (bottom left), and APA of P5 (bottom right). Selective membranous staining (black arrows) was seen in the collecting ducts of the human kidney, whereas using the same parameters, APA of P1 had solitary cytoplasmic bodies staining with AQP2 (red arrows), while APAs of P4 and P5 had apparent cytoplasmic staining (blue arrows).

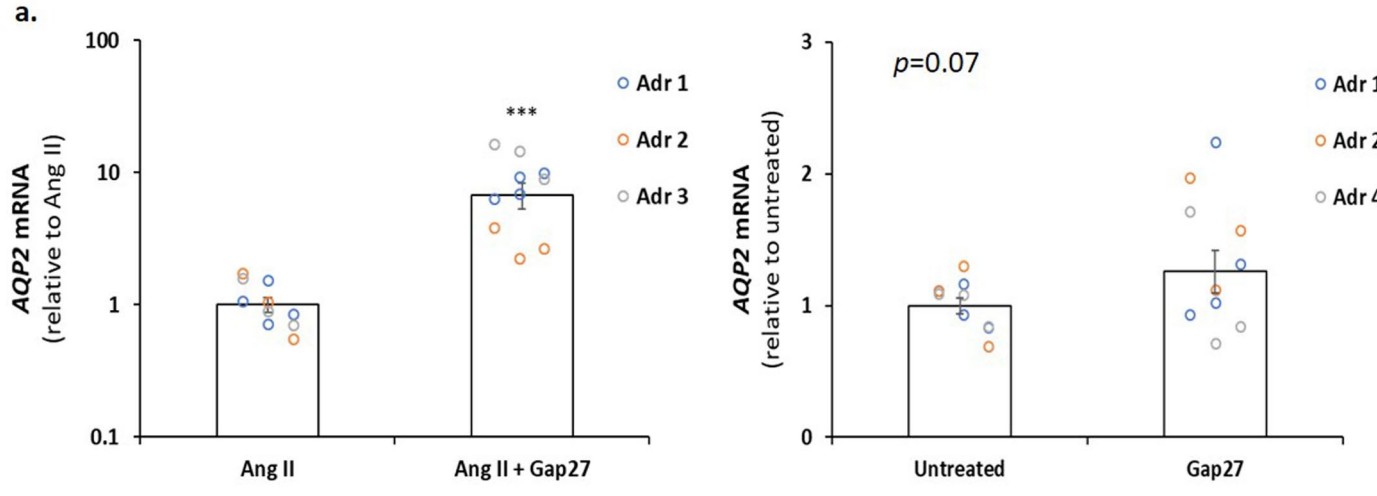

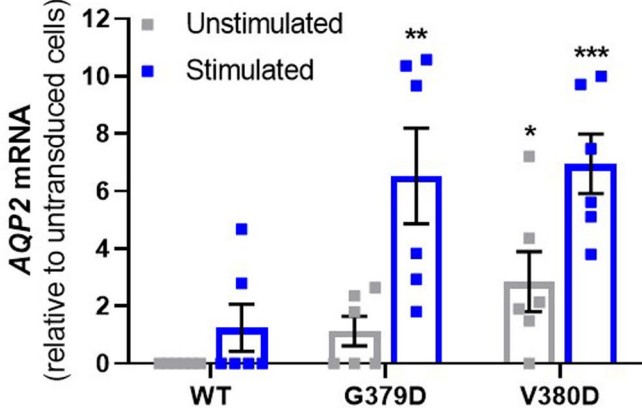

**Extended Data Fig. 9 | AQP2 expression in adrenal cells treated with Gap27 or transduced with mutant *CADM1*. a**, *AQP2* mRNA expression in primary human adrenal cells treated with Gap27 in the presence or absence of angiotensin II (Ang II). *AQP2* mRNA expression of cells from Fig. 5d and Supplementary Fig. 7g is shown. mRNA expression for primary adrenal cells was normalized by β-actin. Fold change was expressed relative to Ang II-stimulated cells (*left*) or untreated cells (*right*) for each adrenal (*n* = 10). Data represent mean, error bars show s.e.m. Statistical analysis was performed using two-sided Student's *t*-test. *AQP2* expression was significantly increased in stimulated cells treated with Gap27 (***P = 0.0002) but not in unstimulated cells (P = 0.730). **b**, *AQP2* mRNA expression in cells transduced with wild-type (WT) or mutant *CADM1* (V380D or G379D). Transduction of mutant *CADM1* (both mutants in 442aa and

453aa isoforms) increased *AQP2* mRNA expression compared to wild-type in Angiotensin II (Ang II) stimulated H295R cells (*n* = 6). This trend was also seen in unstimulated cells. All values expressed as average fold-change of both isoforms (3 independent wells each) relative to untransduced cells. Data represent mean, error bars show s.e.m. Statistical significance between groups were measured using the two-sided Kruskal-Wallis H test; Unstimulated, χ²(2) = 7.362, P = 0.0177 and Ang II-stimulated, χ²(2) = 8.592, P = 0.0076, respectively. Post-hoc analysis performed using Dunn's multiple comparison test, compared to WT. *AQP2* mRNA expression was significantly increased in unstimulated V380D (*P = 0.0138) and stimulated G379D (**P = 0.0335) and V380D (***P = 0.0155) cells. *n* = biological independent replicates from 3 independent experiments. The underlying statistics are provided as Source Data Extended Data Fig. 9.

Morris J. Brown

# Reporting Summary

Nature Research wishes to improve the reproducibility of the work that we publish. This form provides structure for consistency and transparency in reporting. For further information on Nature Research policies, see our Editorial Policies and the Editorial Policy Checklist.

## Statistics

For all statistical analyses, confirm that the following items are present in the figure legend, table legend, main text, or Methods section.

| n/a | Confirmed | |
|---|---|---|
| ☐ | ☒ | The exact sample size (*n*) for each experimental group/condition, given as a discrete number and unit of measurement |
| ☐ | ☒ | A statement on whether measurements were taken from distinct samples or whether the same sample was measured repeatedly |
| ☐ | ☒ | The statistical test(s) used AND whether they are one- or two-sided *Only common tests should be described solely by name; describe more complex techniques in the Methods section.* |
| ☐ | ☒ | A description of all covariates tested |
| ☐ | ☒ | A description of any assumptions or corrections, such as tests of normality and adjustment for multiple comparisons |
| ☐ | ☒ | A full description of the statistical parameters including central tendency (e.g. means) or other basic estimates (e.g. regression coefficient) AND variation (e.g. standard deviation) or associated estimates of uncertainty (e.g. confidence intervals) |
| ☐ | ☒ | For null hypothesis testing, the test statistic (e.g. *F*, *t*, *r*) with confidence intervals, effect sizes, degrees of freedom and *P* value noted *Give P values as exact values whenever suitable.* |
| ☒ | ☐ | For Bayesian analysis, information on the choice of priors and Markov chain Monte Carlo settings |
| ☒ | ☐ | For hierarchical and complex designs, identification of the appropriate level for tests and full reporting of outcomes |
| ☒ | ☐ | Estimates of effect sizes (e.g. Cohen's *d*, Pearson's *r*), indicating how they were calculated |

*Our web collection on statistics for biologists contains articles on many of the points above.*

## Software and code

Policy information about availability of computer code

**Data collection**

3DHISTECH Pannaromic MIDI scanner Software v1.18 (3D HISTCH, Hungary) for IHC slide image acquisition
CaseViewer Software v2.4 for IHC slide image acquisition
Vectra® automated imaging system software v.3.0.5 (Perkin Elmer) for IHC slide image acquisition

TissueFAXS SL Q+ upright epifluorescence microscope and viewed with TissueFAXS slide viewer 7.0 software for IFC image acquisition

NIS-Elements v4.5 (Nikon, Japan): fluorescence and transmitted light microscopy image acquisition

FLUOstar Omega series v5.7 (BMG Labtech, UK): Homogeneous Time Resolved Fluorescence 24 (HTRF) aldosterone assay acquisition

BD FACSDiva Software v6.1.3 (BD Biosciences, USA): Fluorescence-activated cell sorting (FACS) for lentivirus titration

CFX Manage TM Software v3.1 (Bio-Rad, USA) or 7900 SDS v2.4.1 (Applied Biosystems, USA): RT-PCR data acquisition

NextSeq 500 high-output 150 cycle kit v2.5 and bcl2fastq Conversion Software v1.8.4: RNA sequencing

**Data analysis**

For protein structure modelling: QUARK March 2018 (https://zhanglab.ccmb.med.umich.edu/QUARK/) and GROMACs (2019 release) was used for structure analysis of the transmembrane domains and intercellular domains of wild-type and mutant CADM1 because no template structure of these domains for homology modelling were available. PyMOL v.2.1 (Schrödinger, LLC) for visualization of the CADM1 structures. ZDOCK v.3.0.2 (http://zdock.umassmed.edu/) was used for docking analysis for the dimerization and tetramerization of the CADM1. The proteins structure of the transmembrane domain of CADM1 was also separately modelled using Phyre2 v2 (http://www.sbg.bio.ic.ac.uk/~phyre2/html/page.cgi?id=index) and TMDOCK (https://membranome.org/tmdock) was used to predict the insertion and homodimerization

of the transmembrane region of the wild type and Phyre predicted mutant (G379D and V380D) structures.
Statistical analysis was performed using R software, R Core Team. R (v.3.5.0) was used for one-way ANOVA followed by Tukey's post hoc tests (ptukey function of R).

GATC Viewer v.1.00 or UniPro UGENE v.1.28.1 for Sanger sequencing alignment

Flowing Software v.2.5.1 for lentivirus titration using FACS

Microsoft Excel 2016 for HTRF aldosterone assay and qPCR (2-ΔΔCT) calculations

Fiji ImageJ v.1.52p Java 1.8.0_172 for western blot quantification
FIJI ImageJ software v1.8.0_72 was used to prepare the IF images for presentation

Prism (Graphpad software v.8.4.3 or later) for statistical analysis for aldosterone, qPCR, western blot and calcium oscillation data.

Partek Flow software (Partek, St. Louis, Missouri, United States), including its annotation tool and GSA tool was used for RNA sequencing analysis. STAR -2.6.1d was used for sequence alignment and DAVID Bioinformatics Resources v.6.8 (https://david.ncifcrf.gov/tools.jsp) for gene enrichment analysis.

For manuscripts utilizing custom algorithms or software that are central to the research but not yet described in published literature, software must be made available to editors and reviewers. We strongly encourage code deposition in a community repository (e.g. GitHub). See the Nature Research guidelines for submitting code & software for further information.

## Data

Policy information about availability of data

All manuscripts must include a data availability statement. This statement should provide the following information, where applicable:
- Accession codes, unique identifiers, or web links for publicly available datasets
- A list of figures that have associated raw data
- A description of any restrictions on data availability

The RNAseq dataset used to generate Fig. 6, Supplementary Fig. 10, Supplementary Table 5b, Supplementary Tables 6-9 is shown in Supplementary Data 1-3. The WES raw data is publicly available from the Sequence Read Archive (https://www.ncbi.nlm.nih.gov/sra/docs/) under accession nos. PRJNA732946 and PRJNA729738. Source data for Figure 2c-e, Figure 4b, Figure 4d, Figure 5b-d, Extended Figure 9a-b, Supplementary Figure 4a-b, Supplementary Figure 5d, Supplementary Figure 8a-b, Supplementary Figure 8d-e, Supplementary Figure 8g-h, Supplementary Figure 9c-d is provided with the paper.

# Field-specific reporting

Please select the one below that is the best fit for your research. If you are not sure, read the appropriate sections before making your selection.

☒ Life sciences        ☐ Behavioural & social sciences        ☐ Ecological, evolutionary & environmental sciences

For a reference copy of the document with all sections, see nature.com/documents/nr-reporting-summary-flat.pdf

# Life sciences study design

All studies must disclose on these points even when the disclosure is negative.

| | |
|---|---|
| Sample size | Sample size used in experiments were determined by pilot studies which generally indicated that n values of minimum of 3 was required to give a significant result. The following criteria were also used to pre-determined sample size:<br>Consistency in experimental replicates performed.<br>Consistency in difference between isoforms and mutations detected, compared to wild-type or untreated cells.<br>A minimum of 2 independent results to represent experimental variability (number specified in figure legends or methods section).<br>For dye transfer assays, all cells within a 50 um radius of the original cell of interest were included in the analysis.<br>For quantification of gap junction formation in H295R cells, cells within each well were systemically imaged. All successfully transfected cells were accounted for to limit bias.<br>For calcium oscillation experiments, ROIs were determined for all cells in the image field. For quantification the condition with the fewest numbers of cells in a single image field across all experiments was used to determine the sample size (44-46). The 44-46 cells selected for quantification in all other conditions were done so at random. |
| Data exclusions | No data that passed quality control were excluded. |
| Replication | All experiments were repeated, a minimum two times, with independent cohorts of biological samples. All attempts at replication was successful and all experimental findings were reliably reproduced. |
| Randomization | Randomization is not relevant to our study because experimental groups are determined by genotype. |
| Blinding | Investigators were not blinded to experimental group as data collection and quantification was objective and not impacted by investigators presumptions. |

# Reporting for specific materials, systems and methods

We require information from authors about some types of materials, experimental systems and methods used in many studies. Here, indicate whether each material, system or method listed is relevant to your study. If you are not sure if a list item applies to your research, read the appropriate section before selecting a response.

## Materials & experimental systems

| n/a | Involved in the study |
|-----|------------------------|
| ☐ | ☒ Antibodies |
| ☐ | ☒ Eukaryotic cell lines |
| ☒ | ☐ Palaeontology and archaeology |
| ☒ | ☐ Animals and other organisms |
| ☐ | ☒ Human research participants |
| ☒ | ☐ Clinical data |
| ☒ | ☐ Dual use research of concern |

## Methods

| n/a | Involved in the study |
|-----|------------------------|
| ☒ | ☐ ChIP-seq |
| ☒ | ☐ Flow cytometry |
| ☒ | ☐ MRI-based neuroimaging |

## Antibodies

**Antibodies used**

Primary antibodies used
CADM1 C-terminal (polyclonal), 1:5000, Sigma-Aldrich, S4945
CADM1 N-terminal (3E1), MBL, CM004-3
β-actin (6D1), dilution 1:2000, Medical & Biological Laboratories (MBL), PM053
GAPDH (GAPDH-71.1), dilution 1:40,000, Sigma-Aldrich, G8795
GJA1 (polyclonal), dilution 1:8,000 for Western Blot, dilution 1:500 for IHC, Sigma-Aldrich, C6219
GJA1 (polyclonal), dilution 1:100 (for IHC), Sigma-Aldrich, HPA035097
GJC1 (polyclonal), dilution 1:2,500, Invitrogen, PA5-79311
KCNJ5 (polyclonal), dilution 1:100, Sigma-Aldrich, HPA017353
TJP1 (polyclonal), dilution 1:400, Sigma-Aldrich, HPA001636
VSNL1 (clone 2D11), 0.5 uL/ml, EMD Millipore, MABN762
DAB2 (polyclonal), dilution 1:500, Atlas Antibodies, HPA028888
Wheat germ agglutinin, Alexa FluorTM (AF) 647 conjugate, 5ul/ml, Invitrogen, W32466
CYP17A1 (polyclonal), dilution 1:2,000 (10-19-64-7710 mouse IgG 25), kindly gifted from Professor Celso E Gomez-Sanchez (The University of Mississippi, USA)
Custom made antibodies to CYP11B2 (mouse mAb) [1], dilution 1:100 (41-13B19/19/2018), kindly gifted from Professor Celso E Gomez-Sanchez (The University of Mississippi, USA)
Custom made rabbit polyclonal anti-CADM1 antibody [2], dilution 1:2,000

Secondary antibodies used
Anti-rabbit (polyclonal), Sigma-Aldrich, A0545
Anti-chicken (polyclonal), MBL, PM010-7
Anti-mouse (H&L), Vector Laboratories, BA-9200
Anti-mouse 488, Invitrogen, A-10680
Anti-rabbit AF 568 (IgG), Invitrogen, A-11011

**Validation**

Antibodies were selected based on previous experience of the investigators and their use in the literature on human cell lines. Antibodies previously not used by the investigators were validated using negative +/- peptide controls.

All commercial antibodies used have been validated in previously published studies as listed in their manufactures' websites, unless otherwise stated:
CADM1 C-terminal https://www.sigmaaldrich.com/catalog/product/sigma/s4945?lang=en®ion=GB
CADM1 N-terminal https://www.mblbio.com/bio/g/dtl/A/index.html?pcd=CM004-3
tGFP https://www.origene.com/catalog/antibodies/tag-antibodies/ta150041/mouse-monoclonal-turbogfp-antibody-clone-oti2h8
EGFP https://www.abcam.com/gfp-antibody-ab5450.html
β-actin https://ruo.mbl.co.jp/bio/e/dtl/A/?pcd=M177-3
β-actin https://www.sigmaaldrich.com/catalog/product/sigma/a2066?lang=en®ion=GB
GAPDH https://www.sigmaaldrich.com/catalog/product/sigma/g8795?lang=en®ion=GB
Connexin-43 (C6219) https://www.sigmaaldrich.com/catalog/product/sigma/c6219?lang=en®ion=GB
Connexin-43 (HPA035097) https://www.sigmaaldrich.com/catalog/product/sigma/hpa035097?lang=en®ion=GB
Connexin-45 No citations are available on manufactures' website. For Western blot validation, five different CX-45 antibodies were validated in this study by comparing protein expression in two cell lines (HEK293T and H295R) with known high endogenous expression of CX-45, and in silenced cells. The results were correlated with mRNA expression. Only one antibody (selected) was effective, although yielded non-specific bands.
KCNJ5 https://www.sigmaaldrich.com/catalog/product/sigma/hpa017353?ang=en®ion=GB&gclid=Cj0KCQiA0-6ABhDMARIsAFVdQv-y-FwndiyP5lk_zuNbGJ7f9xfjQ8C6TNydycq4Jia22jqYUtWDcQYaAslWEALw_wcB
TJP1 https://www.sigmaaldrich.com/catalog/product/sigma/hpa001636?lang=en®ion=GB
VSNL1 https://www.merckmillipore.com/GB/en/product/Anti-VSNL1-Antibody-clone-2D11,MM_NF-MABN762
DAB2 https://www.atlasantibodies.com/products/antibodies/primary-antibodies/triple-a-polyclonals/dab2-antibody-hpa028888/

Custom made antibodies to CYP11B2 [1] and CADM1 [2] were validated using negative controls. The CYP11B2 antibody has been extensively cited, including [3-5]. Citations for CADM1 antibody include [6-8], for CYP17A1 include [9].

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

# Eukaryotic cell lines

Policy information about cell lines

| Cell line source(s) | H295R cells were purchased from ECACC. NIH-3T3 cells were purchased from ATCC. Hek293T cells were a gift from Professor Xiao's lab, William Harvey Research Institute, Queen Mary University of London, originally form ATCC. |
| --- | --- |
| Authentication | The cell lines used were not authenticated. |
| Mycoplasma contamination | The cell lines were not tested for mycoplasma contamination |
| Commonly misidentified lines (See ICLAC register) | No commonly misidentified cell lines were used in the study |

# Human research participants

Policy information about studies involving human research participants

| Population characteristics | The human research participants were all adults with a diagnosis of Primary Aldosteronism who proceeded to unilateral adrenalectomy. Diagnosis, investigations and decision for surgery were made in accordance with local institutional guidelines. The demographics of the cohort in which WES was performed is detailed in the Supplementary Table 1 |
| --- | --- |
| Recruitment | Participants were all recruited from secondary or tertiary care settings, at the hospital where adrenalectomies were being performed. All patients undergoing adrenalectomy at the specified sites were offered the opportunity to participate, with adrenal tissue collection and genetic analysis performed only in those who gave written informed consent. Potential selection bias includes willingness of patients to be enrolled into the study and the availability of adrenal tumor tissue with DNA suitable for WES. |
| Ethics oversight | Consent for adrenal tissue collection and genetic investigations were taken in accordance with local institutional guidelines and approved by the local ethics committee. The individual ethics committees for each centre are: Cambridgeshire Research Ethics Committee for Addenbrooke's Hospital, University of Cambridge, UK; The Ethics Committee of University Hospital Munich, Germany; Cambridge East Research Ethics Committee for St Bartholomew's Hospital, Queen Mary University of London, UK; Assistance Publique-Hôpitaux de Paris Research Ethics Committee, Paris, France; The Institutional Review Board for Tohoku University Hospital, Sendai, Japan; The Ethics Committee of Hiroshima University, for Graduate School of Biomedical and Health Sciences, Hiroshima University, Hiroshima, Japan; Institutional review board, University of Pennsylvania, Philadelphia, United States of America; University Hospital Hradec Kralove Ethics Committee, Czech Republic; National University of Malaysia Research Ethics Committee, Malaysia. |

Note that full information on the approval of the study protocol must also be provided in the manuscript.

