## [Peer Review File · Nature Genetics]

Peer Review Information

Manuscript Title: Somatic *CADM1* mutations in aldosterone-producing adenomas and gap junction-dependent regulation of aldosterone production

Corresponding author name(s): Professor Morris Brown, Dr Elena Azizan

Reviewer Comments & Decisions:

Decision Letter, initial version:
--

15th September 2020

Dear Morris,

Your Letter entitled "Somatic intramembranous mutations of *CADM1* in aldosterone-producing adenomas, and gap junction dependent regulation of aldosterone production" has now been seen by three referees, whose comments are copied below. In the light of their advice, we have decided that we cannot offer to publish your manuscript in Nature Genetics.

In particular, while the referees find your work of some interest, they raise overlapping concerns about the strength of the novel conclusions that can be drawn at this stage, particularly with regards to the mechanism by which *CADM1* mutations contribute to disease pathophysiology. We are persuaded that these reservations are sufficiently important as to preclude publication of this study in Nature Genetics.

I am sorry we cannot be more positive on this occasion but hope you will find our referees' comments helpful when preparing your paper for submission elsewhere.

Sincerely,
Kyle

Kyle Vogan, PhD
Senior Editor
Nature Genetics
<https://orcid.org/0000-0001-9565-9665>

Referee expertise:

Referee #1: Genetics, hypertension, primary aldosteronism

Referee #2: Genetics, adrenal tumors

Referee #3: Organogenesis, adrenal development

Reviewers' Comments:

Reviewer #1:

Remarks to the Author:

In this manuscript, Wu and colleagues describe the identification of novel heterozygous somatic mutations (p.Gly379Asp and p.Val380Asp) in the gene *CADM1* in 2/163 aldosterone-producing adenomas (1.2%). They demonstrate that expression of mutant *CADM1* has large effects on the expression of *CYP11B2* (aldosterone synthase) in the aldosterone-producing human H295R cell line (>10-fold increase compared to wildtype), with somewhat smaller effects on aldosterone production. The authors proceed to suggest that this observation is linked to an effect of mutant *CADM1* on gap junction formation. Lastly, their study includes RNA sequencing to identify underlying pathways, with a potential link to clock genes.

Even though *CADM1* mutations account for only a very small fraction of aldosterone-producing adenomas, this study is very interesting because the involved pathways appear to be distinct from mutations in the many previously described disease genes (typically ion channels and pumps). The effect on aldosterone synthase expression is impressive. A potential link to clock genes and cyclic variation in aldosterone production could also have clinical implications (need for repeated screening). The manuscript is original and significant, and the data are well presented. I have no concerns about the statistical analysis. My major concern is that despite extensive functional studies, the underlying pathophysiology remains somewhat unclear, in particular with regard to the link to gap junctions, as I will outline in more detail below.

Detailed comments:

Major

1. Figure 4 (c) – Both *CADM1* mutants and sh*CADM1* appear to inhibit dye transfer, suggesting that both prevent gap junction formation. Because the effect on *CYP11B2* expression is very different (expression rises strongly when mutants are expressed (Fig. 3 (a)), but is suppressed when *CADM1* is silenced (Fig. 3 (b) iii)), these results suggest to me that gap junction formation or gap junction function may not be the major underlying pathophysiology of *CADM1* mutations.
2. Related to 1., the CX43 staining shown in Fig. 5 is extremely weak. In the absence of valid controls (e.g., secondary antibody only, peptide control), this could well be interpreted as background. This is of particular importance given prior data that there is little or no CX43 present in the ZG (Bell & Murray, *Front Endocrinol* 2016).
3. In addition, what is labeled as ZG in Fig. 5 (a)i in the adrenal from the patient with *CADM1*

mutation does not appear to be ZG based on morphology (streaks of cells as in fasciculata). Given that CYP11B2 is negative (were these patients not treated with Spironolactone to raise renin to non-suppressed values?), it is unclear how the ZG was identified.

4. A major claim – that mutant CADM1 pushes apposed cells beyond reach of their connexons – is not supported by data and thus remains speculative.

5. Fig. 5 (b): If treatment with Gap27 has a large effect on CYP11B2 expression, but silencing of CX43 does not, then this suggests that the effect of Gap27 treatment is not mediated through inhibition of CX43. The authors demonstrate that even inhibiting CX43 and CX45 together has a much smaller effect on CYP11B2 expression than Gap27 treatment. This adds to my doubts about gap junctions as major part of the pathophysiology of CADM1 mutations.

6. I would suggest showing in a Suppl. Table mutant and WT reads for the CADM1 mutation in tumor and normal tissue. I would also suggest displaying all other somatic mutations discovered. This is relevant because the absence or presence of any additional somatic mutations that may account for aldosteronism or growth in the tumor would help to assess the role of CADM1 mutations.

7. The fraction of tumors without mutations in any known disease genes (20%) is quite large compared with recent papers from the Rainey group cited in the study. I would suggest showing the somatic mutations discovered in these tumors – were any discovered?

Minor

1. Could the authors comment on why the effect on aldosterone synthase expression is so much larger than the effect on aldosterone production?

2. Fig. 4 (a): The claim that silencing of CADM1 increases CX43 expression should be supported by quantitative data. Showing single images with low CX43 expression and a single Western blot appears insufficient.

3. Figure 1 (a) CACNA1ID – this should be CACNA1D (1, not I)

4. Figure 1 (b): Renin mass – this does not appear to be the correct description of what is shown (renin in mU/l, and then plasma renin activity for the lowest value). Looking at aldosterone and renin values, it is striking that the postoperative ARR is similar to the average of the 4 prior measurements, with renin still being borderline suppressed (taking into account the different units used). This is described by the authors as “cyclical aldosteronism”. Could the authors please comment on the time of day of these measurements? In most centers, ARR is determined in the morning, which should limit the effect of clock genes.

5. Figure 1 (c): Only part of the APA is positive for CYP11B2 in IHC – is this an artifact? If considered real, please explain. Otherwise, the staining should be repeated.

6. Fig. S1 (a): It is unclear to me what the numerous bands represent, at least the authors should comment on this or describe the blot appropriately.

7. Legend Fig. 4 (d): Hoechst, not Hoescht

8. Suppl. Fig. S7: Gene names cannot be clearly read.

Reviewer #2:

Remarks to the Author:

The paper by Xilin Wu and colleagues reports on the identification of somatic mutations of the transmembrane of *CADM1* in 2 small aldosterone producing adenomas (APA) that both were causing a moderate and variable level of aldosterone excess.

Functional expression of the *CADM1* mutants in adrenocortical cells was able to stimulate aldosterone synthase gene expression. The authors then demonstrate that this could be caused by longer distances between cells leading to dysregulation of Gap Junction by *CADM1* mutants.

This is the first report of *CADM1* mutations in APA. Although these mutations appears rare the elegant and various functional studies reported clearly demonstrate the role of *CADM1* and GJ in aldosterone regulation.

One interesting speculation is the phenotype of these 2 patients with intermittent/variable or cyclical secretion of aldosterone. This raise the hypothesis that diagnosis of PA would be often missed in patients with *CADM1* mutation. However there was a clear improvement of hypertension in these two patients. This is clearly a fascinating hypothesis because if this was the case one would speculate that moderate variable form of PA would need to be investigated more carefully to search for such patients. However it is at present quite speculative since it is difficult to established a genotype/phenotype correlation on only 2 patients. It would be interesting to investigate more case to build this hypothesis on higher numbers.

Specific comments:

1- It is disappointing that exome identify in 2 different non selected cohorts a somatic mutation in one tumor in each cohort while sanger sequencing of APA selected to be wild type for the known gene fails to identify additional cases. It seems according to the Sanger sequences shown on fig 2 that the mutant allele might be present in less than half of the cells. Could it be possible that the Sanger sequencing screening of the 53 wild type APA would have been less sensitive than ngs ? In this case targeted ngs for *CADM1* might have identified additional APA with *CADM1* mutation. An other option would be to sequence a larger number of sample.

2- Could *CADM1* be mutated in Aldosterone Producing Cells clusters (APCCs) ? This would be an other way to clearly establish the significance and importance of *CADM1* mutations in PA.

3- Pathology, immunohistochemistry and clinical outcome are clear proof of APA in the 2 patients. However the pre-operative investigation are puzzling because PA is not marked nor constant and that adrenal imaging are ambiguous as often with very small APA. Were a suppression test and an adrenal venous sampling done before surgery ?

4- The concept of illegitimate membrane receptor have been previously suggested in PA. Could it be a mechanism for the variable level of aldosterone secretion in these patients ? Does the RNAseq data

show upregulation of potential illegitimate receptor ?

5- The RNAseq experiments give interesting insights. It is unclear how the statistical analysis was done to rule out false discovery considering the numbers of analyzed genes ?

6- It would be interesting to see how frequent is this situation to use the set of genes identified by RNAseq to search in already available APA transcriptome others tumors with the same phenotype.

7- It seems that very few APA were included in the APQ2 IHC analysis. It would be interesting to see in a large cohort how frequent is the loss of APQ2 and this could be also a way to identify others APA with CADM1 mutation.

Reviewer #3:

Remarks to the Author:

This manuscript by Brown and colleagues explores the putative role of two (adjacent) missense mutations in the transmembrane domain of CADM1 in patients diagnosed with aldosterone-producing adenomas, a common and potentially curable cause of hypertension. The introduction of these mutations into the H295R adrenocortical cell line resulted in increased expression of CYP11B2, essential for aldosterone production, by 15-30 fold. In contrast, knock down with shRNA, or introduction of soluble CADM1 Ig-ectodomains abolished dye-transfer through gap junctions (GJs). This led to protein studies and molecular modelling which predicted increased exit angles of mutant (GOF) CADM1 with the resulting longer distance between cells preventing GJ formation. Additional in vitro dye transfer studies with H295R cells and RNAseq analysis following mutant transduction and AngII stimulation led to the conclusion that GJ communication plays an important role in aldosterone regulation.

A major concern with this manuscript is its disregard for the consensus in the adrenal field that the normal ZG layer (including in humans) essentially does not contain gap junctions (nicely reviewed in Bell CL and Murray SA (2016) Adrenocortical Gap Junctions and Their Functions. *Front. Endocrinol.* 7:82. doi: 10.3389/fendo.2016.00082). The evidence to support this conclusion is supported by immunostaining data, freeze fracture techniques, dye transfer studies and the fact that pharmacological inhibition of connexin/pannexin channels does not prevent calcium oscillations in the ZG. In contrast, gap junction proteins are abundant in the ZF layer, in the mesenchymal cells that surround ZG cells, in the connective tissues of the capsule and in the medulla.

Another major issue with this manuscript is its strong reliance on H295R cells, which are not a model of ZG cells. This adrenocortical carcinoma cell line expresses features of multiple adrenal cell types, including both ZG and ZF cells. Therefore, use of this cell type has the high likelihood to giving a mixed cellular phenotype, for example, when trying to draw conclusions about the role gap junctions play in normal aldosterone regulation.

Another major factor that raises doubt about the conclusions is the low quality of the immunostaining: (eg Fig 3C, Fig 4A), which does not allow the conclusions that are drawn. In addition, the CX43 staining data in Fig 5 is less than convincing. Fig 5 would also benefit from showing normal adrenals where CYP11B2 were expressed normally. One cannot conclude that a region of tissue is in fact ZG

when CYP11B2 is suppressed by an APA.

Finally, the role of the clock and aquaporins in aldosterone regulation remains unproven. Perhaps the patients had a mild rather than cyclical phenotype?

The manuscript would be strengthened by allowing the reader to draw his/her own conclusions. Thus, use of fewer suggestive phrases (such as: may be, it is likely, could represent, they are clearly present and appear an important target) would be welcome.

Minor point: The authors make reference to microarrays, do they mean RNA-Seq data sets?

Author Appeal

Dear Kyle

Many thanks for sending this manuscript out for review. Paradoxically, I thought its story was, if anything, richer than in our parallel manuscript which we are currently revising, but perhaps also more complex. Reading the three reviews I would have to agree that your decision was fair, given what you summarise as *'overlapping concerns about the strength of the novel conclusions that can be drawn with regards to the mechanism by which CADM1 mutations contribute to disease pathophysiology'*.

However, I am hoping that you will allow me to point out that the main shared concern, regarding the presence of gap junctions in the aldosterone-producing 'zona glomerulosa' (ZG) of the adrenal, reveals a fundamental misconception of the reviewers, which I suspect was central to your decision; and to suggest that overturning this misconception is reason for considering publication, rather than the reverse. Rather than the supposed absence of gap-junctions in ZG being a reason to reject our interpretation of CADM1's role, we believe the discovery of CADM1 interactions show that hitherto largely over-looked gap-junctions in ZG are important regulators of aldosterone secretion, and illustrate the value of even uncommon mutations in revealing new areas of biology.

The reviewers cite (in evidence of the absence of gap-junctions from ZG) the same review as ourselves (Bell & Murray, Front Endocrinol 2016); but we also cited several of the original papers, which are not accurately reflected by the review. There is considerable variation among species from the picture in rodents which dominates the review, with high ZG expression in some larger mammals; even in rodents there is documentation by EM of gap junctions in the aldosterone-producing zona glomerulosa, albeit smaller and sparser than in the adjacent cortisol-producing 'zona fasciculata' (ZF) (Palacios 1979).

Because we did not expect the very existence of gap-junctions in ZG to be contentious, we cited the supporting evidence as part of explaining (top of p8) how we found *GJA1*, encoding connexin-43, to be much more abundantly expressed in ZG than other gap-junction genes, using systematic analysis of these by laser-capture and microarray of 20 different adrenals. The obvious inference, from one connexin being more abundant than others, is that they are not all absent! Protein Atlas also shows clear ZG staining of *GJA1*/connexin-43 with two different antisera, and this was our experience in a number of adrenals before progressing to IF. We were unsurprised therefore at the positive ZG

immunofluorescence in the two examples we showed (Fig 5a). It seems from the reviewers' responses that we should have been more overt about our inferences, and should clearly have shown the blank no-primary controls for the IF.

I have taken a few weeks to write because I wanted to include some simple, graphic documentation that GJA1 is expressed in ZG, at both RNA and protein level. I am attaching two sets of annotated figures.

- One set shows, by RT-PCR and qPCR, that GJA1 is expressed at RNA level in ZG, at ~20% the level in ZF. The controls illustrate, firstly, the specificity of ZG sampling – we use cresyl violet to stain ZG and laser-capture microdissection of this; and, secondly, the expected high or low level of *GJA1* in other tissues, and expected high or low level of other genes (*LGR5* and *GSTA3*, respectively) that prove the ZG origin of the GJA1 signal.
- The other set shows in two adrenals that GJA1 is detected by immunohistochemistry in ZG, but consistent with the RNA data it is evidently less densely or diffusely expressed than in ZF. This is similar to the immunofluorescence data in the manuscript, but this time we include appropriate, and very clear, positive and negative controls.

We have the impression, finally, that some reviewers' reluctance to accept the existence of gap junctions in ZG added weight to the related – but we think flawed – concern that inhibition of gap junctions, by *gap27* or silencing, had less impact on aldosterone production than mutation of *CADM1*. We were explicit in the manuscript (top of p11) that we expected both N-terminal (extracellular) and C-terminal (intracellular) ends of *CADM1* to be affected by mutations in the membrane, and that effects on gap-junctions represented one, maybe even a minor, one of these effects. This expectation was the justification for exploring and reporting the interaction with clock genes, but we do not claim to have discovered everything that flows from the intramembranous mutations – which we pointed out are only the second, after those in *FGFR3*, to be discovered in the large family of single TM domain proteins. It would not, however, seem a valid reason to downplay the importance of gap junctions in ZG, or even the role of *CADM1* in their regulation, just because *CADM1*, or its mutation, has more than one action. The different potency between *gap27* and silencing, in their effects on aldosterone production, are readily attributed to *gap27* being somewhat more promiscuous than it says on the tin, whereas silencing was typically limited to ~70% knockdown.

I am hoping, Kyle, that you might please give us the opportunity after all to submit a revised version of the manuscript which includes – probably in supplementary – more extensive documentation of the

existence of gap junctions in ZG. I obviously do not know whether the confidential comments to Editor may have been more negative than the concentration on the specific points touched on here. But two of the reviewers seemed clearly positive, and the third was the one who was most 'narked' by our assumption of gap-junctions! We could include more data from primary ZG cells, in response to his/her other criticism. As you can imagine, the supply of fresh tissue from surgery has been very limited this year, and was a factor in choice of experiments.

Many thanks, and my apologies for adding to your in-tray.

Best wishes

Morris

Morris J Brown, MD, FRCP, FMedSci

Professor of Endocrine Hypertension

Queen Mary University of London,

William Harvey Heart Centre,

Charterhouse Square,

London EC1M 6BQ

Decision Letter, Appeal:

4th December 2020

Dear Morris,

Thank you for asking us to reconsider our decision on your manuscript "Somatic intramembranous mutations of CADM1 in aldosterone-producing adenomas, and gap junction dependent regulation of aldosterone production". I have discussed your appeal letter and supporting figures with my editorial colleagues, and we think that you have appropriately addressed concerns regarding the expression of gap junction proteins in the zona glomerulosa. We therefore invite you to revise your manuscript by incorporating these additional data and by responding to all reviewer comments with suitable clarifications and revisions, including additional experiments and analyses where appropriate.

When preparing a revision, please ensure that it fully complies with our editorial requirements for

format and style; details can be found in the Guide to Authors on our website (<http://www.nature.com/ng/>).

Please be sure that your manuscript is accompanied by a separate letter detailing the changes you have made and your response to all reviewer comments. At this stage we will need you to upload:

1) a copy of the manuscript in MS Word .docx format.

2) The Editorial Policy Checklist:

<https://www.nature.com/documents/nr-editorial-policy-checklist.pdf>

3) The Reporting Summary:

(Here you can read about the role of the Reporting Summary in reproducible science:

<https://www.nature.com/news/announcement-towards-greater-reproducibility-for-life-sciences-research-in-nature-1.22062>)

Please use the link below to be taken directly to the site and view and revise your manuscript:

[redacted]

With best wishes,
Kyle

Kyle Vogan, PhD
Senior Editor
Nature Genetics
<https://orcid.org/0000-0001-9565-9665>

Author Rebuttal to Initial comments

Somatic intramembranous mutations of *CADM1* in aldosterone-producing adenomas, and gap-junction dependent regulation of aldosterone production

Rebuttal

(reviewer comments in black, replies in blue)

Reviewer #1:

In this manuscript, Wu and colleagues describe the identification of novel heterozygous somatic mutations (p.Gly379Asp and p.Val380Asp) in the gene *CADM1* in 2/163 aldosterone-producing adenomas (1.2%). They demonstrate that expression of mutant *CADM1* has large effects on the expression of *CYP11B2* (aldosterone synthase) in the aldosterone-producing human H295R cell line (>10-fold increase compared to wildtype), with somewhat smaller effects on aldosterone production. The authors proceed to suggest that this observation is linked to an effect of mutant *CADM1* on gap junction formation. Lastly, their study includes RNA sequencing to identify underlying pathways, with a potential link to clock genes.

Even though *CADM1* mutations account for only a very small fraction of aldosterone-producing adenomas, this study is very interesting because the involved pathways appear to be distinct from mutations in the many previously described disease genes (typically ion channels and pumps). The effect on aldosterone synthase expression is impressive. A potential link to clock genes and cyclic variation in aldosterone production could also have clinical implications (need for repeated screening). The manuscript is original and significant, and the data are well presented. I have no concerns about the statistical analysis. My major concern is that despite extensive functional studies, the underlying pathophysiology remains somewhat unclear, in particular with regard to the link to gap junctions, as I will outline in more detail below.

We thank the reviewer for acknowledging the novelty and potential clinical implication. We think that the new data summarised in the rebuttal will please the reviewer, and that addressing the questions has helped to crystallize the links between *CADM1*, gap junctions and aldosterone production. Even if we had provided no further insight to the pathophysiology of *CADM1* mutation, or had found that gap junction inhibition was a minor contributor to activation of aldosterone production, the interest in uncommon mutations derives largely from their revelation of new physiological pathways, in this case the role of gap junctions in the physiological regulation of aldosterone. But, in the event, we submit that gap junction inhibition is also a major part of the pathophysiology, and that a plausible answer to the reviewer's question is now provided.

As we alluded to at a couple of points in the manuscript, commenting on the multiple previous names of *CADM1*, and several protein-interacting segments at the C-terminus, it is a fascinating, complex molecule (p6, lines 17-19; p12, lines 22-25). Since large increases in *CYP11B2* expression/activity is achieved by each of *CADM1* mutation and gap junction inhibition (new data in figure 5c), and *CADM1* mutation suppresses gap junction communication, it would be hard to say these are not connected, and the question is why silencing *CADM1* does not also activate *CYP11B2*. The 'obvious' answer is that the whole *CADM1* molecule is required for *CYP11B2* activation, and we suggest that the C-terminus of *CADM1* is a key player. Whether C-terminus actions are directly stimulated by mutation of the transmembrane domain, or are secondary to the gap junction inhibition, may remain open questions, although our new data (figure 6d) showing a large stimulation of *AQP2* expression by *Gap27* might point to gap junction inhibition being the driver. Reporting the effects of *CADM1* mutation on aquaporin-2 and

clock genes, without seeking fully to explain these, seems the right balance for a letter to the journal. (The initial papers on other somatic mutations, in *KCNJ5*, *ATP1A1*, *ATP2B3*, *CACNA1D*, were kind enough to leave interesting further questions to be answered by others!).

Detailed comments:

Major

1. Figure 4 (c) – Both *CADM1* mutants and sh*CADM1* appear to inhibit dye transfer, suggesting that both prevent gap junction formation. Because the effect on *CYP11B2* expression is very different (expression rises strongly when mutants are expressed (Fig. 3 (a)), but is suppressed when *CADM1* is silenced (Fig. 3 (b) iii)), these results suggest to me that gap junction formation or gap junction function may not be the major underlying pathophysiology of *CADM1* mutations.

We agree that the similar effect of mutation and silencing on gap junction communication, but opposite effects on *CYP11B2* expression, is a key and fascinating issue. As outlined above, we do not have all the answers, but have sought to clarify our account in the manuscript of the inferences from the observation (p12, final para). As the manuscript offers limited space to convey all our thoughts why the observation is not inherently contradictory, may we expand here the evidence and arguments to support this conclusion?

The observation on which the reviewer comments was a principal reason for our noting the complexity of *CADM1* (p6, lines 17-19; p12, lines 22-25) and progressing to the experiments which discovered its further effects, e.g. on clock genes and *AQP2*. As postulated in the manuscript, and as described with supporting structural (NMR) data for other single-pass membrane proteins, dimerization of *CADM1* in the membrane is a plausible or likely response to homophilic adhesion of *CADM1* ectodomains in adjacent cells, with downstream consequences from conformational changes at the C-terminus. Silencing of *CADM1* will suppress all its actions, including partnerships with the PDZ- and protein 4.1 binding regions at the C-terminus. By contrast, the C-terminus of mutant *CADM1* remains intact to mediate whatever signals arise from gap-junction inhibition, and may even be activated by the mutation (as shown for the intramembrane mutation of *FGFR3*). Whether or not gap-junctions are the sole pathway with which mutated *CADM1* interacts directly to stimulate aldosterone production, the shRNA experiment points to an essential downstream pathway through which the stimulus must work. The clock genes, which are mainly transcription factors, are among the possibilities, as are PDZ-domain containing proteins, such as *TJP1*. Our new illustration of this tight junction protein's highly selective ZG expression, quite different between APCC- and non-secretory ZG cells, appears to be an original observation (Figure 4C[i]). Of 4 APAs that we studied, only the *CADM1*-mutant APA showed diffuse positive staining for *TJP1* (Figure 4D).

An additional or alternative pathway which RNAseq suggests to be downstream of CADM1 is the PPARalpha target gene, ALDH3A2, encoding fatty aldehyde dehydrogenase. In the original manuscript, we commented that no genes changed expression so markedly, and in opposite directions, as CYP11B2 on mutation and silencing of CADM1. We commented on ALDH3A2 as the nearest exception, but its upregulation in mutant-transfected cells was <2-fold, and unlikely to be the direct mediator of the 10-30 fold increases in CYP11B2. However, adrenal cortex (along with skin) is the site of highest ALDH3A2 expression, and its ~2.5 fold suppression by silencing of CADM1 may be significant in view of the recent report of adrenal hypoplasia as part of the syndrome, mainly featuring ichthyosis, which is caused by a germline mutation of ALDH3A2 (p13 line 23-25).

2. Related to 1., the CX43 staining shown in Fig. 5 is extremely weak. In the absence of valid controls (e.g., secondary antibody only, peptide control), this could well be interpreted as background. This is of particular importance given prior data that there is little or no CX43 present in the ZG (Bell & Murray, *Front Endocrinol* 2016).

We probably committed a double sin in not questioning the ZG expression of CX43 either in general (although we too cited the Bell & Murray review), or the specific adrenals illustrated in Fig. 5. Because we have seen high expression of GJA1 (CX43) at RNA level in multiple adrenals (as was cited), and frequently on both IHC and IFC, we were confident of ZG expression, and did not include some of the many positive and negative controls that are in the revised manuscript (Figure 4b and Supplementary Figure S4b). We agree that documenting ZG expression beyond doubt is important to the thrust of this paper. The prompting to do this has actually led to some new observations which support the physiological role of gap junctions in aldosterone regulation in humans.

To address this important issue, we present substantial data at both RNA and protein level. On microarray, RNA from 21 laser-capture microdissected adrenals, adjacent to an APA or pheochromocytoma, is high, ~1/3rd of the even higher expression in ZF. This is similar to cited data from 4 adrenals removed from kidney donors (p9, 1st para). We have now extracted RNA from a further 3 adrenals in order to show qPCR data for each of ZG and ZF, together with results for the genes which our microarray showed to be the most selective for these zones, LGR5 and GSTA3, respectively, (Figure 4a). CYP11B2 itself is not of course usable as a ZG marker because in most adult adrenals its expression is confined to the aldosterone-producing cell clusters, which are not easily visible on LCM of fresh-frozen adrenal.

For demonstration of GJA1/CX43 protein, we have undertaken extensive IHC of a range of adrenals, semi-quantified by two independent blinded pathologists, and a limited amount of IFC (Figure 4b-c). The protein is clearly present, though variable between sites and often sparser (relative to ZF) than the qPCR ratios would predict. Of particular interest on the IFC are the annular gap junctions, which prove gap junction communication (Figure 4c[iii]). As the reviewer may know from the literature, and was illustrated to us by the time-lapse experiments in Figure 3d, and supplementary video, GJ proteins are

highly dynamic, probably more so when forming intact channels; and the annular gap junctions are often the best evidence of recent channel activity (astronomical sights that tell of past events are maybe a useful if inexact analogy). A bonus of this extra work is the emerging evidence that the reviewer is probably correct that GJs are truly sparse in the aldosterone-secreting cells of aldosterone-producing cell clusters.

3. In addition, what is labeled as ZG in Fig. 5 (a) in the adrenal from the patient with *CADM1* mutation does not appear to be ZG based on morphology (streaks of cells as in fasciculata). Given that *CYP11B2* is negative (were these patients not treated with Spironolactone to raise renin to nonsuppressed values?), it is unclear how the ZG was identified.

As the reviewer comments, visualising ZG cells can be a challenge when their hallmark product, *CYP11B2*, is suppressed. It is indeed striking that, APCCs apart, the suppression persists when renin is de-suppressed. Digressing somewhat from the present manuscript, we have shown that suppression could be due in part to molecules which we found (on the microarray showing *GJA1/CX43* expression in ZG) to be selectively expressed in human ZG (*LGR5*, *DACH1*, *ANO4*, *NEFM*), whereas they are absent from the list of genes upregulated in rat adrenal ZG (Nishimoto, *Endocrinology* 2012). Their expression on IHC helped us to acquire confidence in recognising compact ZG cells, with their dentate nuclei. However, antisera from those studies are no longer available, and for the present revision, we have moved the original figure to supplementary, and replaced it with a set of new studies with a number of concordant stains in serial IHC sections, to delineate ZG. These include *CYP17A1*, which is dense in ZF and absent in ZG cells, *KCNJ5*, *CADM1* itself, and the newest discovery, *TJP1*.

Some of these antibodies show a narrow intermediate zone of lower staining between ZF and ZG, and *GJA1* staining also appears intermediate in this zone (e.g. Figure 4b) This has suggested to us that the role of GJs in adrenal cortex might have arisen more to influence change of phenotype during centripetal migration of ZG cells, than to regulate aldosterone production within ZG.

4. A major claim – that mutant *CADM1* pushes apposed cells beyond reach of their connexons – is not supported by data and thus remains speculative.

We have acknowledged that the structural data derives from modelling (albeit two independent modellers, at Imperial College, London, and Kobe University, Japan), but is consistent with NMR studies one of the few previous examples of an intramembranous mutation introducing polarity into the TM domain of a single-pass membrane protein, namely the achondroplasia mutation of *FGFR3*.

5. Fig. 5 (b): If treatment with Gap27 has a large effect on CYP11B2 expression, but silencing of CX43 does not, then this suggests that the effect of Gap27 treatment is not mediated through inhibition of CX43. The authors demonstrate that even inhibiting CX43 and CX45 together has a much smaller effect on CYP11B2 expression than Gap27 treatment. This adds to my doubts about gap junctions as major part of the pathophysiology of CADM1 mutations.

We may disagree somewhat on the inferences here, but agree on the need for some clarification (p11, line 12-13). The two modes of inhibition/blockade are complementary. The silencing is probably specific for its target, and will tend to underestimate the contribution of GJ communication both because of incomplete silencing, as assessed by qPCR and Western, and because complete specificity leaves intact the (?compensatory) contributions from other gap junction proteins with minor expression (e.g. GJB2 and GJC2 – see new supplementary tables S2). Gap27 has not been formally tested against all gap junctions, and there is only 1 amino acid residue different, for instance, between CX43 and CX45 at the site of Gap27 binding. Given that the difference in efficacy, in different experiments, between Gap27 and silencing is <2-fold (compare fig. 5a with 5b[iii]) the factors discussed seem more likely explanations of the difference, than off-target effects of Gap27, which has been used in >100 published studies in the last 25 years.

6. I would suggest showing in a Suppl. Table mutant and WT reads for the CADM1 mutation in tumor and normal tissue. I would also suggest displaying all other somatic mutations discovered. This is relevant because the absence or presence of any additional somatic mutations that may account for aldosteronism or growth in the tumor would help to assess the role of CADM1 mutations.

We have added a table (S1b) to Supplementary Table 1, containing the reads for CADM1 and the only two other genes with somatic variants with alternative allele frequency at least 10% of total. This table also now includes chromosomal co-ordinates, which were previously omitted.

7. The fraction of tumors without mutations in any known disease genes (20%) is quite large compared with recent papers from the Rainey group cited in the study. I would suggest showing the somatic mutations discovered in these tumors – were any discovered?

Review papers tend to suggest a value <80%, and the exceptions have come from studies where DNA is taken from CYP11B2 positive areas of the APA. However, quoted percentages – both for any mutation, and for each individual mutation – vary with the ascertainment criteria for the cohort in question. We consider that our use of routine plasma renin measurement in all patients with resistant hypertension, and of our 11C-metomidate PET CT, have allowed us to detect a higher proportion of what, in our 2013 letter in this journal, we described as ZG-like APAs. Whilst a very high proportion, probably >90%, of ZF-like APAs have KCNJ5 mutations, the ZG-like APAs have more heterogeneous genotype, and greater likelihood of no known mutations being found. There is a certain irony in our seeming less good at

finding mutations than other groups, when at the same time we are reporting the first, or almost first, completely novel somatic mutation in APAs since 2013! (An exception is the recent report of two APAs with somatic CLCN2 mutation, which followed the two reports in this journal that CLCN2 is the germline mutation causing FH2.)

Minor

1. Could the authors comment on why the effect on aldosterone synthase expression is so much larger than the effect on aldosterone production?

We have added a short comment (p.15, line 5-8). Our speculation is that this is linked to the possibility, suggested by the high expression of AQP2, and variable plasma aldosterone in the index case, that at very high aldosterone production rates aldosterone is not immediately secreted, but stored in secretion granules whose release is triggered or regulated.

2. Fig. 4 (a): The claim that silencing of CADM1 increases CX43 expression should be supported by quantitative data. Showing single images with low CX43 expression and a single Western blot appears insufficient.

We agree, and have deleted the Western blot and the claim, which is not an essential component to the implication that gap junction communication is regulated by CADM1 and in turn controls aldosterone production.

3. Figure 1 (a) CACNA1ID – this should be CACNA1D (1, not I)

Thank you for drawing our attention to this error, which has been corrected, to our attention

4. Figure 1 (b): Renin mass – this does not appear to be the correct description of what is shown (renin in mU/l, and then plasma renin activity for the lowest value). Looking at aldosterone and renin values, it is striking that the postoperative ARR is similar to the average of the 4 prior measurements, with renin still being borderline suppressed (taking into account the different units used). This is described by the authors as “cyclical aldosteronism”. Could the authors please comment on the time of day of these measurements? In most centers, ARR is determined in the morning, which should limit the effect of clock genes.

The data reported here is as it was reported to us by the measuring laboratory – the post-op result is clarified as being two years post-op, when the senior investigator had moved from a centre using renin mass to a centre using renin activity. We apologise for an error in quoting the normal range for ARR measured by the latter laboratory, which should be <1000.

5. Figure 1 (c): Only part of the APA is positive for CYP11B2 in IHC – is this an artifact? If considered real, please explain. Otherwise, the staining should be repeated.

This is an artefact, and has been repeated (Figure 1c[iii]).

6. Fig. S1 (a): It is unclear to me what the numerous bands represent, at least the authors should comment on this or describe the blot appropriately.

One of the bands is likely to be a product of α -secretase, and the other is unknown. We have added labels.

7. Legend Fig. 4 (d): Hoechst, not Hoescht

Thank you for pointing out the error, which has been corrected.

8. Suppl. Fig. S7: Gene names cannot be clearly read.

We apologise, and are uncertain why the same program generated different fonts for the two heatmaps. Those in Supplementary Figure S7 have been reprinted.

Reviewer #2:

The paper by Xilin Wu and colleagues reports on the identification of somatic mutations of the transmembrane of *CADM1* in 2 small aldosterone producing adenomas (APA) that both were causing a moderate and variable level of aldosterone excess.

Functional expression of the *CADM1* mutants in adrenocortical cells was able to stimulate aldosterone synthase gene expression. The authors then demonstrate that this could be caused by longer distances between cells leading to dysregulation of Gap Junction by *CADM1* mutants.

This is the first report of *CADM1* mutations in APA. Although these mutations appears rare the elegant and various functional studies reported clearly demonstrate the role of *CADM1* and GJ in aldosterone regulation.

One interesting speculation is the phenotype of these 2 patients with intermittent/variable or cyclical secretion of aldosterone. This raise the hypothesis that diagnosis of PA would be often missed in patients with *CADM1* mutation. However there was a clear improvement of hypertension in these two

patients. This is clearly a fascinating hypothesis because if this was the case one would speculate that moderate variable form of PA would need to be investigated more carefully to search for such patients. However it is at present quite speculative since it is difficult to establish a genotype/phenotype correlation on only 2 patients. It would be interesting to investigate more cases to build this hypothesis on higher numbers.

We thank the reviewer for these nice comments. We agree of course that the hypothesized connection between CADM1 mutation and 'variable PA' will remain hypothesis until more cases are identified. As detailed a little more below, we have already sought to do this, but hopefully it will be the combination of this report, if accepted, and the recent Annals paper (Brown et al – no relative! - Ann Intern Med. 2020;173:10-20) which will catalyse a serious, sufficiently large search for the link.

Specific comments:

1- It is disappointing that exome identify in 2 different non selected cohorts a somatic mutation in one tumor in each cohort while sanger sequencing of APA selected to be wild type for the known gene fails to identify additional cases. It seems according to the Sanger sequences shown on fig 2 that the mutant allele might be present in less than half of the cells. Could it be possible that the Sanger sequencing screening of the 53 wild type APA would have been less sensitive than ngs? In this case targeted ngs for CADM1 might have identified additional APA with CADM1 mutation. An other option would be to sequence a larger number of sample.

We share the disappointment! There is little doubt that the mutations of CADM1 are uncommon in APAs from conventionally diagnosed PA. Our index case was the only case in the 40 APAs of the discovery set, and the second case was one of 70 studied by NGS. The peak height for the minor allele of somatic mutations is characteristically less, sometimes much less, than that of the wild-type allele. Whilst this could reflect heterogeneity among tumour cells, we have always assumed the main factor is dilution of 'true tumour' cells by other cell-types, e.g. vascular and connective tissue; this seems to be borne out by generally finding higher peak height for the mutant allele when we look at cDNA (which for most of the APA mutant genes is more abundant in the aldosterone producing than other cell types) than gDNA.

2- Could CADM1 be mutated in Aldosterone Producing Cells clusters (APCCs)? This would be an other way to clearly establish the significance and importance of CADM1 mutations in PA.

We absolutely share this speculation with the reviewer. No-one has yet undertaken whole-exome sequencing of APCCs. Given the considerable labour required to accumulate sufficient DNA, we think we should wait till we have enough samples for this, rather than concentrate on CADM1. Our impression from the published microarray data for APCCs, from Rainey's group, is probably against CADM1 mutation being common, as there is not much overlap between upregulated genes in APCCs and those we found in the index case (e.g. AQP2) or the transfected cell experiment.

3- Pathology, immunohistochemistry and clinical outcome are clear proof of APA in the 2 patients. However the pre-operative investigation are puzzling because PA is not marked nor constant and that adrenal imaging are ambiguous as often with very small APA. Were a suppression test and an adrenal venous sampling done before surgery ?

The reviewer's observations are interesting, and to a degree bear out our response to the final major point of reviewer 1. For us, this size of APA is the norm rather than exception, but in routine clinical practice such a lesion on CT is often overlooked or dismissed. Adrenal vein sampling was strongly positive (ratio of >7:1), but no suppression test was conducted.

4- The concept of illegitimate membrane receptor have been previously suggested in PA. Could it be a mechanism for the variable level of aldosterone secretion in these patients ? Does the RNAseq data show upregulation of potential illegitimate receptor ?

The RNAseq of the index case and of transfected cells show a number of interesting genes, as discussed, but not the illegitimate receptors previously linked to APAs. We hypothesize that variable secretion may, rather, be related either to development of secretory granules (which are only intermittently discharge), or to exaggeration of biological rhythms, or to expression of adrenomedullary genes (e.g. SCG2, VGF), though these were upregulated only in the transfected H295R cells, not the index tumour.

5- The RNAseq experiments give interesting insights. It is unclear how the statistical analysis was done to rule out false discovery considering the numbers of analyzed genes ?

Because of the large n number, resulting from testing multiple mutants and isoforms, with highly consistent results (as illustrated in the heatmap, Fig 6a), pooled results for up- and down-regulated genes were of very high significance, and we showed only genes in the table (S2) with fold-change >2.5 fold, at p values between 10^{-7} and 10^{-20} . FDR is then shown for the further analyses (Supplementary Table S4) of pathways, with Biological Rhythms having an FDR of 0.0014.

6- It would be interesting to see how frequent is this situation to use the set of genes identified by RNAseq to search in already available APA transcriptome others tumors with the same phenotype.

We had the same idea! Two of the replication sets were deliberately requested from co-authors in Japan who had previously published microarray data showing elevated expression of AQP2 in several of their APAs. We did not find any CADM1 mutations in these samples, and our repeat qPCR found lower values than our index case.

7- It seems that very few APA were included in the APQ2 IHC analysis. It would be interesting to see in a large cohort how frequent is the loss of APQ2 and this could be also a way to identify others APA with CADM1 mutation.

We agree again. We did originally hope/plan that the APAs from which we received cDNA for sequencing would be studied by their host labs, but last year has not of course been a good time for such additional studies. As seen from our adrenal and control kidney stains, the antibody is a nice one, and maybe after this report a number of labs may like to add it to their routine. If CADM1 mutations are rare, high AQP2 expression may be a better way of detecting APAs in which the underlying biology of cyclical aldosteronism can be investigated.

Reviewer #3:

This manuscript by Brown and colleagues explores the putative role of two (adjacent) missense mutations in the transmembrane domain of CADM1 in patients diagnosed with aldosterone-producing adenomas, a common and potentially curable cause of hypertension. The introduction of these mutations into the H295R adrenocortical cell line resulted in increased expression of CYP11B2, essential for aldosterone production, by 15-30 fold. In contrast, knock down with shRNA, or introduction of soluble CADM1 Ig-ectodomains abolished dye-transfer through gap junctions (GJs). This led to protein studies and molecular modelling which predicted increased exit angles of mutant (GOF) CADM1 with the resulting longer distance between cells preventing GJ formation. Additional in vitro dye transfer studies with H295R cells and RNAseq analysis following mutant transduction and AngII stimulation led to the conclusion that GJ communication plays an important role in aldosterone regulation.

A major concern with this manuscript is its disregard for the consensus in the adrenal field that the normal ZG layer (including in humans) essentially does not contain gap junctions (nicely reviewed in Bell CL and Murray SA (2016) Adrenocortical Gap Junctions and Their Functions. *Front. Endocrinol.* 7:82. doi: 10.3389/fendo.2016.00082). The evidence to support this conclusion is supported by immunostaining data, freeze fracture techniques, dye transfer studies and the fact that pharmacological inhibition of connexin/pannexin channels does not prevent calcium oscillations in the ZG. In contrast, gap junction proteins are abundant in the ZF layer, in the mesenchymal cells that surround ZG cells, in the connective tissues of the capsule and in the medulla.

We should thank the reviewer for this challenge, and the new results to which the challenge has led. As discussed earlier, in response to reviewer 1, we understand that Bell and Murray's review (which we also cited) may convey the impression outlined by reviewer 3. However, for the reasons stated earlier – microarray data from 21 laser-capture microdissected adrenals, and immunohistochemistry of many adrenals from UK, Czech and Malaysian cohorts – we looked critically at the review and did not find convincing endorsement of the view that gap junctions are absent from human ZG. So, although facts need not be wrong when they contradict a consensus (it could be the consensus which is wrong!), we did not think as seriously as perhaps we should have that we needed to issue this challenge. No human data is shown in Bell and Murray, and their figures showing ZG are taken from a reference (30 in the review) whose abstract starts “Mouse and monkey adrenal glands were used to study the relationships between gap junction protein expression ... and adrenal zonation”. Even so, the review does not categorically dismiss ZG communication, saying “lucifer yellow dye communication between cells was more abundant in the inner zones of the adrenal cortex” and “little or no Cx43 gap junction protein was detected between adrenal cells in the zona glomerulosa”. There clearly are species differences because in her 1997 paper (*MICROSCOPY RESEARCH AND TECHNIQUE* 36:510–519) Sandra Murray reported that ‘in the bovine adrenal, ZG cells were connected by large gap junction.’ But it was in the rat that Palacios demonstrated by EM clear pictures of gap junctions in ZG, emphasizing however that they are smaller and rarer, and age-dependent. (*J. Anat.*, 129, 695-701, 1979)

Because the reviewer was obviously correct that the presence of gap junctions is critical to the manuscript's conclusions, and we could not prove the identity of ZG cells in the original Figure 5, we have undertaken extensive further work, including qPCR of freshly dissected laser-capture material, and immunohistochemistry with peroxidase or fluorescent stains, to document ZG expression of GJA1 (connexin 43). Almost all of Figure 4 and Supplementary Figure S4, and the text reporting these on p10-11, are new except for a graphical representation of the microarray data to which the original manuscript referred (p9 first para). As described in 1979 by Palacios, the gap junctions in ZG are clearly sparser than in ZF. An interesting consequence of the new work is the finding, still tentative, that gap junction density is inversely related to that of CYP11B2, and that the residual aldosterone-producing cells of human ZG are similar, as regards gap-junction expression, to the ZG of the species discussed by Bell and Murray.

Another major issue with this manuscript is its strong reliance on H295R cells, which are not a model of ZG cells. This adrenocortical carcinoma cell line expresses features of multiple adrenal cell types, including both ZG and ZF cells. Therefore, use of this cell type has the high likelihood to giving a mixed cellular phenotype, for example, when trying to draw conclusions about the role gap junctions play in normal aldosterone regulation.

We agree that this was a limitation of the data presented, and have now included striking evidence of tonic inhibition by gap junctions of aldosterone production from primary adrenocortical cells, cultured from adrenal glands removed at surgery (Figure 5c and p11, lines 20-25). Although ZG cells are the minority of cells in these primary cell cultures, they are the only source of aldosterone production. We deliberately grew these cultures at higher cell density than our usual, in order to maximise the chance of gap junctions forming, and the variation in CYP11B2 increase appeared related to cell density. The largest increase in CYP11B2 was now higher than seen in the transfection experiments with CADM1 mutants, and similar to the typical difference seen between APAs and adjacent adrenal on RNAseq.

Another major factor that raises doubt about the conclusions is the low quality of the immunostaining: (eg Fig 3C, Fig 4A), which does not allow the conclusions that are drawn. In addition, the CX43 staining data in Fig 5 is less than convincing. Fig 5 would also benefit from showing normal adrenals where CYP11B2 were expressed normally. One cannot conclude that a region of tissue is in fact ZG when CYP11B2 is suppressed by an APA.

As discussed earlier, and in our discussion with reviewer 1, we agree that the figure did not completely meet the challenges posed by both reviewers, and have replaced it with a selection of adrenals, and stains of these (Figure 4b and Supplementary Figure S4). The new images use absence of CYP17A1, in the zone external to ZF, as the objective means of defining ZG, along with dense staining of KCNJ5, and –

an apparently new observation – dense membrane staining of TJP1. We will continue to evaluate antisera for our ZG-selective genes, and use confocal imaging to seek GJA1/CX43 expression in the same cells as DACH1, NPNT or LGR5. In normal times we might have been able to include such evidence with the revision. But limited access to the various facilities, and need to test several new antisera, means that such work cannot be scheduled at present. We trust that the reviewer agrees that the question whether gap junctions are in ZG is now adequately answered, and that further work is outside the scope and space of this letter.

Finally, the role of the clock and aquaporins in aldosterone regulation remains unproven. Perhaps the patients had a mild rather than cyclical phenotype?

We agree that we have left plenty of further work to be done, as has fortunately been true of most papers which first reported the somatic mutations in APAs! As discussed with other reviewers, we have concentrated on interactions between CADM1 and GJA1/CX43, whilst also seeking and following clues to other pathways which are activated. These could be either a downstream consequence of gap-junction inhibition, or follow conformational changes at the C-terminus of CADM1 (as predicted from the analogous intramembranous mutations of another single TM domain protein, FGFR3).

The index case had severe hypertension, and his variable PA was only detected because of his participation in our 'PATHWAY-TWO' study of resistant hypertension (Lancet 2015). The interesting speculation is that both patients' complete clinical cure by adrenalectomy was due to their shorter total exposure to aldosterone + salt excess causing secondary vascular changes than typically occurs before diagnosis is finally made in other patients with PA.

The manuscript would be strengthened by allowing the reader to draw his/her own conclusions. Thus, use of fewer suggestive phrases (such as: may be, it is likely, could represent, they are clearly present and appear an important target) would be welcome.

We have taken this advice on board during revision. Having asked us, maybe, to speculate in the previous question to us, the reviewer probably sympathises with the temptation to include such phrases, but we do understand the point being made.

Minor point: The authors make reference to microarrays, do they mean RNA-Seq data sets?

This is hopefully clearer in the revision. The original data in the manuscript is RNAseq, but we also now show previously unpublished data from the microarray (Figure 4a[ii]) that importantly led us to recognise the abundance, at least at RNA level, of GJA1 in human ZG.

Decision Letter, first revision:

11th March 2021

Dear Morris,

Your resubmission entitled "Somatic intramembranous mutations of CADM1 in aldosterone-producing adenomas, and gap junction dependent regulation of aldosterone production" has been seen by the original referees, whose comments are copied below. Based on their feedback, we have decided that we cannot offer to publish your study in Nature Genetics.

In particular, while Reviewer #2 is satisfied with the revision, Reviewers #1 and #3 have ongoing concerns regarding the experimental evidence supporting the study's claims. We are persuaded that these ongoing concerns are sufficiently important as to preclude publication of this study in Nature Genetics.

We understand your disappointment with our decision. Unfortunately, the number of papers with possible claims on space in Nature Genetics vastly exceeds the number that we can publish, and we are therefore frequently forced to make difficult decisions. We are confident that you will have no difficulty in obtaining publication elsewhere and we wish you every success in so doing.

Sincerely,
Kyle

Kyle Vogan, PhD
Senior Editor
Nature Genetics
<https://orcid.org/0000-0001-9565-9665>

Referee expertise:

Referee #1: Genetics, hypertension, primary aldosteronism

Referee #2: Genetics, adrenal tumors

Referee #3: Organogenesis, adrenal development

Reviewers' Comments:

Reviewer #1:
Remarks to the Author:

The rebuttal is difficult to read because line numbers given in the rebuttal do not correspond to those

in the manuscript. I am unable to find changes in the manuscript that the rebuttal refers to (e.g., “p12, final para” shows no marked changes, and I do not see an explanation of the divergence between effects of silencing on CYP11B2 and gap junction formation there).

Regarding the RNA sequencing data, p. 5, l. 170 mentions GSTA3 as a marker. The corresponding Figure indicates GSTA1 was studied – unclear what was done. In any case, both would be uncommon marker for ZF. Valid ZG and ZF markers would be CYP11B2 and CYP11B1, which could also demonstrate how well the laser capture procedure (contamination will occur) worked. To clarify the pathophysiology of increased CYP11B2 expression, demonstration of weak expression in so-called glomerulosa cells that do not express CYP11B2 does not seem particularly useful. Fig. 4Ciii: absence of CX43 in CYP11B2-positive cells – how is CX43 expected to have the proposed effect on aldosterone production if it is not present in the cells that express aldosterone synthase? Normal adrenals of young individuals rather than tumor-adjacent material might be helpful.

Fig. 4b: There seem to be CYP17A1-positive cells in what is apparently assumed to be ZG. KCNJ5 is not an ideal ZG marker because its expression extends into the ZF.

The legend claims CX43 is found in mutant APAs. It is unclear where an APA is shown in the figure (label?).

Supplementary Fig. S4a is unlabeled – it is unclear what is shown. Is this an adrenal gland with a tumor?

Regarding the modeling (Supplementary Fig. S1 (d)): The lower panel is confusing regarding the scale. For example, the length of the WT TM domain should be 33 Å according to the upper panel. In the lower panel, the height is depicted as 33 Å, which cannot be correct if the model is depicted on scale. Similarly, whereas in the upper panel, distances appear to be scaled, that appears not to be the case in the lower panel, with 33 Å, 31.74 Å and 31.42 Å all depicted as the same height. More importantly, the TM is shorter in mutants than in the WT, which acts contrary to the tilt with regard to the point of cleavage. That is not discussed in the main text.

My suggestion to show the somatic mutations discovered in tumors without mutations in known genes has not been addressed.

Similarly, my question about the time of day of the aldosterone and renin measurements was not answered, and why this is considered cyclic remains unclear. Fig. 1(b)ii shows three normal ARR and one moderately elevated ARR pre-op.

It remains unclear to me what the many bands in Supplementary Fig. S1 represent. Showing lysates of untransduced cells would be standard.

Upon re-reading the manuscript, I note that in Fig. 1(b)i, the variant G379D is not correctly displayed. If V380D results in the sequence AVIGGDVAV..., G379D should result in the sequence AVIGDVV... rather than AVIGGDV... as given in the figure.

On p. 6, ll. 65-67, the authors claim that the presence of the mutation in cDNA is proof that the consequences of the mutation are due to expression of an abnormal protein. That is formally not correct because they study cDNA rather than protein.

Minor:

Supplementary Fig. S1 (d), legend: "helix with the lipid bilayer was increased"

Fig. 4a, legend: "fasciculate" -> fasciculata

Suppl. Fig. S4d legend: chi shows as "?"; 2 independent histologists (not histologist) (same in Fig. 4(c) legend), p. 47 two histopathologists (not histopathologist)

Fig. 4(d): what does the "+" symbol mean?

Methods, p. 47: reference to "Supplementary Figure SX"- that Supplementary Figure does not exist.

I. 201: "The effects were dramatic" – suggest to delete. I. 203: 20-200-fold: That does not seem to be correct. AngII + Gap27 should be compared to AngII, which probably corresponds to less than 100-fold elevation.

L. 230-232: unclear

Reviewer #2:

Remarks to the Author:

The answers to my previous comments are acceptable.

Reviewer #3:

Remarks to the Author:

The phenotypic data presented in the original manuscript appears to be contradictory with the dye transfer and GAP27 inhibition studies. That is, the results from the CADM1 overexpression and knockdown studies seem to differ from the rest of the story. The earlier studies seem to suggest that GJ communication leads to enhanced aldosterone production (with the exception of overexpression of WT CADM1, which has no effect at all), whereas the dye transfer and GAP27 peptide studies argue that less GJ communication promotes aldosterone production.

I remain unconvinced that the ZG contains gap junctions, a central premise of this manuscript. While the LCM (Fig 4a) appear to show that CX43 is expressed in the ZG (albeit at lower levels than in the ZF), but is LGR5 the best ZG marker to be using here? Should also use well established ZG marker genes such as DAB2 and Gq? Unfortunately, the CX43 immunostaining data remains unconvincing and may represent background staining--it is certainly not marking gap junctions (e.g., Fig 4b, 4c, 4d). While the control staining maybe somewhat helpful at addressing the background issue, its presence in the cytoplasm or as puncta rather than at the membrane (where gap junctions should reside and do appear to within in ZF regions) undermines the authors primary hypothesis and conclusion.

The additional GAP27 data are interesting but appear to raise important new questions and are not fully convincing.

Regarding the experiments using H295R cells and Gap27 the following concerns or questions arise:

One cannot use two T-tests to assess significance--one must use ANOVA. In this case, the effect of Gap27 on basal expression would likely be eliminated.

It appears that N=8 (for example in Fig 5a) likely represents technical replicates. Whatever the number is should indicate the number of independent studies not the number of replicates from a few studies.

The methods were not entirely clear. How long was the Gap27 incubation? Was it enough time to change expression of Cyp11b2 and other ZG markers (such as DAB2 and Gq)?

If correct, these data seem to suggest that GAP27, or knockdown of CX43, enhance the zG phenotype, consistent with the notion that the lack of GJ communication makes H295R cells more zG-like and less zF-like. It would be important to point out in the discussion that such cells can be pushed in either direction. Such a result could be consistent with the possibility that ZG cells do not utilize GJs at all for communication while ZF cells do.

Regarding the experiments measuring aldosterone secretion from human primary cultures:

There is a concern that the primary cells do not appear to be healthy since only 1 adrenal showed Ang II stimulated aldosterone production and in that adrenal Gap27 had no statistical effect. Curiously in the other adrenals, which did not show Ang II stimulated aldosterone production, an effect of Gap27 was observed.

Regarding the following statement: "A feature of the experiments with CADM1 mutation and GJ inhibition has been the greater increase in CYP11B2 than aldosterone. A speculative explanation which connects to the AQP2 findings is that, if secretory granules are formed, aldosterone secretion may be more episodic or regulated than in other causes of increased aldosterone release." Are the authors speculating that formation of secretory vesicles is related to aldosterone release? As far as I am aware, no steroid hormone is released in vesicles. Such speculation should be beyond the scope of this manuscript without the appropriate data to support the claim.

Finally, I have questions and concerns related to the experiments measuring calcium flux in H295R cells.

The authors need to show representative traces for Ang II-induced Ca changes with and without Gap27. If the cells are truly oscillating, then Ang II should produce a large transient rise in Ca, which then returns close to baseline. If there is only a large rise that does not return the cell is dead.

Regarding the statistical analysis, I do not believe that a T-test is appropriate for comparing intensities. In addition, a Fisher exact test is more commonly used to compare 2 x 2 contingency tables. Is this an appropriate method here? Might consider using a non-parametric ANOVA.

Author Appeal

Dear Kyle

NG-LE55401R1 - Somatic intramembranous mutations of *CADM1* in aldosterone-producing adenomas, and gap junction dependent regulation of aldosterone production

We should like please to appeal the decision of 11th March. The appeal is underpinned by our discovery of three further cases with exactly the same mutations, and the attached, detailed critique of Professor Dr Sandra Murray. In reviews of our initial manuscript, she was rightly regarded by both critical reviewers, and by ourselves, as the doyenne of adrenal gap junctions. Further work in our revision was largely prompted by the reviewers' suggestions how to meet her published criteria for demonstration of functional gap junctions. We therefore approached Dr Murray for independent advice and a critical look at our manuscript. Dr Murray's opinion (please see her final paragraph for a summary) is that we were right to infer presence of functional gap junctions in human adrenal zona glomerulosa (ZG). This conclusion is further supported by fresh confocal imaging which we have undertaken since submission of the revised manuscript; this new work includes a specific ZG marker (DAB2) requested by one of the reviewers.

Further mutations

You kindly let me know in follow-up correspondence that you were influenced in part by the small number of cases in our report, limited to the index case with the Val380Asp mutation, and second case with mutation of the adjacent residue, Gly379Asp. We have therefore approached further investigators with cohorts of aldosterone-producing adenomas, and this has led to the discovery of three further patients, with somatic mutation of either Val380 or Gly379, once again to Asp. All were in the same cohort (that of our Paris colleagues who co-authored our recent GNA11/Q mutation paper). This brings to five the total number of cases.

As was argued in the manuscript, the *CADM1* mutation is likely to be under-recognised as a consequence of the mutation causing exaggerated diurnal variation in aldosterone production. Our inference was drawn from the limited clinical histories, and the robust transfection data (accepted by all reviewers) showing biological rhythms to be the main pathway altered by both *CADM1* mutations. A recent study from Harvard indeed reported a ~2-fold higher prevalence of primary aldosteronism if 24h urine aldosterone, rather than spot blood measurements, are used. The Harvard authors, independently from ourselves, inferred the likely existence of patients with exaggerated diurnal variation.

Gap junctions in adrenal zona glomerulosa (ZG)

Regarding the reviewers' criticisms, the main objection to our revision remains the view of two reviewers that gap-junctions are not present in the aldosterone-producing zone of the adrenal cortex, ZG. Our view remains that reviewers are putting pre-conceived opinion ahead of our new experimental data, instead of allowing that the previous opinion based on a limited number of nonhuman adrenals is

over-turned by our RNA, protein and pharmacological data from multiple human adrenal glands. Dr Murray's opinion, as you will see, arises from detailed consideration of our manuscript, fresh data as I will summarise here, the two reviewers' comments, and review of both her own and others' published data. I attach a few representative images from the manuscript which seemed to particularly impress Dr Murray (re-numbered in the figure-references below), together with new confocal imaging which confirms co-localisation of GJA1 and ZG-cell markers.

Dr Murray agrees that transfection of *CADM1* mutations into adrenocortical cells inhibit gap junction communication, as shown both by our dye-transfer experiment (**Fig 1a**) and by reduction or loss of gap junction plaques and annular gap junctions. Dr Murray and others have shown the latter to be the distinctive and tell-tale sign of recent gap junction communication, incorporating gap junction protein from both communicating cells (**Fig 1b**) – a cellular supernova, if you like, which tells of a now-defunct star. Dr Murray agrees that inhibition of gap junction communication either by silencing RNA (**Fig 2a**) or gap27 (**Fig 2b**) increases aldosterone production. And, most importantly, she is certain that gap junctions formed by GJA1 (also known as Cx43) are present in ZG. Key images from the manuscript and previous appeal letter illustrating presence of GJA1 RNA and protein in ZG are shown in **Fig 3**. Additional serial or confocal images shared with Dr Murray are shown in **Fig 4** and **Fig 5**, which should remove any doubt whether gap junction proteins are present in ZG cells, or functional on the plasma membranes (as seen from budding of annular gap junctions). Dr Murray is not concerned by the evidently smaller and scarcer membranous gap junctions in ZG compared to ZF, and their apparent distribution within cytosol of ZG cells. Indeed she draws attention to relevant literature on why gap junction size differs among tissues.

Reviewer 1 queried why gap junctions are rarely seen in cells positive for CYP11B2 (aldosterone synthase), if present in ZG. As was explained in our rebuttal, the paucity of gap junctions in aldosterone-secreting cells is actually just what is predicted by gap junction's *inhibition* of aldosterone secretion. In non-human adrenal, where CYP11B2 is expressed throughout ZG, and in the occasional CYP11B2-positive 'aldosterone-producing cell clusters (APCC)' of human adrenal, we do not expect to find many gap junctions, and that is precisely what is observed. But as has been clear in every immunohistochemical study of human adrenal since development of selective CYP11B2 antisera in 2011, and Gomez-Sanchez' monoclonal in 2014, most ZG cells in adult human adrenal have switched off aldosterone production. This is assumed due to high salt intake, obviating need for aldosterone-induced salt retention. In **Fig 4**, the reciprocal relation between high CYP11B2 (APCC) and low GJA1 expression is apparent. Whether gap junctions are, indeed, the major mechanism for the suppression of CYP11B2 in human adrenal is clearly an interesting question – beyond our present scope, and maybe awaiting discovery of somatic loss-of-function GJA1 mutations to address the question *in vivo*.

Reviewer 3 appreciates that most human ZG fails to express CYP11B2, but has a different view to us about choice of alternative markers. Although we believe that we have a strong track record in this area,

we have now undertaken one of his suggested stains, DAB2, from which it is clear that GJA1 is expressed in the DAB2-positive ZG cells (white boxes in **Fig 4b** and **4e**).

The best commercially available antisera for GJA1 and DAB2 are both rabbit, so we have used serial sections rather than confocal imaging. Nevertheless, confocal evidence that GJA1 is expressed in subcapsular non-ZF cells (as predicted by the absence of the green ZF marker CYP17A1) is apparent in **Fig 4f**. We could pursue antibody conjugation, if the latter were considered essential. Meanwhile, we feel that these data are clear and will be readily interpretable by readers less expert than Dr Murray. For instance in **Fig 5**, it is evident both on immunohistochemistry and immunofluorescence that ZG and ZF cells are very different – elongated vs round nuclei, compact vs large cells – and that gap junction plaques and annular gap junctions (white arrows) are present in both cell types. The expected abundant linear staining for GJA1 is seen in cells positive for CYP17A1, an obligatory transcript in ZF cells; whilst less abundant staining, partly linear, partly punctate, is seen in the ZG cells of the image **Fig 5b(ii)**. Membranous expression of GJs is less in ZG regions than in ZF regions, confirmed in **Fig 5c** where an overlay is performed with WGA (yellow) as membrane marker, yet both punctate and annular gap junctions are clearly present within the membrane as well as internalised. Dr Murray particularly liked the image from **Fig 6** ('Emily's figure' in Dr Murray's text, which was the solitary confocal figure in our revised manuscript). This image not only exemplified formation of annular gap junctions (white arrows) from small gap junction plaques (yellow arrow), but also confirmed intramembranous origin of the punctate GJs as shown by co-localization of GJA1 and WGA (orange staining where green and red overlays).

Summary

We have reported five spontaneous examples of the same somatic mutations, in adjacent residues of *CADM1*, and shown these to increase by 10-30 fold the production of aldosterone. This of itself is interesting, because it is the first mutation causing autonomous aldosterone production from an adenoma that is not in a gene encoding either an ion-channel/transporter (*KCNJ5*, *ATP1A1*, *ATP2B3*, *CACNA1D*, *CACNA1H*, *CLCN2* – all except *KCNJ5* first reported in *Nat Genet*) or within the known aldosterone synthesis pathway (*GNA11/Q*). It is also a rare example of an intramembranous mutation of the many thousands of genes encoding single-pass membrane proteins, and our structural modelling tallied nicely with the previous NMR analyses of the dwarfism causing mutations of *FGFR3*.

However we also discovered two effects of the mutations which bring to attention two novel clinical and mechanistic aspects of aldosteronism. The clinically-relevant aspect is the possible exaggeration by *CADM1* mutations of diurnal variation in aldosterone secretion: upregulating RORE clock genes expressed at night, downregulating E-box genes expressed by day. For future study is the question whether *CADM1* mutations explain the recently reported frequent patients who are detected by 24h urine, but not randomly timed blood analyses of aldosterone.

The mechanistic discovery, to which a larger part of the manuscript is devoted, and the main subject of this appeal, is that – unlike non-human adrenals in which aldosterone production occurs throughout the zona glomerulosa (ZG) – gap junction communication does take place in those parts of human ZG where aldosterone production is absent. When this communication is inhibited (our silencing and gap27 experiments in cultured adrenocortical cells) aldosterone production returns. In the cell clusters of intact adrenals where gap junction communication is absent or rare, aldosterone production is high, as is the case diffusely in the ZG of other species.

Thus the *CADM1* mutations not only provide a new mechanism of abnormal aldosterone production in patients with PA, but an experiment of nature which has unmasked novel biology within the *normal* adrenal. It is these impacts, alongside their support from extensive new experiments, and endorsement of the doyenne of the field, that we believe of sufficient interest to merit further review by Nature Genetics. We are grateful to you for your consideration of this appeal.

Decision Letter, Appeal:

13th January 2022

Dear Morris,

Thank you for asking us to reconsider our decision on your manuscript "Somatic intramembranous mutations of *CADM1* in aldosterone-producing adenomas, and gap junction dependent regulation of aldosterone production". I have discussed your appeal with my editorial colleagues, and based on the new genetic data and the additional evidence supporting the mechanistic interpretations of the study, we invite you to revise your manuscript as proposed for further peer review.

When preparing a revision, please ensure that it fully complies with our editorial requirements for format and style; details can be found in the Guide to Authors on our website (<http://www.nature.com/ng/>). In particular, please ensure that each figure is sized to fit onto a single page and labeled according to journal style requirements.

Please be sure that your manuscript is also accompanied by a separate point-by-point detailing the changes you have made and your response to each point raised at the previous round of review. At this stage, we will also need you to upload:

- 1) A copy of the revised manuscript in MS Word .docx format.
- 2) The Editorial Policy Checklist:
<https://www.nature.com/documents/nr-editorial-policy-checklist.pdf>
- 3) The Reporting Summary:
<https://www.nature.com/documents/nr-reporting-summary.pdf>

(Here you can read about the role of the Reporting Summary in reproducible science:
<https://www.nature.com/news/announcement-towards-greater-reproducibility-for-life-sciences-research-in-nature-1.22062>)

Please use the link below to be taken directly to the site and view and revise your manuscript:

[redacted]

With kind wishes,
Kyle

Kyle Vogan, PhD
Senior Editor
Nature Genetics
<https://orcid.org/0000-0001-9565-9665>

Author Rebuttal, first revision:

Main changes made to revised submission NG-LE55401R3

The major changes arise from new studies prompted by reviewer and editorial suggestions.

The 3 'headline' findings are [i] 3 further patients with one of the same somatic mutations of *CADM1* as we previously reported; [ii] co-localisation of *GJA1* with several markers of ZG cells, using immunofluorescence in serial or confocally analysed sections; [iii] a dose-related 10-100 fold increase in aldosterone secretion and *CYP11B2* expression when H295R adrenocortical cells are incubated with Gap27.

- **New Fig. 1. Discovery of *CADM1* somatic mutations in aldosterone-producing adenomas (APAs)** ○ Includes the finding of 3 new patients with *CADM1* mutations in the exact same residues as the first 2 reported patients.
- **New Fig. 5. Inhibition of gap junction communications increase aldosterone production** ○ Expanded the n number and inclusion of several concentrations for the Gap27 experiments and *SiGJA1/GJC1* co-silenced experiments.
 - Included the IFC for *GJA1* in subcapsular cells not expressing *CYP17A1*.
- **New Extended Data Fig. 1. Characteristics of *CADM1*-mutant APAs** ○ Includes the finding of a ZG-like phenotype for *CADM1* mutant APAs and intense membranous expression of *CADM1*.
 - Includes chromatogram of the *CADM1* mutations in the 3 new patients.
- **New Extended Data Fig. 2. *GJA1* expression in human adrenals** ○ Includes IFC that detects punctate expression of *GJA1* co-localizing with WGA (a cell membrane marker), GJ plaques, and AGJ in ZG cells as determined by lack of *CYP17A1* (a ZF marker) and expression of ZG marker *VSNL1*.

- Similarly, aldosterone-producing micronodules, as determined by CYP11B2 expression, also have AGJ, but general membranous expression of GJA1 is reduced compared to ZF or even ZG.
- **New Extended Data Fig. 3. AQP2 expression in *CADM1*-mutant APAs** ○ Includes more IHC data showing AQP2 expression in *CADM1*-mutant APAs and increased gene expression of AQP2 in transduced human adrenocortical cells transduced with mutant *CADM1*.
- **New Supplementary Fig. 1. *CADM1* somatic mutations found in APAs affect protein tertiary structure** ○ Clarified the western blots and predicted schema of *CADM1* mutation
- **New Supplementary Fig. 4. Differential expression of GJA1 in human adrenal cortex** ○ Includes new IFC for DAB2, a selective ZG marker.
 - Includes image of an AGJ in a CYP17A1 negative cell.
- **New Supplementary Fig. 5. GJA1 is least expressed in aldosterone-producing micronodules (APM)** ○ Includes new IFC for DAB2, a selective ZG marker.
 - Includes image of an AGJ in a CYP11B2 positive cell (APM).
- **New Supplementary Fig. 7. Inhibition of gap junction communication in adrenocortical cells**
 - Expanded the n number and inclusion of several concentrations for the Gap27 experiments.
 - Addition of a Gap27 treatment study in adrenal cells cultured from the adrenal cortex of a patient with a cortisol-producing adenoma

Reviewer #1:

Remarks to the Author:

The rebuttal is difficult to read because line numbers given in the rebuttal do not correspond to those in the manuscript. I am unable to find changes in the manuscript that the rebuttal refers to (e.g., “p12, final para” shows no marked changes, and I do not see an explanation of the divergence between effects of silencing on CYP11B2 and gap junction formation there).

We apologise that this was not clear. The explanation referred to above was within the final paragraph of page 12 and 1st paragraph of page 13, but with marked changes only shown on page 13. Our explanation for the divergence between effects of silencing on CYP11B2 and gap junction formation is that “The inter-cellular adhesion of *CADM1* ectodomains is likely to have two roles. One is extracellular, concerning communication between cells. The other is the equivalent of ligand binding to many of the single-TM domain proteins, which results in intra-membrane dimerization, conformational change and intracellular actions. The C-terminus of *CADM1* has several partners, PDZ, 4.1B, MPP3, whose activation or inhibition may be triggered by conformation change or intracellular cleavage of the CTF. It is likely that loss of both the C-terminus and N-terminus of the *CADM1* molecule on silencing, whereas activation of the cytosolic tail region by mutation, underlies the opposite effects of silencing and mutation on aldosterone production.” (Lines 242-249 Page 13).

Regarding the RNA sequencing data, p. 5, l. 170 mentions *GSTA3* as a marker. The corresponding Figure indicates *GSTA1* was studied – unclear what was done. In any case, both would be uncommon marker for ZF. Valid ZG and ZF markers would be *CYP11B2* and *CYP11B1*, which could also demonstrate how well the laser capture procedure (contamination will occur) worked. To clarify the pathophysiology of increased *CYP11B2* expression, demonstration of weak expression in so-called glomerulosa cells that do not express *CYP11B2* does not seem particularly useful. Fig. 4Ciii: absence of *CX43* in *CYP11B2*-positive cells – how is *CX43* expected to have the proposed effect on aldosterone production if it is not present in the cells that express aldosterone synthase? Normal adrenals of young individuals rather than tumor-adjacent material might be helpful.

We are grateful for the reviewer's scrutiny of the data. The *GSTA1* label was a typing error in the figure. *GSTA3* was studied in the cells isolated by laser capture, as the microarray data, to which we refer, had found both *GSTA1* and *GSTA3* to be the top differentially expressed gene in ZF compared to ZG (Zhou *et al.*, *Hypertension* 2016, <https://doi.org/10.1161/HYPERTENSIONAHA.116.08033>). Although our group has not verified the selective expression of the protein in ZF, images from the Human Protein Atlas (<https://www.proteinatlas.org/>), shows *GSTA3* protein is expressed in the ZF while absent in peri-capsular cells.

The laser capture microdissection technique we use requires fresh frozen adrenal sections for RNA acquisition. Due to current circumstances, availability of normal adrenals from

young individuals are limited. The majority of adrenals available to us are excised due to an APA. Hence, we could not use *CYP11B2* as a marker for ZG as this is usually downregulated in the adrenal adjacent to an APA (except for in autonomous aldosterone-producing micronodules, APMs). (Please see striking evidence of this in the following references: Gomez-Sanchez CE *et al.*, *Mol Cell Endocrinol* 2014, doi: 10.1016/j.mce.2013.11.022.10; Nishimoto K *et al.*, *PNAS* 2015, <https://doi.org/10.1073/pnas.1505529112>; Nishimoto K *et al.*, *JCEM* 2010, doi: 10.1210/jc.2009-2010). Expression of *LGR5*, our top differential expressed gene when we compared ZG and ZF from adrenals adjacent to an APA, and *GSTA3*, were used as an adjunct to Cresyl Violet, a staining validated by others to selectively capture ZG and ZF cells from both human adrenals and adrenals from rodents (Nishimoto K *et al.*, *PNAS* 2015, <https://doi.org/10.1073/pnas.1505529112>; Nishimoto K *et al.*, *Endocrinology* 2012, doi: 10.1210/en.2011-1915).

The reviewer's doubt over relevance of CX43 (GJA1) pathophysiology with aldosterone production prompted us to perform further IFC staining in human adrenals to capture, unequivocally, annular GJA1 (AGJs), a product of gap junction (GJ) inter-cell communications (GJIC) in CYP11B2-positive cells (**New Extended Fig. 2f-g** and **New Supplementary Fig. 5b**). Additionally, we hope that the reviewer finds our new IFC images showing punctate GJA1 and AGJs in the ZG (**New Extended Data Fig. 2d-e** and **New Supplementary Fig. 4e**) to be supportive of our mRNA findings in LCM samples. Also, in IFC, we see that GJA1 is more abundant in the ZG than APMs as previously seen with IHC (**New Extended Data Fig. 2f-h** and **New Supplementary Fig. 5**).

We have considerably expanded our data showing that inhibition of gap junctions increases aldosterone production (**Fig. 5** and **Supplementary Fig. 7**), from which we infer that endogenous gap junctions are inhibiting aldosterone production. In answer to the reviewer's puzzle, how can gap junctions inhibit CYP11B2 if not present in the cells where it is expressed, it may be helpful to draw attention to the observation that CYP11B2 is entirely switched off in most of adult human ZG (see refs 10, 44 and 60). This observation has been widely replicated since development of monoclonal antibodies which differentiate CYP11B2 from the 93% identical enzyme making cortisol (CYP11B1). The suppression of CYP11B2 expression is assumed to be due to high human salt intake. But this is unproven, and we do not know if the inhibition of aldosterone production is due solely to a reduction (by salt) of circulating renin and angiotensin, or involves local pathways, whether of angiotensin response or sodium sensing. Against this background, it becomes plausible that ZG cells which have suppressed CYP11B2 expression may be found to express one or more factors (such as gap junctions) which reduce CYP11B2 expression. Conversely, autonomous aldosterone-producing micronodules (where CYP11B2 expression is high), are expected to have low or absent expression of factors which inhibit CYP11B2 expression, and thus reduced GJs.

Because of the importance of this part of our paper, we have sought advice from Prof. Dr. Sandra Murray, whose work on adrenal gap junctions we and reviewers had previously cited. She analysed all our pictures of gap junctions, and other gap junction data, and agreed with our conclusions, but suggested some of the newer images cited above. Dr Murray has joined us as co-author of the paper.

Fig. 4b: There seem to be CYP17A1-positive cells in what is apparently assumed to be ZG. KCNJ5 is not an ideal ZG marker because its expression extends into the ZF.

Due to the limitations of serial sections of IHC, we have now provided additional new IFC data which unequivocally shows CYP17A1-negative peri-capsular cells to express GJA1, and AGJs (**New Fig. 5a**, **New Extended Data Fig. 2d** and **New Supplementary Fig. 4e**). We have also included VSNL1 and DAB2 as a ZG marker (**New Extended Fig. 2e** and **New Supplementary Fig. 4d**). Both these proteins have been validated by other groups to be ZG selective (*Trejter M et al., Peptides 2015, doi:*

10.1016/j.peptides.2014.10.017; Seccia TM et al., *Hypertension* 2017, doi:10.1161/HYPERTENSIONAHA.117.09991). In our own microarray of 20 LCM-extracted adrenals, mentioned above, VSNL1 was the 2nd most upregulated ZG gene after LGR5, 23.5-fold compared to ZF, $p=3.65 \times 10^{-23}$.

The legend claims CX43 is found in mutant APAs. It is unclear where an APA is shown in the figure (label?).

We apologise for the ambiguity. The previous legend of Fig. 4b was in reference to previous Fig. 4d. The legend has now been clarified (**New Supplementary Fig. 4c**).

Supplementary Fig. S4a is unlabeled – it is unclear what is shown. Is this an adrenal gland with a tumor?

Many thanks for pointing out this obscurity. We have now labelled the figure to show the APA that is present in the adrenal and highlighted it in the figure legend (**New Supplementary Fig. 4a**).

Regarding the modelling (Supplementary Fig. S1 (d)): The lower panel is confusing regarding the scale. For example, the length of the WT TM domain should be 33 Å according to the upper panel. In the lower panel, the height is depicted as 33 Å, which cannot be correct if the model is depicted on scale. Similarly, whereas in the upper panel, distances appear to be scaled, that appears not to be the case in the lower panel, with 33 Å, 31.74 Å and 31.42 Å all depicted as the same height. More importantly, the TM is shorter in mutants than in the WT, which acts contrary to the tilt with regard to the point of cleavage. That is not discussed in the main text.

We apologise for the discrepancy. The arrow should have been parallel to the TM domain and the numbers should have shown the predicted TM helix length rather than the predicted TM domain length. We have now corrected and clarified accordingly (**New Supplementary Fig. 1d**). We have also added a critical reference to the text (*Kaczur et al, J Mol Recognit* 2007, doi:10.1002/jmr.851. 2007) regarding the cleavage site for ADAM10 being at a defined distance from the membrane, with an estimate for various substrates being in the 10-20 residue range. We infer from this number that the predicted shortening of the TM domain is small compared to the effect of an angle change on the number of residues between plasma membrane and ADAM10 cleavage, and have – if the reviewer agrees – left the schema in **New Fig. 3c** unaltered. Because our modelling, and the cited experimental data for TM domain mutations of FGFR3, concerns the TM domain, and only indirectly the extracellular domain, the best evidence for the cleavage site being nearer to the membrane in mutant than wild-type CADM1 is the shorter CTFs consistently seen on Western blots (**New Fig. 3b** and **Supplementary Fig. 1a-b**). This is discussed in the manuscript between **Lines 108-122 Page 8**.

My suggestion to show the somatic mutations discovered in tumors without mutations in known genes has not been addressed.

We apologise for the misunderstanding. We had thought the previous Supplementary Table S1b showing all verified somatic mutations affecting coding regions discovered in the index case 184T addressed this issue (**New Extended Data Table 1b**). We however feel that as we have not verified the non-recurrent somatic mutations discovered in other tumours without known mutations, and the number of these is larger than in the smaller whole exome studies published in 2013, interested readers can nowadays download the raw WES data publicly available (from the Sequence Read Archive under accession nos. PRJNA732946 and PRJNA729738). This information has now been added to the Data Availability section (**Lines 494-495 Page 21**).

Similarly, my question about the time of day of the aldosterone and renin measurements was not answered, and why this is considered cyclic remains unclear. Fig. 1(b)ii shows three normal ARR's and one moderately elevated ARR pre-op.

Unfortunately, the time of day the aldosterone and renin measurements were taken is not available. From the data we presented we cannot say whether the three apparently normal ARR's seen in the index case were due to cyclical/periodic secretion of aldosterone akin to the recognised phenomenon described in Cushing's, or whether these were due to an exaggerated diurnal rhythm. RNAseq data of H295R cells transduced with vector, WT and mutant *CADM1* showing up- and downregulation of the clock genes supports the latter. This degree of clinical data is also not available from the three extra patients newly identified to have *CADM1* mutations.

We too think the diurnal variation in aldosterone is not sufficiently emphasised (and accounted for) in routine clinical practice, in the way as for cortisol. The changes in expression of the clock genes in our RNAseq experiments highlights this. A recently published paper from Boston highlighted the potential number of undiagnosed patients with PA (*Brown, JM et al. Ann Intern Med 2020, doi: 10.7326/M20-0065*). We think individuals with *CADM1* mutations may be within this group of patients whose diagnosis may be missed by conventional methods of diagnosis and are planning a project further to investigate this.

We have now referred both cases to have "episodic hyperaldosteronism, which identified variable diagnostic aldosterone-renin ratio (ARR)" (**Lines 60-62 Page 6**).

It remains unclear to me what the many bands in Supplementary Fig. S1 represent. Showing lysates of untransduced cells would be standard.

We have taken on board the reviewer's comments regarding showing lysates of untransduced (UT) cells (**New Supplementary Fig. 1a**). We have also simplified the immunoblot and clearly labelled all relevant bands as suggested.

Upon re-reading the manuscript, I note that in Fig. 1(b)i, the variant G379D is not correctly displayed. If V380D results in the sequence AVIGGDVAV..., G379D should result in the sequence AVIGDVV... rather than AVIGGDV... as given in the figure.

Apologies for this typing error. This has now been rectified (**New Extended Data Fig. 1a**).

On p. 6, ll. 65-67, the authors claim that the presence of the mutation in cDNA is proof that the consequences of the mutation are due to expression of an abnormal protein. That is formally not correct because they study cDNA rather than protein.

Thank you for pointing this out. We have clarified this in the manuscript (**Lines 67-70 Page 6**).

Minor:

Supplementary Fig. S1 (d), legend: "helix with the lipid bilayer was increased"

R: This has been amended (**New Supplementary Fig. 1d**). Thank you.

Fig. 4a, legend: "fasciculate" -> fasciculata

R: Noted with thanks (**New Extended Data Fig. 2a**).

Suppl. Fig. S4d legend: chi shows as "?"; 2 independent histologists (not histologist) (same in Fig. 4(c) legend), p. 47 two histopathologists (not histopathologist)

R: This has been amended (**New Supplementary Fig. 5c**). Thank you.

Fig. 4(d): what does the "+" symbol mean?

R: The centre of the + denotes area of interest, which is shown at higher power in the right-hand panels. We have now clarified this in the figure legend (**New Supplementary Fig. 6a**).

Methods, p. 47: reference to “Supplementary Figure SX” - that Supplementary Figure does not exist.

This has been amended (**Line 131 Page 48 – Methods**). Thank you.

L. 201: “The effects were dramatic” – suggest to delete.

R: We have amended as suggested.

L. 203: 20-200-fold: That does not seem to be correct. AngII + Gap27 should be compared to AngII, which probably corresponds to less than 100-fold elevation.

R: Thank you for pointing this out. We have rectified this as an average in the manuscript (**Lines 223225 Page 12/13**).

L. 230-232: unclear

R: We have now edited for clarity (**Lines 247-249 Page 13**).

Reviewer #2:

Remarks to the Author:

The answers to my previous comments are acceptable.

Many thanks.

Reviewer #3:

Remarks to the Author:

The phenotypic data presented in the original manuscript appears to be contradictory with the dye transfer and GAP27 inhibition studies. That is, the results from the *CADM1* overexpression and knockdown studies seem to differ from the rest of the story. The earlier studies seem to suggest that GJ communication leads to enhanced aldosterone production (with the exception of overexpression of WT *CADM1*, which has no effect at all), whereas the dye transfer and GAP27 peptide studies argue that less GJ communication promotes aldosterone production.

Even in the original manuscript, we do not believe to have shown any results suggesting overexpression of mutant *CADM1* increases GJ communication which leads to enhanced aldosterone production. From our data, we are proposing that mutant *CADM1* reduces GJ communication as supported by our dye transfer experiment (**New Fig. 4** and **New Supplementary Fig. 2**) and predicted by our protein modelling of the mutant (**New Fig. 3** and **New Supplementary Fig. 1**).

Perhaps the reviewer was thinking of our experiment with ShRNA where we showed reduction in CYP11B2 mRNA expression and aldosterone production in H295R cells transduced with Sh*CADM1* compared to non-targeting vectors (Fig. 4b in original manuscript). There, immunofluorescence of transduced cells and western blot of cell lysates found silencing of *CADM1* associated with increased GJA1 expression. However, our dye transfer clearly shows that GJ communications is decreased in Sh*CADM1* cells (**New Fig. 4b** and **New Supplementary Fig. 2b-c**). One interpretation of this could be that the reduced GJ communications led to overexpression of GJA1, though to note the most abundant GJ in H295R cells (unlike primary human adrenal cells) is GJC1, not GJA1.

We agree with reviewer #3, as also mention to reviewer #1, that the similar effect of mutation and silencing of *CADM1* on GJ communication, but opposite effects on CYP11B2 expression, is a key and fascinating issue. As postulated in the manuscript, and as described with supporting structural (NMR) data for other single-pass membrane proteins, dimerization of *CADM1* in the membrane is a plausible or likely response to homophilic adhesion of *CADM1* ectodomains in adjacent cells, with downstream consequences from conformational changes at the C-terminus. Silencing of *CADM1* will suppress all its actions, including partnerships with the PDZ- and protein 4.1 binding regions at the Cterminus. By contrast, the C-terminus of mutant *CADM1* remains intact to mediate whatever signals arise from gap-junction inhibition, and may even be activated by the mutation (as shown for the intramembrane mutation of *FGFR3*). Whether or not gap-junctions are the sole pathway with which mutated *CADM1* interacts directly to stimulate aldosterone production, the shRNA experiment points to an essential downstream pathway through which the stimulus must work. This has now been highlighted in the manuscript (**Lines 242-249 Page 13**).

The clock genes, which are mainly transcription factors, are among the possibilities, as are PDZdomain containing proteins, such as TJP1. Our new illustration of this tight junction protein's highly selective ZG expression, quite different between APM- and non-secretory ZG cells, appears to be an original observation (**New Supplementary Fig. 6b-c**). To note, the index *CADM1*-mutant APA had punctate expression of both GJA1 and TJP1 (**New Supplementary Fig. 6a**).

I remain unconvinced that the ZG contains gap junctions, a central premise of this manuscript. While the LCM (Fig 4a) appear to show that CX43 is expressed in the ZG (albeit at lower levels than in the ZF), but is LGR5 the best ZG marker to be using here? Should also use well established ZG marker genes such as DAB2 and Gq? Unfortunately, the CX43 immunostaining data remains unconvincing and may represent background staining--it is certainly not marking gap junctions (e.g., Fig 4b, 4c, 4d). While the control staining maybe somewhat helpful at addressing the background issue, its presence in the cytoplasm or as puncta rather than at the membrane (where gap junctions should reside and do appear to within in ZF regions) undermines the authors primary hypothesis and conclusion.

We thank the reviewer for scrutiny of the IHC data as this prompted us to perform further IFC staining in human adrenals to capture, without doubt, the presence of GJs and AGJs, as a by-product of GJ inter-cell communications (GJIC), in the ZG (**New Fig. 5a**, **New Extended Data Fig. 2d**, and **New Supplementary Fig. 4e**). This includes with staining of DAB2 as a ZG marker (**New Supplementary Figure 4d**), as suggested by the reviewer, and also VSNL1 (**New Extended Data Fig. 2e**) another ZG protein that has been found by ourselves and others to be ZG selective (*Trejter M et al., Peptides 2015, doi: 10.1016/j.peptides.2014.10.017*). In addition, IFC has allowed us to show, unequivocally, that GJA1 in the ZG is expressed on the cell membrane through co-localization with the cell membrane marker wheat germ agglutinin (WGA) (**New Extended Data Fig. 2e**).

We were also fortunate enough, considering the dynamics of GJIC, to have captured the budding of AGJs from a GJ plaque which should remove all doubt that these AGJs seen in the cytoplasm of both ZG and ZF, arise from GJ plaques present on the cell membrane (**New Extended Data Fig. 2e** and **2g**). To note, IFC had also allowed us to see reduced expression of GJA1 in CYP11B2 positive cells (APM)(**New Extended Data Fig. 2f-g** and **New Supplementary Fig. 5a-b**) supporting our primary hypothesis and conclusion that inhibition of GJIC is associated with aldosterone production and is one of the mechanism that contributes to the *CADM1* mutant APA phenotype.

We had also taken the opportunity to present our data to the GJ expert cited by both reviewer #1 and #3 in their first review to the original manuscript, Prof. Dr. Sandra Murray, due to previous concern of our inexperienced interpretation of the immunostaining. From the new IFC data and other relevant evidence in the manuscript, we quote excerpts of her expert opinion after reviewing our data as follows:

“convincing evidence was presented in this manuscript that there are small areas of Cx43 (GJA1) punctate staining at the areas of cell-cell contacts, consistent with the presence of gap junction plaques. In addition, annular gap junctions are present within the cytoplasm of the ZG cells, indicative of gap junction plaque internalization from the plasma membrane into one of two contacting cells. Further, strong evidence for Cx43 in the ZG of human was provided by mRNA microarray data collected with Laser Capture microscopy.”

“Most of the gap junction plaques seen in the ZG were small and appeared punctate with immunocytochemical imaging. This is unlike the larger and typical linear gap junction plaques found in the ZF. The smaller and fewer numbers of gap junction plaques at points of cell contact in the ZG would generally be thought to suggest that this zone is less dependent on cell-cell communication. However, the question of the relationship between numbers and sizes of gap junctions and functional capacity has not been studied extensively enough. In general, plaques in tissues and primary cells are smaller than plaques in expression systems... Dieter Hülser once published that HeLa are coupled and he detected very small punctate Cx45 plaques that were way below 100 channels (Hülser, D., Rütz, ML., Eckert, R. et al. Pflügers Arch - Eur J Physiol 441, 521–528 (2001). <https://doi.org/10.1007/s004240000460>).”

“Given the morphological and functional evidence of gap junctions in the ZG and the detection of Cx43 mRNA in the human adrenal tissue being used and adrenal cell lines, it is clear Cx43 proteins are present in the ZG, that they form small Cx43 gap junctions capable of dye communication between cells that makes a strong argument for a physiological role for the gap junction regulation of events in ZG.” –Prof. Dr. Sandra Murray

Dr Murray joined us as co-author of this revision, for which she has advised and commented on further IF images undertaken since her report above. All GJ and AGJ assignments in the figures have been checked by her.

We therefore hope the reviewer will find our new IFC images showing presence of GJA1 (and AGJs) in both ZG and ZF cells convincing, complementing data already shown with IHC, and including all the standard controls to take into account background staining (i.e. antigen peptide controls and no primary antibody controls). As for our LCM results, expression of *LGR5*, our top differential expressed gene when we compared ZG and ZF from adrenals adjacent to an APA (Zhou J et al., *Hypertension* 2016, <https://doi.org/10.1161/HYPERTENSIONAHA.116.08033>), were used as an adjunct to Cresyl Violet, a staining validated by others to selectively capture ZG and ZF cells from both human adrenals and adrenals from rodents (Nishimoto K et al., *PNAS* 2015, <https://doi.org/10.1073/pnas.1505529112>; Nishimoto K et al., *Endocrinology* 2012, doi: 10.1210/en.2011-1915).

The additional GAP27 data are interesting but appear to raise important new questions and are not fully convincing.

Regarding the experiments using H295R cells and Gap27 the following concerns or questions arise:

One cannot use two T-tests to assess significance--one must use ANOVA. In this case, the effect of Gap27 on basal expression would likely be eliminated.

It appears that N=8 (for example in Fig 5a) likely represents technical replicates. Whatever the number is should indicate the number of independent studies not the number of replicates from a few studies.

The methods were not entirely clear. How long was the Gap27 incubation? Was it enough time to change expression of Cyp11b2 and other ZG markers (such as DAB2 and Gq)?

If correct, these data seem to suggest that GAP27, or knockdown of CX43, enhance the zG phenotype, consistent with the notion that the lack of GJ communication makes H295R cells more zG-like and less zF-like. It would be important to point out in the discussion that such cells can be pushed in either direction. Such a result could be consistent with the possibility that ZG cells do not utilize GJs at all for communication while ZF cells do.

We apologise for the lack of clarity. In both Gap27 experiments, H295R cells and human primary adrenal cells, incubation with Gap27 was for 24 hours. This detail has now been added to the methods (**Lines 193-195, page 50 - Methods**). All data shown previously were not technical replicates but biological replicates (independent wells). The results of these biological replicates are the average of 2-3 technical replicates as this is our standard protocol when doing qPCR measurements or aldosterone measurements (only the average is shown). We have now increased the number of biological replicates for the experiments using H295R cells and have clearly stated in the legend of the figures that these biological replicates are from 3 independent studies (**New Fig. 5b** and **New Supplementary Fig. 7a**). The statistical significance for these experiments has been recalculated using the non-parametric ANOVA, Kruskal-Wallis H test, as recommended. The individual data points for these figures are now provided

To note, we have consistently seen, in both H295R and primary cell experiments, much larger effects of Gap27 on CYP11B2 expression and aldosterone secretion in angiotensin II-stimulated cells though we have yet to find a robust explanation for. In the revised manuscript, we have shown the new data in which a complete dose-response for Gap27 was performed in angiotensin II-stimulated cells (**New Fig. 5b**), and not attempted to pool this data with the previous experiments now shown in the supplementary figures (**New Supplementary Fig. 7a**). The new experiments (**New Fig. 5b**) were performed at higher seeding density than the previous experiments (**New Supplementary Fig. 7a**) as we saw that the confluency of the cells during the treatment can affect angiotensin II response (**New Supplementary Fig. 7h**). Indeed, a ten-fold greater response was seen in the cells plated at higher seeding density even at the lower dose of Gap27 (25 μ M), 2.7-fold vs 24-fold. Nevertheless, both sets of experiments (both repeated independently 3 times) found Gap27 to have a dose response in increasing CYP11B2 of angiotensin II-stimulated H295R cells which should remove any doubts the reviewer has had of the significance of this data.

We thank the reviewer for the suggestion that the role of GJ may be in regulating ZG cell phenotype rather than aldosterone production per se. Considering the theory of centripetal migration of cells in the adrenal cortex, i.e. ZG differentiating into ZF cells, the reviewer's suggestion would explain why only small punctate GJs are seen in ZG cells as they begin to change phenotype as soon as large linear GJ plaques are formed. This may also explain why the increase of aldosterone with Gap27 in H295R cells only occurs in the presence of angiotensin II; Gap27 changes the phenotype of the cells to more ZG-like (seen with increase of CYP11B2), yet stimulation of cells is still required to see the difference in aldosterone production. As these are just postulation and hypothesis we have not discussed this in the manuscript, but would do so if deemed appropriate for the scope of this manuscript. Instead, we have complemented the H295R results with experiments from primary human adrenal cells for the readers to interpret accordingly (**New Fig. 5d** and **New Supplementary Fig. 7g**).

Regarding the experiments measuring aldosterone secretion from human primary cultures:

There is a concern that the primary cells do not appear to be healthy since only 1 adrenal showed Ang II stimulated aldosterone production and in that adrenal Gap27 had no statistical effect. Curiously in the other adrenals, which did not show Ang II stimulated aldosterone production, an effect of Gap27 was observed.

We took on board the reviewer's concern and have included results for cells treated with DMSO 10% to demonstrate that the increased expression of *CYP11B2* and increased aldosterone production seen with treatment of Gap27 is not as a result of dead/unhealthy cells leaking aldosterone (**New Fig. 5b**). As we were using adrenal cells adjacent to APAs, and APAs themselves are known to have variable response to Ang II-stimulation (*Tunny TJ, et al., Clinical Endocrinology 1991, <https://doi.org/10.1111/j.1365-2265.1991.tb00306.x>*), we suspect the variable results reflect the different impact the APAs has had on the adjacent adrenal rather than due to the cells not being healthy.

We agree that the effect of Gap27 in Ang II-stimulated cells appears inversely proportional to the Ang II-stimulation response. In adrenal 5 (Adr 5) where the cells adjacent to a cortisol-producing adenoma were cultured, a 20-fold increase of ang II-induced *CYP11B2* was seen, but no augmentation when treated with Gap27. This is consistent (but, we acknowledge, does not prove) the suggestion of reviewer 3 that GJIC inhibition leads to an enhanced ZG cell phenotype (and ability to produce aldosterone). The adrenal cells adjacent to a cortisol-producing adenoma are expected to have a reduced proportion of ZF-like cells due to the high secretion of cortisol, and thus reduced scope for differentiation, by Gap27 from ZF- to ZG-like phenotype. If the reviewer agrees, we think that, whilst variation between samples prompts interesting hypotheses, these are not required in (and beyond the scope of) the present manuscript.

Variation of response to Gap27 in the different adrenals could also reflect the confluency of the cells treated. For GJ communication to occur cells need to be within close proximity of neighbouring cells, therefore a high confluency of cells is required for these experiments. However, due to paucity of primary adrenal cells, the primary cell experiments were plated at the maximum cell density that still allowed for a minimum of 3 biological replicates ($n=3-4$ wells). The seeding density of the four adrenals in a 48 well plate were: 12×10^4 cells/well (Adr 1); 1×10^4 cells/well (Adr 2); 1.2×10^4 cells/well (previously Adr 3, currently Adr 4); and 7×10^4 cells/well (previously Adr 4, currently Adr 3). To note, the adrenal with the smallest response to Gap27 also had the lowest number of cells in a well (Adr 2). The importance of cell confluency in this experiment has been highlighted in the manuscript by **New Supplementary Fig. 7h**.

Regarding the following statement: "A feature of the experiments with CADM1 mutation and GJ inhibition has been the greater increase in CYP11B2 than aldosterone. A speculative explanation which connects to the AQP2 findings is that, if secretory granules are formed, aldosterone secretion may be more episodic or regulated than in other causes of increased aldosterone release." Are the authors speculating that formation of secretory vesicles is related to aldosterone release? As far as I am aware, no steroid hormone is released in vesicles. Such speculation should be beyond the scope of this manuscript without the appropriate data to support the claim.

We have started immunoelectron microscopy analysis of APAs and APMs, in part to determine whether granules are found at sites of high synthesis of aldosterone production. However, we agree that our Discussion should concentrate on explanations for which we sought experimental evidence and therefore deleted this speculation from the manuscript.

Finally, I have questions and concerns related to the experiments measuring calcium flux in H295R cells.

The authors need to show representative traces for Ang II-induced Ca changes with and without Gap27. If the cells are truly oscillating, then Ang II should produce a large transient rise in Ca, which then returns close to baseline. If there is only a large rise that does not return the cell is dead.

We can confirm, as per the representative traces below, that the mean cell intensity of all oscillating cells returned to the baseline, demonstrating all cells were viable. This is now highlighted in the legend of **New Supplementary Fig. 8b**.

Regarding the statistical analysis, I do not believe that a T-test is appropriate for comparing intensities. In addition, a Fisher exact test is more commonly used to compare 2 x 2 contingency tables. Is this an appropriate method here? Might consider using a non-parametric ANOVA.

We used the Fisher Exact test to assess whether treatment with Gap27 affected the number of cells which were oscillating (P -value=0.5223). The contingency table used for calculations in **New Supplementary Fig. 8c** are as below.

	number of cells showing oscillations (Oscillation)	number of cells not oscillating (No oscillation)
Untreated	19	25
Gap27 treated	23	21

For the second set of experiments (**New Supplementary Fig. 8d**) where 4 conditions were tested, the Chi Square test shows overall there is a difference in number of oscillations between the different treatment groups ($p < 0.0001$):

	Oscillation*	No oscillation**
Untreated	27	18
Gap27 treated	30	15

AngII	46	0
AngII + Gap 27	42	4

However our interest is on the effect on Gap27 therefore Fisher Exact tests were performed on the following conditions:

Untreated vs. Gap27, p -value=0.6621

AngII vs. AngII + Gap 27, p -value=0.1168

Using this statistical method, the expected increase in oscillating cells between Untreated vs AngII treated cells was highly significant, p -value<0.0001.

With regards to the statistics for comparing the mean cell intensity of at baseline, we have taken the reviewer's suggestions for a non-parametric statistical test and have repeated the analysis using the Mann-Whitney U test which confirmed the difference in baseline cell intensity between untreated and Gap27 treated cells (p <0.0001).

For **New Supplementary Fig. 8d** we have now performed the non-parametric ANOVA, Kruskal-Wallis H test, as suggested, which confirmed difference between the 4 treatment groups, $p = <0.0001$. Further comparisons between groups were made using Dunn's multiple comparison test which showed difference in baseline calcium intensity between the untreated and Gap27 treated group ($p=0.0013$), and between untreated and AngII treated ($p=0.0094$), but not between the AngII treated and AngII+ Gap 27 treated group (0.1489).

Decision Letter, second revision:

9th June 2022

Dear Morris,

Your revised Letter "Somatic intramembranous mutations of CADM1 in aldosterone-producing adenomas, and gap junction-dependent regulation of aldosterone production" has been seen by two of the original referees. You will see from their comments below that, while Reviewer #1 finds the study improved, both referees have expressed ongoing concerns about aspects of the presentation and interpretation of the findings. We remain interested in the possibility of publishing this study in Nature Genetics, but we would like to consider your response to these ongoing concerns in the form of a further revision before we make a final decision on publication.

To guide the scope of the revisions, the editors discuss the referee reports in detail within the team, including with the chief editor, with a view to identifying key priorities that should be addressed in revision, and sometimes overruling referee requests that are deemed beyond the scope of the current study. In this case, we ask that you estimate the prevalence of CADM1 mutation across the study

cohorts (incorporating the newly discovered CADM1 mutation in addition to the five currently reported) and address all other comments with clarifications and revisions to the text and display items where warranted. We hope you will find this prioritized set of referee points to be useful when revising your study. Please do not hesitate to get in touch if you would like to discuss these issues further.

We therefore invite you to revise your manuscript taking into account all reviewer and editor comments. Please highlight all changes in the manuscript text file. At this stage we will need you to upload a copy of the manuscript in MS Word .docx or similar editable format.

*2) If you have not done so already please begin to revise your manuscript so that it conforms to our Letter format instructions, available [here](http://www.nature.com/ng/authors/article_types/index.html). Refer also to any guidelines provided in this letter.

[redacted]

We hope to receive your revised manuscript within 4-8 weeks. If you cannot send it within this time, please let us know.

Sincerely,
Kyle

Kyle Vogan, PhD
Senior Editor
Nature Genetics
<https://orcid.org/0000-0001-9565-9665>

Referee expertise:

Referee #1: Genetics, hypertension, primary aldosteronism

Referee #3: Organogenesis, adrenal development

Reviewers' Comments:

Reviewer #1:
Remarks to the Author:

This manuscript has improved compared to the last version. Specifically, additional tumors with CADM1 mutations have been identified, stainings have been improved, and functional studies have been expanded. My comments have also largely been addressed.

Questions that arise from reading this new version:

- What is the prevalence of these mutations in APAs?

This could now be addressed, after addition of another large cohort. The "2.5%" shown in extended Data Table 1 only refers to one case in the small discovery cohort. The true numerator is now 5. What is the denominator? 40 samples from the Brown lab WES study + X other samples from the Brown lab + 53 from UK and Japan + 70 from Germany without mutations in known genes + X from Germany with mutations in known genes + 43 from France without mutations in known genes + X from France with mutations in known genes. Given that approximately 95% of tumors have mutations in known genes, the number of samples with mutations in known genes should be much larger than the number

of samples without mutations in known genes.

- The clock gene part remains speculative, in particular with lack of clinical data (time of measurement, apparently no variability of aldosterone measurements in the three new patients). I would suggest to shorten this part and remove the speculation on a contribution of CADM1 to unrecognized primary aldosteronism.

- II. 243-249 seem out of place and are very hard to understand.

- It also remains somewhat unclear to me why inhibition of gap junction signaling increases aldosterone production and whether this is the only mechanism underlying CADM1-induced aldosteronism. As discussed before, the difference in the magnitude of effects (CADM1 mutation versus gap junction inhibition) argues against gap junctions as single or major mechanism. This should at least be discussed.

Reviewer #3:

Remarks to the Author:

The authors have provided additional data to support their conclusion that somatic mutations in CADM1 are a cause of APA and that gap junctions serve to negatively regulate aldosterone production. I will focus my comments on the authors' "Main changes made to the revised submission."

1) Co-localisation of GJA1 with several markers of ZG cells, using immunofluorescence in serial or confocally analysed sections.

These data remain unconvincing to this reviewer. While the addition of Dr. Murray's expertise to the manuscript is welcome, she also acknowledges significant uncertainty in this area of investigation. For example, she notes "The smaller and fewer numbers of gap junction plaques at points of cell contact in the ZG would generally be thought to suggest that this zone is less dependent on cell-cell communication. However, the question of the relationship between numbers and sizes of gap junctions and functional capacity has not been studied extensively enough." While annular gap junctions might be present as rare events in normal ZG, it remains unclear how this translates into a mechanism, which when inhibited, that can lead to unregulated aldosterone production in APAs and in aldo-producing micronodules.

2) Discovery of CADM1 somatic mutations in aldosterone-producing adenomas (APAs). Includes the finding of 3 new patients with CADM1 mutations in the exact same residues as the first 2 reported patients.

While the identification of three new patients with mutations in CADM1 from the French cohort, using directed sequencing, is interesting, it is not at all certain that these mutations represent the driver mutations for these APAs. To convince this reviewer that other mutations do not exist in these adenomas would require WES. Additional discrepancies (both clinically and morphologically) make these 3 new patients seem distinct from the initial 2 patients presented. First, unlike the first two

patients, which reportedly had “cyclic” disease and did not present with a suppressed renin level, the 3 new patients appear to have had a more classical presentation with fully suppressed renin levels (Fig 1c). Second, the IHC data shown for patients P4 and P5 are nowhere near as compelling as the original data (P2) (Ex Data 1i). The staining appears non-specific.

3) Regarding Patient 1, (Ex Data 1d), if upregulation of CADM1 expression is a result of a somatic mutation in this gene (the main point of this paper), why is CADM1 also upregulated in an unrelated (APM) Aldosterone-producing micronodule (see blue box)? The authors claim that this APM is “adjacent” to the APA (therefore implying it is somehow related) is misleading.

4) Regarding the claim in New Fig. 5 showing IFC for GJA1 in subcapsular cells is not expressing CYP17A1.

It remains unclear how one cell with an annular staining pattern for GJA1 (that is non Cyp17A1+ cell) is compelling evidence for GAP junctions playing a key role in regulating ZG cells. Lack of Cyp17A1 expression does not automatically make such a cell a ZG cell. This same image also shows CYP17A1 cells being adjacent to the capsule, which is highly unusual.

5) Regarding New Extended Data Fig. 2. GJA1 expression in human adrenals.

Taken together, these data are unconvincing. As one example, in Ex Data Fig 2F, despite the presence of a large APA in this image, which should suppress CYP11B2 expression in normal ZG cells, the authors are able to detect multiple examples of small clusters of CYP11B2+ cells. The concern is raised for non-specific staining with the CYP11B2 antibody.

In addition, it is not appropriate to argue (as in Ex Data Fig 2G) that the APM regions have “less GJA1 expression.” The concern raised by this reviewer and review #1 is that expression is low to begin with in the ZG. The authors have not presented a quantitative argument to support this conclusion.

6) Regarding New Extended Data Fig. 3. AQP2 expression in CADM1-mutant APAs. Includes more IHC data showing AQP2 expression in CADM1-mutant APAs and increased gene expression of AQP2 in transduced human adrenocortical cells transduced with mutant CADM1.

These new data appear to be nonspecific.

Author Rebuttal, first revision:

[Reviews in blue, responses in black].

Reviewer #1 (Remarks to the Authors):

This manuscript has improved compared to the last version. Specifically, additional tumors with *CADM1* mutations have been identified, stainings have been improved, and functional studies have been expanded. My comments have also largely been addressed.

We thank the reviewer for your appreciation of the changes, and further suggestions to improve the manuscript. As more of our colleagues add *CADM1* to their NGS screens, we hope to detect further tumours with the mutations. A 6th example is included in this revision (**New Figure 1c**).

Questions that arise from reading this new version:

- What is the prevalence of these mutations in APAs?

This could now be addressed, after addition of another large cohort. The "2.5%" shown in extended Data Table 1 only refers to one case in the small discovery cohort. The true numerator is now 5. What is the denominator? 40 samples from the Brown lab WES study + X other samples from the Brown lab + 53 from UK and Japan + 70 from Germany without mutations in known genes + X from Germany with mutations in known genes + 43 from France without mutations in known genes + X from France with mutations in known genes. Given that approximately 95% of tumors have mutations in known genes, the number of samples with mutations in known genes should be much larger than the number of samples without mutations in known genes.

We agree with the reviewer that the discovery of further mutations allows greater precision. We apologise if the denominators of some labs are not as 'clean' as the reviewer would wish because samples sent for WES were not necessarily all those in which no known mutation was found, and reference laboratories do not know how selection of tumours for referral was made. For the Brown lab, X other samples is 12. For France, X samples with known mutations is 281. For the German cohort WES was performed after exclusion of *KCNJ5* mutations by Sanger Sequencing. We originally stated the denominator as 70. When we looked back at our correspondence with the German investigator, we realised that he actually said 'roughly 70' at a time when he had recently moved from Munich to Zurich. In response to the reviewer's request for more precision about prevalence, we asked the investigator to go back to the original lab, and the correct denominator is 81. He also clarified that these 81 were samples "in which no gene with recurrent somatic mutations had been found, other than those previously reported (**page 7 line 48**)", i.e. no new gene mutation was identified in more than one APA. For the new US cohort, 1 *CADM1* mutation was identified out of 200 APAs sequenced. This denominator is influenced by the criteria set by different referring centers for sending samples for sequencing. We do not have information on what these criteria are. A conversation with the Penn doctors, who referred the sample with the sixth *CADM1* mutation, suggests that many of their samples are from African American patients, in whom our Michigan colleague (Bill Rainey) has already reported a different relative prevalence of the commoner mutations.

The information on the different cohorts are now made available to the readers in the new Extended Data Table 1c. The reviewer is correct that 95% sensitivity can be achieved by targeting CYP11B2-positive areas of the tumour, but this was not undertaken in most cases.

- The clock gene part remains speculative, in particular with lack of clinical data (time of measurement, apparently no variability of aldosterone measurements in the three new patients). I would suggest to shorten this part and remove the speculation on a contribution of CADM1 to unrecognized primary aldosteronism.

We agree with the reviewer that we have not demonstrated a connection between the clinical presentations of the first two patients and finding (in the transduction experiment) that biological rhythms was the most upregulated pathway, with a reciprocal effect on the two out-of-phase (E-box and RORE) set of transcription factors. Indeed it is likely that, if there is a link, it will be because exaggerated diurnal variation of aldosterone introduces a need for aldosterone (like cortisol) to be measured in the early morning, rather than because of truly cyclical aldosteronism. This is the speculation of the Boston authors of the Ann Int Med paper showing a doubling of PA diagnosis using 24h urine rather than blood measurements of aldosterone (**reference 75**). We now have a research grant testing this speculation, and whether CADM1 mutations will be found in patients diagnosed by 24h urine measurements, but this is clearly for the future. We therefore agree with the Reviewer's recommendation to shorten the section on clock genes (**page 15, l. 266 to page 16, l. 276**) and have removed the possible link between index case and clock data. We hope that the reviewer is content for us to retain the reference to the factual publication from the Boston authors (**page 17, l. 296 – 299**).

- ll. 243-249 seem out of place and are very hard to understand.

These lines refer to the diverse actions of CADM1, and were inserted in response to the reviewer's question, which we take to be re-iterated in his first sentence below, whether gap junction inhibition is the sole mechanism underlying CADM1-induced aldosteronism. The 3 partner domains in the C-terminus, for each of PDZ, 4.1B, MPP3, point to multiple potential actions of CADM1, and probably account for some of our observed consequences of silencing CADM1. However, we have not investigated them, and do not need to involve them in explaining upregulation of aldosterone production (see next section). We have therefore abridged these lines (Page 8 lines 85-90), and can delete entirely if the reviewer or Editor wishes.

- It also remains somewhat unclear to me why inhibition of gap junction signaling increases aldosterone production and whether this is the only mechanism underlying CADM1-induced aldosteronism. As discussed before, the difference in the magnitude of effects (CADM1 mutation versus gap junction inhibition) argues against gap junctions as single or major mechanism. This should at least be discussed.

May we take the last point first, since this seems the only point at which reviewer 1 queries the significance of what we have discovered. We wonder whether the reviewer may have missed our new data in Figure 5b, or its log scale, showing 1-2 orders of magnitude increase in CYP11B2 expression and aldosterone secretion? This is comparable to, or even greater than, the increases following transduction of mutant CADM1 (15-30-fold). On inspection of the data in **Figure 5b**, re-drawn here with a non-log scale, we hope the reviewer would agree that ‘the magnitude of effects argue for gap junctions as a major mechanism’.

b. Mimetic peptide Gap27 increases CYP11B2 expression and aldosterone production in a dose response manner in angiotensin II-stimulated H295R cells

The DMSO control in the figure was prompted by reviewer 3, who questioned whether stimulation of aldosterone production by Gap27 was due to decreased cell viability. The large increase of CYP11B2 expression further negates the idea of ‘leakage’ of aldosterone from the cells (CYP11B2 data redrawn now on a non-log scale for clarity).

The reviewer asks why gap junction inhibition should increase aldosterone production, and whether it is the sole mechanism. As reported in the manuscript, we were prompted to investigate the gap junctions because of previous observations that their inhibition in another endocrine gland, pancreatic islets, increases glucagon secretion (**page 9, l.132-133**). **Reference 40** draws attention to the opposite effects of gap junctions in alpha and beta islet cells, which is an intriguing parallel to the apparent situation in

ZG and ZF of adrenal cortex. We have added a further relevant references (Benninger et al. J Physiol 2011;589:5453– 5466, **Reference 60**), together with references to other tissues where small or sparse gap junctions are important (**Page 13 l. 204-206**). There are also several papers on the role of gap junctions in synchronising pulsatile insulin secretion, and (in other tissues) synchronising clocks. Mindful, however, of the reviewer's advice not to speculate in the manuscript, we have not done so, and limited the citations.

Reviewer #3 (Remarks to the Authors):

The authors have provided additional data to support their conclusion that somatic mutations in *CADM1* are a cause of APA and that gap junctions serve to negatively regulate aldosterone production. I will focus my comments on the authors' "Main changes made to the revised submission."

We thank the reviewer for his further detailed comments. However, these convey – at least to us – the impression that reviewer 3 has a well-grounded view of how things should look in the rodent adrenal cortex and therefore concludes that our findings must be wrong if they infer otherwise in human adrenal. The remarks, below, that [i] the recurrent somatic mutations in two adjacent residues of the TM domain of *CADM1* are likely to be incidental findings, and that [ii] the findings of CYP11B2-positive APCCs (aldosterone producing cell clusters) adjacent to APAs reflect non-specificity of Celso Gomez Sanchez's anti-CYP11B2 antibodies, are (we respectfully suggest) extreme examples of what can happen if a reviewer has a preconceived notion of what authors should have discovered.

1) Co-localisation of GJA1 with several markers of ZG cells, using immunofluorescence in serial or confocally analysed sections.

These data remain unconvincing to this reviewer. While the addition of Dr. Murray's expertise to the manuscript is welcome, she also acknowledges significant uncertainty in this area of investigation. For example, she notes "The smaller and fewer numbers of gap junction plaques at points of cell contact in the ZG would generally be thought to suggest that this zone is less dependent on cell-cell communication. However, the question of the relationship between numbers and sizes of gap junctions and functional capacity has not been studied extensively enough." While annular gap junctions might be present as rare events in normal ZG, it remains unclear how this translates into a mechanism, which when inhibited, that can lead to unregulated aldosterone production in APAs and in aldoproducing micronodules.

A brief answer, to reassure the reviewer, is that the two sentences in inverted commas were intended within the context of the rest of Dr Murray's report (as submitted in full to the Editor), to convey the opposite interpretation to the reviewer's. She is clear that we have found functioning gap junctions in human ZG as stated in her report – **“Given the morphological and functional evidence of gap junctions in the ZG...it is clear Cx43 proteins are present in the ZG, (and) that they form small Cx43 gap junctions capable of dye communication between cells”**. It will be important in future studies to work out why they are small and scarce, and why their blockade causes large increases in aldosterone production. Possible mechanisms for the latter were offered at the end of our responses to reviewer 1, and in general low abundance of rate-limiting steps in pathways is a clue for rather than against their importance.

A longer answer is that we suspect the difference of opinion with the reviewer about data interpretation may lie in the differences between mouse and human ZG. This is enhanced by the recent report in Sci Adv of a different origin for adrenal in rodents and primates (doi:

10.1126/sciadv.abn8485.)

Adult human ZG differs from other species in that the main enzyme (CYP11B2 encoding aldosterone synthase) is almost entirely switched off except in microscopic clusters which usually have somatic mutations of CACNA1D, or sometimes other aldosterone-driver genes. From the perspective of an expert in rodent adrenal, scepticism about presence or role of gap junctions in ZG is understandable. From the perspective of those studying human adrenal physiology and pathophysiology, the suppression of CYP11B2 (aldosterone synthase) in most of ZG is, as yet, not adequately explained, and it may well be that the explanation lies in a pathway absent from rodent adrenal.

Circulating renin and angiotensin, the principle agonist to aldosterone production, are unsuppressed in most humans, including (indeed, especially) our PA patients, since they are receiving doses of spironolactone at the time of adrenalectomy, which have often been titrated to relieve suppression of plasma renin. The finding of an inhibitory pathway, namely gap junctions, within the ZG, and that expression is diminished in the aldosterone-producing microclusters, is therefore of considerable interest. To overlook this on the grounds of rodent adrenal behaviour would be unfortunate, especially following the Cheng et al paper cited above.

The reviewer's major concern with the original manuscript was 'its disregard for the consensus in the adrenal field that the normal ZG layer (including in humans) essentially does not contain gap junctions', citing Dr Murray's 2016 review. We too had cited the review, and on re-reading it and the cited source data, did not take the same message as the reviewer, but respectfully took the comments on board and conducted further work as suggested. This extensive qualitative and semi-quantitative data documenting presence of gap junctions in ZG failed to persuade this reviewer, who commented in the

second review that *'the CX43 immunostaining data is certainly not marking gap junctions ... its presence in the cytoplasm or as puncta rather than at the membrane undermines the authors primary hypothesis and conclusion.'* It was at this point, recognising (as did reviewer 3) that Dr Murray is the doyenne of the field, that we approached her directly, providing her with our experimental material including additional experiments as requested by the reviewers and the reviewers' comments, for a detached, second opinion. After detailed consideration, she could have agreed with the reviewer, and concluded that gap junctions are too few or small to be of possible consequence in ZG. Instead, her comments, included within the last rebuttal, refer to other examples where functional gap junctions are small and punctate, and concludes that **'convincing evidence was presented in this manuscript that there are small areas of Cx43 punctate staining at the areas of cell-cell contacts, consistent with the presence of gap junction plaques ... and makes a strong argument for a physiological role for the gap junction regulation of events in ZG.'**

We fully acknowledge that further questions about mechanism can now be investigated, as discussed above with reviewer 1. Mechanistic questions need to start from the demonstration that gap junctions exist in ZG, and that their inhibition increases aldosterone production. Both of these observations and the supporting evidence were previously questioned by reviewer 3, who requested that we repeat the immunostaining with his preferred ZG markers, DAB2 and Gq. We have done that and added another widely used ZG marker, VSNL1. The reviewer concedes (*'annular gap junctions might be present as rare events'*) that GJs are present, but is now concerned that there are not enough. Our literature citations plus supplementary video illustrate how fleeting annular GJs are within the life-cycle of a gap junction protein, making significant the chance capture of even a single AGJ, within a high magnification image at a single point in time. As observed in our opening paragraph, the sparsity of GJs in ZG is not an argument against their importance. We have now added as further supplementary evidence, a confocal image that co-localizes GJA1 with WGA, a cell membrane marker, in ZF, ZG, and APM (**New Supplementary Fig. 5e**) which reinforces that the immunostaining data is clearly marking gap junctions.

2) Discovery of CADM1 somatic mutations in aldosterone-producing adenomas (APAs). Includes the finding of 3 new patients with CADM1 mutations in the exact same residues as the first 2 reported patients.

While the identification of three new patients with mutations in CADM1 from the French cohort, using directed sequencing, is interesting, it is not at all certain that these mutations represent the driver mutations for these APAs. To convince this reviewer that other mutations do not exist in these adenomas would require WES. Additional discrepancies (both clinically and morphologically) make these 3 new patients seem distinct from the initial 2 patients presented. First, unlike the first two patients,

which reportedly had “cyclic” disease and did not present with a suppressed renin level, the 3 new patients appear to have had a more classical presentation with fully suppressed renin levels (Fig 1c). Second, the IHC data shown for patients P4 and P5 are nowhere near as compelling as the original data (P2) (Ex Data 1i). The staining appears non-specific.

The short answer here is surprise, even disbelief, at these suggestions. On balance of probabilities, it is implausible to us that a pair of adjacent charge-changing transmembrane domain mutations found in five (now six) patients could be considered more likely to be incidental – and their tumour staining for CYP11B2 (aldosterone synthase) and CADM1 to be non-specific – than that the 15-30 fold increase in CYP11B2 seen on transduction of the mutations caused the primary aldosteronism in these five patients. These increases were seen on transduction of either the longer or shorter isoform of either variant (i.e. 4 independent experiments) into adrenocortical cells. The comparison in each case was to CYP11B2 expression and aldosterone secretion when the wild-type isoforms were similarly transduced.

The reviewer’s suggestion appears to originate in his view that all samples used for replication of newly discovered mutations should themselves undergo whole exome sequencing. With respect, this is inappropriate. Whole exome sequencing is undertaken in the discovery cohort, and targeted sequencing in the replication cohorts. The reviewer’s requirement for WES of all replication samples does not reflect practice, e.g. when work for previous Science and Nat Gen papers were undertaken which correctly identified functional somatic mutations in *KCNJ5*, *ATP1A1*, *ATP2B3*, *CACNA1D*, *GNA11*, *GNAQ*. To reassure the reviewer (and to answer reviewer’s 1 question on prevalence), we have now included a breakdown by cohorts of the number of WES performed (n=168) that would have been able to identify co-existence of CADM1 mutations with other known aldosterone-driver mutations (**New Extended Data Table 1c**).

Variability in clinical phenotype is one of the challenges in detecting clear genotype:phenotype pictures, coupled sometimes to a relative lack of clinical data in centres which do a lot of sequencing for referred samples. In our Nat Gen paper last year, the links between pregnancy, and double-mutation of CTNNB1 with GNA11/Q, would probably not have come to light in France, and even minor germline differences across the English Channel might underlie why 6/12 French but 0/10 British patients had solitary CTNNB1 mutations.

Regarding variable immunohistochemistry for CYP11B2 and CADM1, the variability is in resolution of the photograph, not specificity of the same antibody as used for previous tumours. It is still clear that there is abundant membranous CADM1 protein expression, which is the main message of the figure (since non-membrane mutations of CADM1, in small-cell lung cancer, cause complete loss of the protein); and Celso Gomez Sanchez’s antibody for CYP11B2 is one of the most specific of all antibodies (see response to [5], below). We apologise for the low resolution of the CADM1 images, which have now been replaced (**New Extended Data Fig. 1i**).

3) Regarding Patient 1, (Ex Data 1d), if upregulation of CADM1 expression is a result of a somatic mutation in this gene (the main point of this paper), why is CADM1 also upregulated in an unrelated (APM) Aldosterone-producing micronodule (see blue box)? The authors claim that this APM is “adjacent” to the APA (therefore implying it is somehow related) is misleading.

With respect, the reviewer is mistaken. We have not said (or intentionally implied) that CADM1 is upregulated as a result of its somatic mutation, still less that this is the main point of the paper. Early in the manuscript we cite Williams et al who had previously reported ‘Cell Adhesion Molecule 1 (CADM1) as a transcript upregulated in [all] APAs.¹²’ If upregulation of CADM1 by its mutation had been present, it would have been reported alongside other upregulated genes, like AQP2, in **Supplementary Table 5**.

4) Regarding the claim in New Fig. 5 showing IFC for GJA1 in subcapsular cells is not expressing CYP17A1.

It remains unclear how one cell with an annular staining pattern for GJA1 (that is non Cyp17A1+ cell) is compelling evidence for GAP junctions playing a key role in regulating ZG cells. Lack of Cyp17A1 expression does not automatically make such a cell a ZG cell. This same image also shows CYP17A1 cells being adjacent to the capsule, which is highly unusual.

The short answer is, as above, that Dr Murray reviewed all our pictures, and reached her conclusion as an ‘independent witness’. Among the several images, IHC or IF, of gap junctions, some were chosen to illustrate multiple examples in the field (generally punctate which Dr Murray could appreciate as gap junctions) and others to show individual annular gap junctions, as evidence of dynamic gap junction formation.

As discussed above (and in the manuscript and previous rebuttal), we used multiple stains for ZG cells, some at reviewer 1’s and reviewer 3’s request, so that our conclusion was not based on any one of these conditions. Absence of CYP17A1 staining was just one of these conditions. Historically, and in routine clinical practice, ZG is recognisable to clinical pathologists from the appearance and position of the cells: compact cells between capsule and the larger, lipid-rich cells of ZF, with dentate, not round, nuclei. The best impression of GJA1 distribution in ZG is the immunohistochemistry already presented in the previous version of the paper and RNA expression in LCM samples, however the reviewers requested further stainings with ZG markers. Immunofluorescence allows for multiple stainings and at higher magnification/resolution allows visualization of annular gap junctions. On the other hand, a well-known downside of immunofluorescence is autofluorescence, which can be particularly problematic in adult adrenals. High-power fields, in areas where autofluorescence is confirmed to be minimal based on serial controls, allowed the capturing of annular gap junctions as the most specific evidence of gap junction

formation. This should be taken together with both low-power field of immunofluorescence Supplementary Fig. 4d

(ZG identified by the ZG marker DAB2 as requested by the reviewer) and immunohistochemistry data presented in **Extended Data Fig. 2** (ZG identified by compact cell with dentate nuclei and location adjacent to capsule) as evidence of gap junctions in ZG cells. To note, in adult human adrenals (unlike rodent adrenal which expresses little CYP17A1), CYP17A1 is often seen close to the capsule as ZG is often atrophic (e.g. Figure 1 in Celso Gomez Sanchez landmark paper (MCE, 2014;392:73-9)).

5) Regarding New Extended Data Fig. 2. GJA1 expression in human adrenals.

Taken together, these data are unconvincing. As one example, in Ex Data Fig 2F, despite the presence of a large APA in this image, which should suppress CYP11B2 expression in normal ZG cells, the authors able to detect multiple examples of small clusters of CYP11B2+ cells. The concern is raised for non-specific staining with the CYP11B2 antibody.

With respect, this comment reveals lack of familiarity with human adrenocortical literature of the last 10 years – since APCCs (APMs) were first reported by Nishimoto et al in 2010, and their routine visualisation made possible by Gomez Sanchez's wonderful CYP11B2 monoclonal, (MCE paper already mentioned, of 2014). The APCCs frequently have CACNA1D mutations and are postulated to be precursors to the pathology.

The field now recognises the term 'AATL = APCC to APA transitioning lesion', and the picture below illustrates multiple APCCs, highlighted by blue arrows, adjacent to an APA, from the

Nishimoto et al 2017 paper in MCE which first reported AATLs. The aldosterone from the APA does not suppress the CYP11B2 in the APMs because the latter are themselves autonomous, and frequently found to have somatic mutations, mainly of CACNA1D.

For those working in the adrenocortical field, Gomez Sanchez's generously donated CYP11B2 antibody is regarded as one of the most sensitive and specific of all antibodies, differentiating – as in the figure above – between CYP11B2 in blue and the 93% identical CYP11B1 in brown. Its impact is illustrated by some 45 papers since 2014 in which he is coauthor (including our double-mutant paper in Nat Gen) and

a similar number in which he is acknowledged (as in the first line of Acknowledgements of the present manuscript).

In case a second example is needed, to illustrate the widespread occurrence of APCCs adjacent to an APA, the second figure, showing 3 boxed APCCs, is from Gomez Sanchez's own review *Horm Metab Res.* 2020; 52: 421–426. (Immunohistochemistry Of The Human Adrenal Cyp11b2 In Normal Individuals And In Patients With Primary Aldosteronism)

In addition, it is not appropriate to argue (as in Ex Data Fig 2G) that the APM regions have “less GJA1 expression.” The concern raised by this reviewer and review #1 is that expression is low to be begin with in the ZG. The authors have not presented a quantitative argument to support this conclusion.

The reviewer most likely missed Supplementary Fig. 5c which shows reduced staining for GJA1 in APMs, quantified across 14 adrenals ($p < 0.001$). The scoring used by the pathologists is illustrated in Supplementary Fig. 5d and representative immunofluorescence of an adrenal section co-stained with GJA1 and either a ZF marker (CYP17A1), a ZG marker (VSNL1), or an APM marker (CYP11B2) that illustrate the gradient is now shown in **New Supplementary Fig. 5e**.

6) Regarding New Extended Data Fig. 3. AQP2 expression in CADM1-mutant APAs. Includes more IHC data showing AQP2 expression in CADM1-mutant APAs and increased gene expression of AQP2 in transduced human adrenocortical cells transduced with mutant CADM1.

These new data appear to be nonspecific.

With respect, it is hard to follow this reviewer's criteria for use of 'non-specific'. The positive tissue control staining for AQP2, in **Extended Data Fig. 3a**, shows AQP2 only in a subset of renal tubular cells, predominantly luminal membrane, partly cytoplasm, just as would be expected.

Decision Letter, second revision:

Our ref: NG-A55401R4

2nd September 2022

Dear Morris,

Your revised manuscript "Somatic intramembranous mutations of CADM1 in aldosterone-producing adenomas, and gap junction-dependent regulation of aldosterone production" (NG-A55401R4) has been seen again by Reviewer #1. As you will see from the comments below, Reviewer #1 is generally satisfied with your responses to the previous concerns, and therefore we will be happy in principle to publish your work in Nature Genetics as an Article pending final revisions to satisfy Reviewer #1's remaining requests and to comply with our editorial and formatting guidelines.

We are now performing detailed checks on your paper and we will send you a checklist detailing our editorial and formatting requirements soon. Please do not upload the final materials or make any revisions until you receive this additional information from us.

Thank you again for your interest in Nature Genetics. Please do not hesitate to contact me if you have any questions.

Sincerely,
Kyle

Kyle Vogan, PhD
Senior Editor
Nature Genetics
<https://orcid.org/0000-0001-9565-9665>

Reviewer #1 (Remarks to the Author):

Most of my comments have been addressed. The prevalence of CADM1 mutations is unfortunately not correctly calculated. Samples with known mutations that were excluded are not indicated for several cohorts. These values should be available from the co-authors. For example, it should be known how many samples in the German cohort had KCNJ5 mutations and were excluded, or how many others with recurrent somatic mutations were excluded. I would recommend to obtain these values from co-authors and include them in the manuscript.

Please also note the typo in the title of Suppl. Fig. 1 (tertiery -> tertiary).

Final Decision Letter:

20th April 2023

Dear Morris,

I am delighted to say that your manuscript "Somatic CADM1 mutations in aldosterone-producing adenomas and gap junction-dependent regulation of aldosterone production" has been accepted for publication in an upcoming issue of Nature Genetics.

Your paper will be published online after we receive your corrections and will appear in print in the next available issue. You can find out your date of online publication by contacting the Nature Press Office (press@nature.com) after sending your e-proof corrections. Now is the time to inform your Public Relations or Press Office about your paper, as they might be interested in promoting its publication. This will allow them time to prepare an accurate and satisfactory press release. Include your manuscript tracking number (NG-A55401R5) and the name of the journal, which they will need when they contact our Press Office.

Before your paper is published online, we will be distributing a press release to news organizations worldwide, which may very well include details of your work. We are happy for your institution or

funding agency to prepare its own press release, but it must mention the embargo date and Nature Genetics. Our Press Office may contact you closer to the time of publication, but if you or your Press Office have any enquiries in the meantime, please contact press@nature.com.

Please note that Nature Genetics is a Transformative Journal (TJ). Authors may publish their research with us through the traditional subscription access route or make their paper immediately open access through payment of an article-processing charge (APC). Authors will not be required to make a final decision about access to their article until it has been accepted. [Find out more about Transformative Journals](https://www.springernature.com/gp/open-research/transformative-journals)

Authors may need to take specific actions to achieve [compliance](https://www.springernature.com/gp/open-research/funding/policy-compliance-faqs) with funder and institutional open access mandates. If your research is supported by a funder that requires immediate open access (e.g. according to [Plan S principles](https://www.springernature.com/gp/open-research/plan-s-compliance)), then you should select the gold OA route, and we will direct you to the compliant route where possible. For authors selecting the subscription publication route, the journal's standard licensing terms will need to be accepted, including [self-archiving-and-license-to-publish](https://www.nature.com/nature-portfolio/editorial-policies/self-archiving-and-license-to-publish). Those licensing terms will supersede any other terms that the author or any third party may assert apply to any version of the manuscript.

Please note that Nature Portfolio offers an immediate open access option only for papers that were first submitted after 1 January 2021.

If you have not already done so, we invite you to upload the step-by-step protocols used in this manuscript to the Protocols Exchange, part of our on-line web resource, natureprotocols.com. If you complete the upload by the time you receive your manuscript proofs, we can insert links in your article that lead directly to the protocol details. Your protocol will be made freely available upon publication of your paper. By participating in natureprotocols.com, you are enabling researchers to more readily reproduce or adapt the methodology you use. [Natureprotocols.com](https://natureprotocols.com) is fully searchable, providing your protocols and paper with increased utility and visibility. Please submit your protocol to <https://protocolexchange.researchsquare.com/>. After entering your [nature.com](https://www.nature.com) username and password you will need to enter your manuscript number (NG-A55401R5). Further information can be found at <https://www.nature.com/nature-portfolio/editorial-policies/reporting-standards#protocols>

Sincerely,
Kyle

Kyle Vogan, PhD
Senior Editor
Nature Genetics
<https://orcid.org/0000-0001-9565-9665>